# Global remapping of the sensory homunculus emerges early in childhood development

Raffaele Tucciarelli [1,2,6] ✉, Laura Bird[3], Zdenek Straka [4], Maggie Szymanska[1], Mathew Kollamkulam[2], Harshal Arun Sonar[5], Jamie Paik [5], Danielle Clode[1], Matej Hoffmann [4], Dorothy Cowie [3,6] ✉ & Tamar R. Makin [1,2,6] ✉

Some of the most dramatic examples of neuroplasticity in the human brain follow congenital sensory deprivation, yet the plasticity mechanisms producing this large-scale cortical remapping remain poorly understood. Congenital malformation of the upper-limb provides a unique temporal dissociation of developmental plasticity mechanisms: While sensory deprivation from the absent hand is triggered before birth, compensatory motor behaviours develop gradually throughout childhood. Using paediatric neuroimaging and semi-ecological behavioural analysis in children (5-7 years old) and adults (>25 years old) with unilateral upper-limb congenital limb difference, we studied deprivation- and use-dependent plasticity in the deprived primary somatosensory cortex and beyond. We reveal that global remapping, encompassing the entire sensory homunculus, is established early and maintained in adulthood. Modelling indicates that deprivation-driven homeostatic plasticity can account for this global remapping. Hebbian-based compensatory learning further contributes to the magnitude of inter-individual differences observed at both childhood and adulthood. Our findings emphasise the early establishment and stability of cortical maps, despite extensive daily-life behavioural adaptation.

The brain's ability to adapt to unique sensory and motor experiences remains one of the most compelling mechanisms within neuroscience. Particularly striking illustrations of brain remapping are apparent in individuals born with different abilities, such as congenital blindness or upper limb differences, providing naturalistic paradigms through which we can study the capacity for plasticity. In congenitally blind individuals, for example, in the absence of visual input, the visual cortex shows enhanced activity for tactile[1,2] and auditory stimuli[3,4]. Similarly, individuals born with upper limb difference exhibit increased activity in response to movements of body parts such as the lower face, residual arm and feet, in their deprived (missing) sensorimotor hand cortex[5–10]. This extensive remapping of multiple body parts into the deprived cortex has been considered to reflect the brain's intrinsic ability to repurpose its architecture in response to developmental variations[11]. Traditionally, these striking differences in sensory maps were attributed to the deprivation of neurotypical sensory input[12], which triggers intrinsic homeostatic plasticity mechanisms[13]. The primary function of homeostatic plasticity is to provide stabilising mechanisms that regulate changes in the system, preventing activity levels from becoming excessively high or low[14]. One

[1]MRC Cognition & Brain Sciences Unit, University of Cambridge, Cambridge, UK. [2]Institute of Cognitive Neuroscience, University College London, London, UK. [3]Department of Psychology, Durham University, Durham, UK. [4]Department of Cybernetics, Faculty of Electrical Engineering, Czech Technical University in Prague, Prague, Czechia. [5]Reconfigurable Robotics Lab, EPFL, Lausanne, Switzerland. [6]These authors contributed equally: Dorothy Cowie, Tamar R. Makin. ✉e-mail: r.tucciarelli@ucl.ac.uk; dorothy.cowie@durham.ac.uk; tamar.makin@mrc-cbu.cam.ac.uk

such homeostatic mechanism is synaptic scaling: When there is a decrease in firing rate, the synaptic weights of the excitatory post-synapses on a cell are scaled up by a multiplicative factor, preserving the relative weights of the synapses. Within this framework, due to the underlying topography of the sensory cortex, inputs neighbouring the deprived cortex may have the physiological advantage to dominate the cortical territory of the missing input[15].

A second important facilitator for brain plasticity is the process of tuning receptive fields by Hebbian learning[16], which is thought to be experience-driven. Let us consider the lived experience of those growing up with one hand. For example, imagine a child, born with a single hand, encountering the challenge of opening a jar to reach a treat inside. Lacking the conventional approach afforded by two hands, the child embarks on a journey of exploration and adaptation. The brain orchestrates a series of trials—perhaps stabilising the jar between the knees or pressing it to the torso with the residual limb, maybe even using the mouth to pull the lid. This drive to adapt and overcome, fuelled by necessity and reward, exemplifies what we define as 'behavioural pressure'. This pressure is thought to be particularly potent in moulding changes in neural organisation and function during early ('sensitive') periods of development[17–19]. It has been speculated that these extraordinary behavioural shifts in individuals with different abilities guide (or are guided by) brain reorganisation[11]. For example, it has been suggested that the increased activity found in the deprived hand cortex for multiple other body parts reflects compensatory strategies in one-handed individuals[5,9,20]. This idea was instrumental in explaining why inputs of body parts that are not neighbouring the deprived hand cortex, such as the feet, could benefit from increased activity in the deprived cortex. This theory was further propelled by studies demonstrating that single-pulse TMS to the deprived hand M1 area of congenital limb difference (CLD) individuals elicited evoked responses in the feet[7,21] and lower face[22], demonstrating that the missing hand cortex supports motor control of other body parts that are not necessarily cortical neighbours. While conceptually compelling, previous studies struggled to find a one-to-one relationship between brain remapping and behaviour, e.g. on an individual level[5,23].

Revisiting the extreme examples where cortical maps diverge significantly from their neurotypical protoarchitecture, it is reasonable to expect a collaborative process between homeostatic and Hebbian plasticity[15]. In scenarios where developmental anomalies occur, such as the underdevelopment of a hand, residual inputs from other body parts, normally masked or not functionally relevant to the hand area, become potentiated. This unmasking is likely intensified by homeostatic plasticity processes, ensuring neural equilibrium in response to the altered sensory input[14]. Subsequently, behavioural demands, interwoven with top-down directive influences from higher brain functions, modulate these inputs to produce context-dependent responses—which are therefore behaviourally relevant. Paradoxically, this dynamic interplay between brain and behaviour may also explain why previous studies looking at links between adaptive behaviour and brain remapping in adults have failed to find direct correlations. It is because the behaviour most relevant for shaping brain remapping has likely occurred in earlier life, which has since altered through adulthood. Interestingly, this mechanism calls for the prediction that, given a change in adaptive behaviour between children and adults, children might display different cortical body maps than adults.

Yet, recent studies challenge this view by showing that the perinatal sensory cortex is already relatively mature. For example, a recent study in macaque monkeys has shown that, despite behavioural improvements in texture perception during early life, the neural features in areas V1, V2, V4 and inferotemporal cortex (IT) remain stable from the earliest ages tested[24] (see ref. 25 for related results in auditory cortex). Another study, examining body maps in developing monkeys, found that sensory hand maps are already present at birth and remain relatively stable even in visually impaired monkeys relying more heavily on tactile interactions[26]. These and other studies[27,28] indicate that impressive changes in early sensory experiences do not necessarily drive extraordinary changes in sensory cortical organisation. Instead, the neural representation in these sensory areas may be established much earlier than previously thought, suggesting a fundamental protoarchitecture present from birth. This perspective will predict that any brain changes observed in adults with congenital deprivation would already be established in their early life and resilient to behavioural changes.

We wanted to disentangle some of the mechanisms that will ultimately determine the distinct topography in adults with one missing hand due to a developmental difference, namely, deprivation and adaptive behaviour. We recruited people born with one missing hand due to a unilateral upper limb malformation (hereafter, congenital limb difference, or CLD). We studied their brain and behaviour in two age groups—children (age range 5–7 years; $n = 16$), and adults (age range 25–61 years; $n = 16$), and age-matched participants with neurotypical development (hereafter the control group, or CTR; 21 children and 16 adults). Inspired by mounting evidence that suggests a greater capacity for reorganisation in mid-level sensory cortex[11], we focused on higher-level S1 (Brodmann areas 1 and 2). We asked if behaviour and brain differences between children and adults with CLD are uniquely distinct from developmental differences observed in neurotypical development, and whether these brain and behavioural changes are linked. In other words, we asked whether differences in body topography found in the CLD population could be attributed to behavioural-dominant (Hebbian plasticity) changes, in addition to deprivation-related (Homeostatic) mechanisms, or simply age (development).

We predicted that children would display a more extensive behavioural repertoire than adults, and that this would be particularly relevant for compensatory strategies in children with CLD. We further predicted that this more diverse behaviour will cause greater expression of those body parts in CLD children's S1 versus both control children and CLD adults. We finally predicted that the cortical body maps of adults with CLD could be linked with the CLD children's behaviour.

## Results
### Greater behavioural diversity in children with upper limb differences

We first examined the behavioural compensatory adaptations by CLD children and adults in a task designed to mimic real-life object manipulation. We used a custom-designed battery of semi-ecological tasks, which would largely be completed using two hands in the typical population. Tasks, completed in a fixed order while sitting on a mat, included pulling a lid off a tin, unscrewing a nut from a small wooden bolt, and separating Lego bricks (for a complete list of the 15 tasks, see Supplementary Table 1). Behaviour was video recorded and analysed offline to ascertain which body parts were engaged and the proportion of time each body part was used during each of the sub-tasks (Fig. S1).

We first calculated 'co-use' for each sub-task, which takes into account how many body parts were used while also considering how consistently they were used. For example, while CTR participants tend to use both hands to open a tin box, CLD individuals might stabilise the tin with their legs, torso, residual arm (or any combination of these body parts), while lifting the lid with their intact hand. A higher index means more body parts are reliably used. A lower index means either fewer body parts are used, or the usage is uneven across the body parts. The average co-use indices are presented across groups in Fig. 1A. For control adults, the index averages just over 2, indicating they mainly completed the tasks bimanually. For all other groups, the indices are higher, hinting at more complex incorporation of additional body parts. CLD children showed values close to 3, meaning they

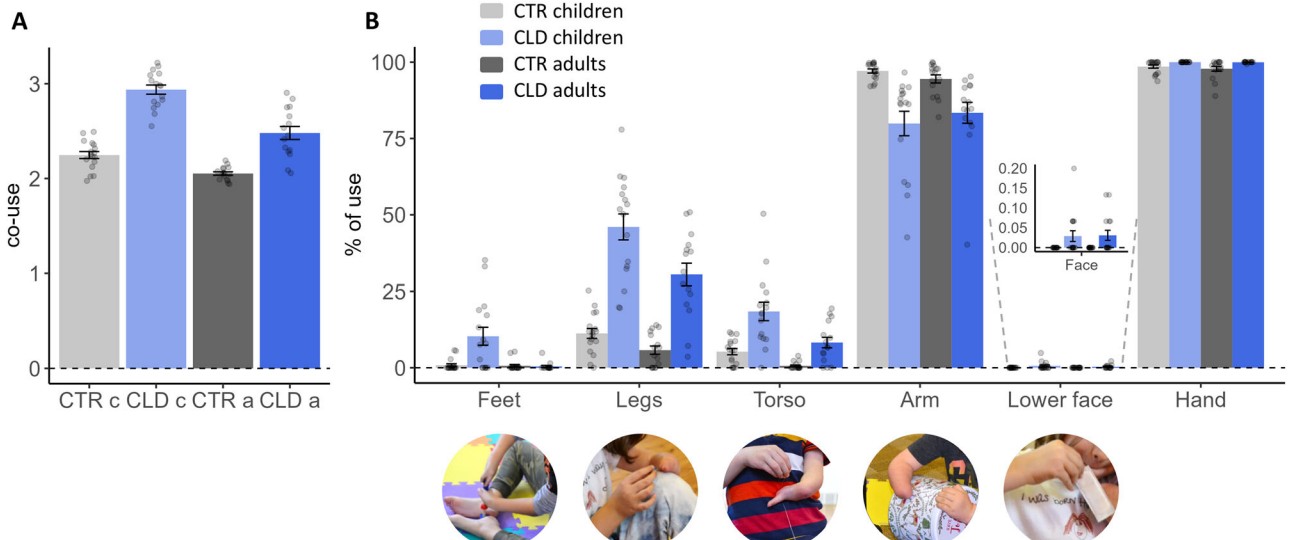

**Fig. 1 | Children with congenital limb difference (CLD) show more versatile compensatory behaviour with multiple body parts, relative to CLD adults and their peer age group.** Participants were asked to complete 15 different bimanual tasks to assess how various body parts are used in daily activities. **A** The co-use index determines how many body parts were consistently used to complete the tasks. Both children and adults with CLD (CLD c and CLD a, respectively) used more body parts to perform the tasks than their age-matched controls, as indicated by higher co-use indices. CLD children used more body parts than CLD adults, even when accounting for developmental effects. **B** The percentage of use measures the proportion of time each body part was used relative to the total task time. CLD participants, and children in particular, showed a higher percentage of use of their feet, legs, torso and lower face. The residual arm was similarly used by CLD children and adults, and less so in comparison to the CTR's non-dominant hand. These results indicate that individuals with CLD, in particular children, use a diverse range of body parts for compensation. Grey colours indicate CTR and blue individuals with CLD; Dark shades indicate adults and light shades indicate children. The bar graphs for the lower face are enlarged for visualisation purposes. Error bars indicate standard error of the mean (SEM). Images are provided to illustrate functional contributions by various body parts to aid the intact hand during task completion. Please note that multiple body parts might be simultaneously used, e.g. torso, residual arm and hand. CLD children ($N = 16$), CLD adults ($N = 15$), CTR children ($N = 17$), CTR adults ($N = 16$). Source data are provided as a Source data file.

incorporated, on average, nearly 3 different body parts to complete the tasks. Here, we found a significant interaction between age and limb difference ($F_{1, 60.04} = 8.08$, $p = 0.006$), alongside main effects for each of these factors ($F_{1, 60.03} = 49.94$, $p < 0.001$ for age; $F_{1, 60.04} = 128.19$, $p < 0.001$ for limb difference). The results provide evidence that physically more diverse compensatory behaviours are present in CLD children relative to both control children ($t_{60.10} = 10.84$, $p < 0.001$) and CLD adults ($t_{60.10} = -6.96$, $p < 0.001$).

Further analysis centred on the proportion of time each of the different body parts was used across the sub-tasks. The average values across the sub-tasks are presented in Fig. 1B for each of the four study groups. As expected, the intact hand (or dominant hand in controls) was universally engaged, with no significant differences across groups.

The omnibus analysis, which combined all body parts tested for non-manual compensatory purposes (i.e. excluding the intact hand), revealed a significant interaction between age and limb difference ($F_{1,60.08} = 6.28$, $p = 0.015$). This interaction shows that CLD children behave differently from the CLD adults, even after we account for overall developmental effects. The omnibus analysis also revealed main effects of age, limb difference and body part (all $p ≤ 0.001$, $F > 34.31$) (see Supplementary Table 3 for full results of the mixed models). It further identified interactions between body part and age ($F_{4625.23} = 12.80$, $p < 0.001$), meaning that different body parts are utilised more by children than adults; and an interaction between body part and limb difference ($F_{4625.22} = 156.50$, $p < 0.001$), which could be explained by the inclusion of the residual/non-dominant hand, which was under-used by the CLD population, relative to controls.

When analysing compensatory behaviours for each of the different body parts, we found that CLD relied most on their residual arm. While there were no differences between CLD children and adults in terms of frequency of residual arm use ($t_{544.99} = 1.55$, $p = 0.121$), both used the residual arm significantly less than control participants used their non-dominant hand ($F_{1,600.33} = 104.34$, $p < 0.001$; main effect of

limb difference). Compared to CTR, CLD participants exhibited a higher proportion of use of their feet, legs and torso (all $p ≤ 0.001$, all $F ≥ 11.17$). Age-related main effects were also found for feet, legs and torso (all $p ≤ 0.001$, all $F ≥ 13.18$), highlighting more non-manual exploratory behaviour in children relative to adults, and this was further confirmed in significant differences between CLD children and adults (all $p ≤ 0.001$, all $t ≥ 4.95$). This age-dependent diversification in body part usage was particularly pronounced in CLD children, even relative to control children (Feet: $t_{601.25} = 4.90$, $p < 0.001$; Legs: $t_{601.25} = 18.03$, $p < 0.001$; Torso: $t_{601.25} = 6.84$, $p < 0.001$). These findings suggest that CLD children demonstrated greater versatility in their actions compared to CLD adults and two-handed peers (Fig. S2 for absolute time analysis).

The study was conducted immediately after the COVID-19 lockdown and overlapped with ensuing safety restrictions, during which some children wore masks and were generally hesitant to use their mouths to interact with objects. We therefore conducted a separate analysis for mouth use due to the unique circumstances during data collection. Out of the 65 participants who completed the behavioural tasks, only 10 (15.4%) used their mouths or chins in at least one of the 15 tasks. Notably, all of these participants were from the CLD groups, including five children, resulting in a significant group difference, based on a Chi-square test of independence ($X^2 = 12.64$, df = 3, $p = 0.005$). A two-sample bootstrap analysis was conducted to assess the difference in mean use percentage scores between CLD children and adults ($p = 0.027$), indicating that children used their mouths more on average compared to the adults. This finding highlights that even in such an unusual situation, CLD participants, and children in particular, still tended to use their mouths for object manipulation.

In summary, our results demonstrate that the compensatory repertoire of CLD involves multiple body parts. The compensatory behaviour of the CLD children is richer than found in adults. While our analysis does not allow us to ascertain whether this also reflects greater

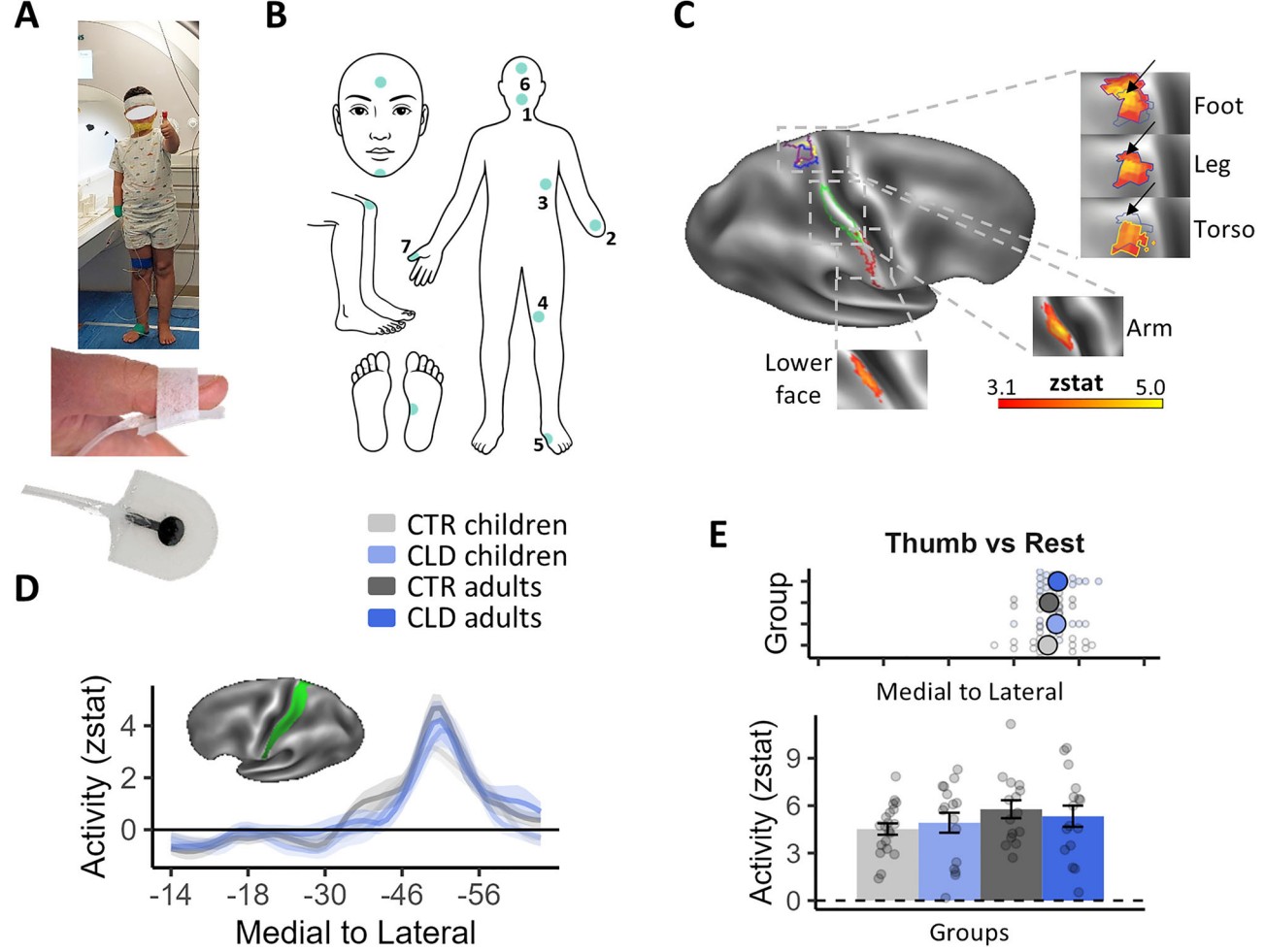

**Fig. 2 | Stimulated body parts in an fMRI experiment and validation analysis for the non-deprived cortex. A**, **B** Participants were fitted with custom-made soft pneumatic actuators (SPAs), which delivered vibrations to various body parts relevant for compensatory behaviours: the 1-lower face (middle), 2-residual arm, 3-torso, 4-leg and 5-foot (ipsilateral to the missing/non-dominant hand in CLD/CTR participants, respectively). Two additional control body parts were also included: the 6-thumb of the intact/dominant hand to account for potential age differences in overall activity; and the 7-upper face, a cortical neighbour of the hand not commonly utilised for compensatory behaviours. This setup allowed us to examine the specific contributions of adaptive behaviours to topographic differences. **C** Thresholded fMRI activity ($z$ values), averaged across all participants, for each of the body parts (versus baseline) along the S1 Brodmann Areas (BA)1–2 strip (**D** in green) of the hemisphere contralateral to the missing/non-dominant hand. On average, distinct activity for each stimulated body part in the contralateral hemisphere was observed, corresponding to the expected S1 topography. Coloured

outlines indicate the contours of clusters corresponding to each body part (see top-right inset plot in Fig. 4 for the classical homunculus): purple-foot; yellow-leg; blue-torso; light green-arm; red-lower face, with clusters of activity shown in the inflated panels. **D** Averaged activity ($z$ values) for the thumb condition is plotted for the four groups along the BA1-2 strip (in green). This corresponds to the hemisphere ipsilateral to the missing/non-dominant hand in CLD/CTR and contralateral to the stimulated thumb. $X$ values indicate MNI coordinates (medial to lateral). **E** Spatial position (top panel) and amplitude (bottom panel) of each participant's peak activity (and group means) along the S1 BA1-2 strip (**D**), showing no group differences in peak position or activity for the intact hand. This confirms comparable activity across groups, despite age differences. Error bars in (**E**) (bottom inset) and shaded areas in (**D**) indicate SEM. Coloured and greyed large dots in (**E**) (top inset) indicate mean position values. CLD children ($N = 15$), CLD adults ($N = 16$), CTR children ($N = 20$), CTR adults ($N = 15$). Source data are provided as a Source data file.

exploration or task switching, our finding indicates that adaptive behaviour at the tested age range (5–7 years) still has not matured. This leads to the pivotal question of whether such behavioural diversity is mirrored in the somatotopic organisation of the brain.

### Investigating activity along S1 for multiple body parts—validation

We aimed to compare body map characteristics in BA1-2 of the primary somatosensory cortex (S1; see also Fig. S3) across groups. Participants of all groups were fitted with custom-made soft pneumatic actuators (SPAs), designed to deliver vibration in the magnetic resonance imaging (MRI) environment across a range of body parts that are relevant for compensatory behaviours, namely: the chin (middle; hereafter, lower face), residual arm (or the wrist of the non-

dominant hand (CTR)), torso, leg and foot (ipsilateral to the missing hand (CLD)/non-dominant hand (CTR)) (Fig. 2A, B). As demonstrated in Fig. 2C, on average, the fMRI paradigm generated distinct activity for each of the stimulated body parts in the contralateral hemisphere, matching the expected topography of S1 (see also Fig. S4). We also stimulated two body parts as control conditions. First, the thumb of the intact/dominant hand, which we included to account for any potential general inter-group differences in activity levels. This includes any differences in the quality of the data, e.g. due to potential head movements, head size or level of arousal. Second, we stimulated the upper face—which serves as the cortical neighbour of the hand but was not commonly used for compensatory strategies. This allows us to consider the distinct contributions of adaptive behaviour.

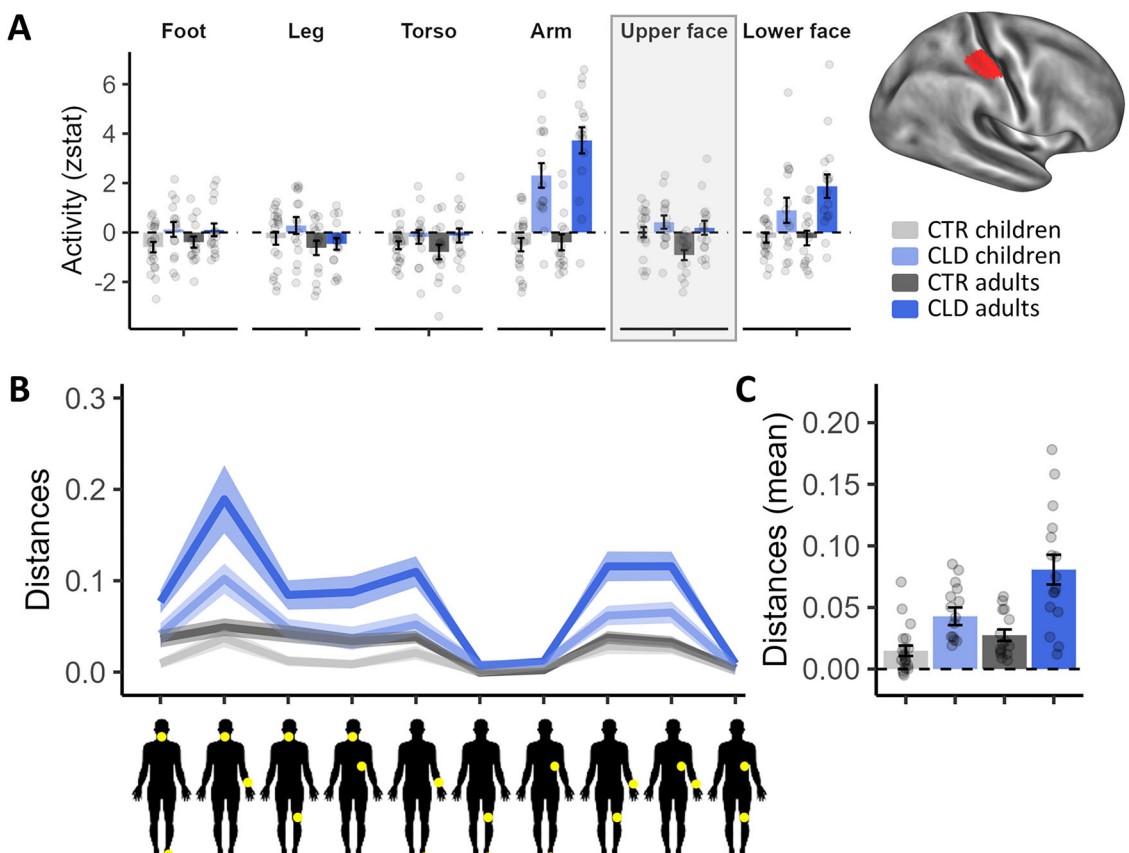

**Fig. 3 | Missing hand ROI: univariate activity and multivariate distances show deprivation effects are established in early life. A** Activity ($Z$ values) within the individual deprived/non-dominant hand ROI in CLD/CTR, respectively, averaged across participants for each group and body part (feet, legs, torso, arm and lower face). The top-right panel displays a participant's ROI after it has been resampled onto a standard template brain for visualisation purposes. Across body parts, activity was greater in CLD individuals, with remapping effects observed for the deprived-cortex neighbours (arm and lower face), as well as the topographically furthest body part—the feet. **B** Euclidean cross-validated Mahalanobis distance using a representational similarity analysis (RSA) for comparing information content between body parts within the S1-deprived hand area. The line plot shows the pairwise comparisons for each group. The yellow dots on the human body silhouettes indicate the compared body parts. **C** An average of the 10 distances shown in (**C**) demonstrates more representation of different body parts (increased distances) in CLD individuals. These results indicate that local topographic remapping in the deprived cortex may already be established in children. Other annotations are as in the figures above. Error bars in (**A**, **C**) and shaded areas in (**B**) indicate the standard error of the mean. CLD children ($N = 15$), CLD adults ($N = 16$), CTR children ($N = 20$), CTR adults ($N = 15$). Source data are provided as a Source data file.

We first compared the activity for the control thumb condition in the intact/dominant hand hemisphere across groups. Averaged activity ($z$ values) along the postcentral gyrus (medial−left to lateral−right; values on the $x$-axis indicate Montreal Neurological Institute, or MNI, coordinates) is plotted for the four groups in Fig. 2D. We focused on the spatial position and amplitude of each individual participants' peak activity along S1 (see Fig. S5 for single participant plots). We found no significant main effects nor interaction effects across our groups for peak activity level (all $p > 0.110$, $F < 2.64$). We found no significant main effects (all $p > 0.791$, $F < 0.01$) and a weak trend towards an interaction for the position of the peaks ($F_{1,62} = 2.96$, $p = 0.091$) that was not supported by Bayesian analysis ($BF_{10} = 0.14$, providing substantial evidence in favour of the null hypothesis) (Fig. 2E). We also found no significant pairwise group differences (all $p > 0.109$, $t < 1.63$, not corrected for 9 comparisons). This control analysis confirms previous findings in adults with CLD[23,29] and allowed us to ascertain that, despite age differences, the activity across groups is comparable. Therefore, any group differences observed in the deprived hand hemisphere are not due to methodological confounds in the quality of the fMRI recordings.

## Local remapping of body parts in the deprived hand area

The very localised deprivation in CLD individuals generates specific predictions of local remapping in the deprived hand cortex. To identify the missing hand area in individual participants, we mirrored the spatial position of peak activity for the intact/dominant thumb identified in its contralateral hemisphere in the deprived hand hemisphere. Considering the thumb comprises the lateral boundary of the hand map, we used the mirror-projection of the thumb peak as the reference point to define the missing hand region of interest (ROI). More specifically, for each participant, we selected one bin lateral and eight bins medial from the mirror-projection of the thumb peak for a total of 10 bins. An example of a functional ROI for an exemplar participant can be seen in Fig. 3A (brain plot). We then extracted activity levels within this individual hand ROI for all body parts (Fig. 3A). When comparing the averaged activity across the study's main body parts (i.e. the ones typically used for compensatory behaviour: feet, legs, torso, arm and lower face) within the deprived hand ROI, we found evidence for increased activity in CLD participants, as indicated by a main effect of limb difference ($F_{1, 62} = 34.26$, $p < 0.001$)—reflecting the classical remapping effect reported in CLD adults[5,6]. This effect did not interact with age ($F_{1,62} = 0.86$, $p = 0.357$); however, we did find a significant interaction between age and body part ($F_{4,248} = 3.73$, $p = 0.006$). A weakly trending 3-way interaction between age, limb difference and body part ($F_{4,248} = 2.01$, $p = 0.094$) was supported by Bayes analysis ($BF_{10} = 1.31e^{30}$), indicating that remapping across different body parts

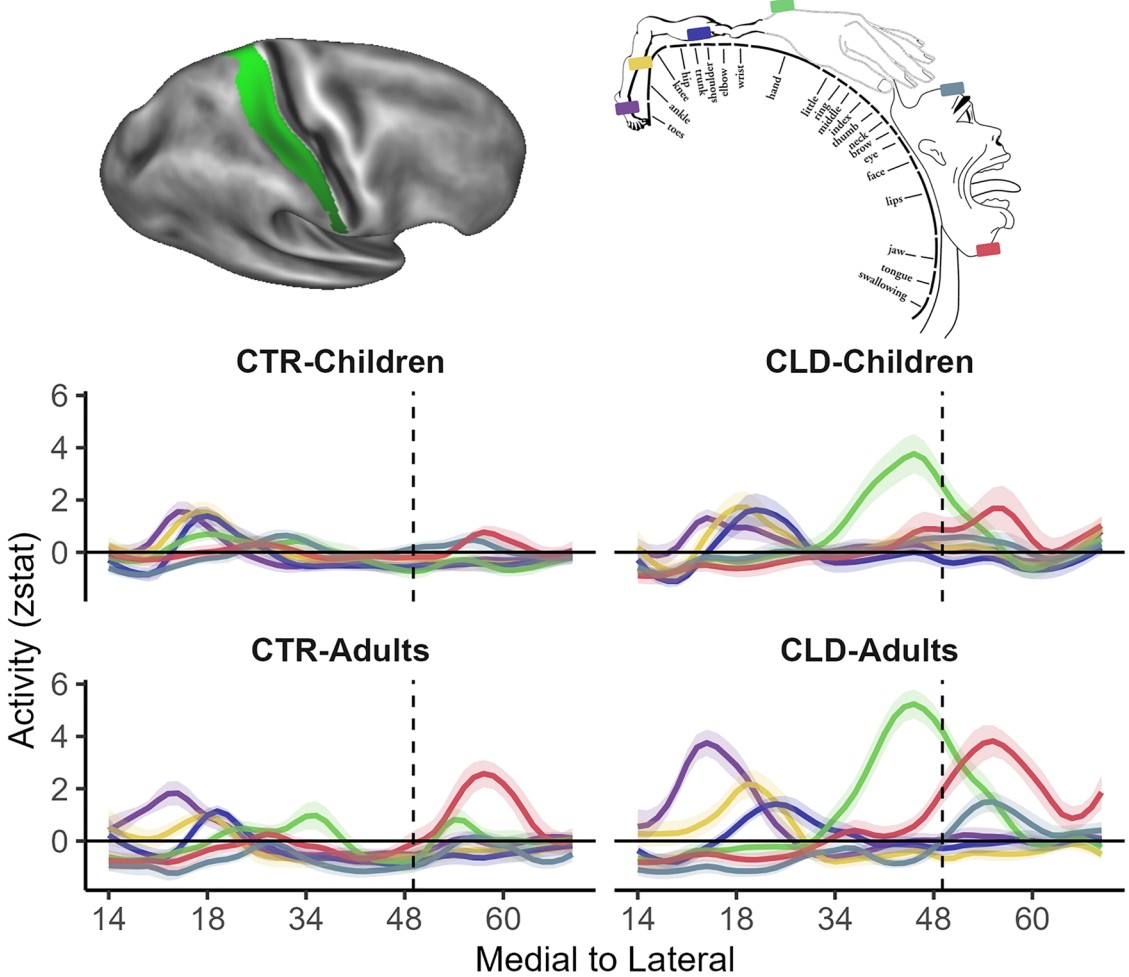

**Fig. 4 | Global topography: descriptive plots showing remapping across the entire S1 homunculus.** Line plots show the mean activity profile plots (*Z* values) for each body part and group (compared to baseline) along S1 in the hemisphere contralateral to the deprived/non-dominant hand (in CLD/CTR participants, respectively). The top left image shows the BA1-2 region of interest (ROI) projected on an inflated brain with shaded gradients of green. The top right image displays the layout of the classical topography in this region (taken from ref. 103) with the feet represented in the most medial regions and the lower face in the most lateral regions. The (missing) hand position is indicated by the vertical dashed line, based on the centre of gravity of the five digits from a meta-analysis[95]. Activity along S1 is varied between groups in peak position and amplitude. For the CLD groups, the arm (green line) and lower face (red line) were most clearly shifted towards the hand region, especially in adults. Similar differences can also be observed for the other body parts. The *x*-axis values indicate MNI coordinates (medial to lateral). Shaded areas indicate the standard error of the mean. Other annotations as indicated in the figures above.

might be differently affected by development (see Supplementary Table 6 for full results of the mixed models).

When looking at individual body parts separately, we find clear effects of remapping (expressed as a main effect of limb difference) by the cortical neighbours (arm: $F_{1,209.16} = 117.02$, $p < 0.001$; and lower face: $F_{1,209.16} = 25.20$, $p < 0.001$). Whole-brain analysis further confirmed the remapping effect for the arm and the other body parts (Fig. S6). We also find a trend towards a deprivation-driven main effect of remapping of the foot, which is topographically furthest from the hand, as supported by a Bayes analysis ($F_{1,203.74} = 3.59$, $p = 0.06$, $BF_{10} = 3.67$). Evidence of remapping for the legs and torso did not reach significance ($F < 2.47$, $p > 0.289$). We further find that the effect of limb difference, on arm remapping specifically, interacts with age ($F_{1,203.34} = 4.17$, $p = 0.043$). This supports the interpretation that, for the most part, local topographic remapping in the deprived cortex may already be established in children.

To complement this analysis, we next employed a representation similarity analysis (RSA) to quantify how distinct the information is for the different body parts with the S1 hand area (Fig. 3B). Here, we examined differences in activity patterns between each of the five studied body parts and found a clear main effect of limb difference ($F_{1,62} = 28.33$, $p < 0.001$), and a main effect of age ($F_{1,62} = 10.78$, $p = 0.002$). That is, the representation of the different body parts was more distinct in CLD individuals and in adults.

**Topographic shifts in CLD individuals are global across S1**
We next examined the global topography for the study's main body parts (feet, legs, torso, arm and lower face) along the full extent of S1 (BA1-2) in the deprived hand hemisphere (see Fig. S7 for homologous results in BA3b). Figure 4 shows the mean activity for each body part and group (contrasted with baseline). Qualitatively, perhaps the most striking results are the activity patterns of the residual arm (green) and lower face (red) in the CLD groups, which were already recorded in the deprived hand area analysis (Fig. 3). More broadly, activity seemed to vary between groups, both in terms of activity peak shift and amplitude.

We first characterise the spatial position of the activity peaks based on individual participants' datasets along S1, for each body part of interest. As apparent from Fig. 5, all body parts of interest show clear shifts in the CLD groups relative to controls, resulting in main effects of

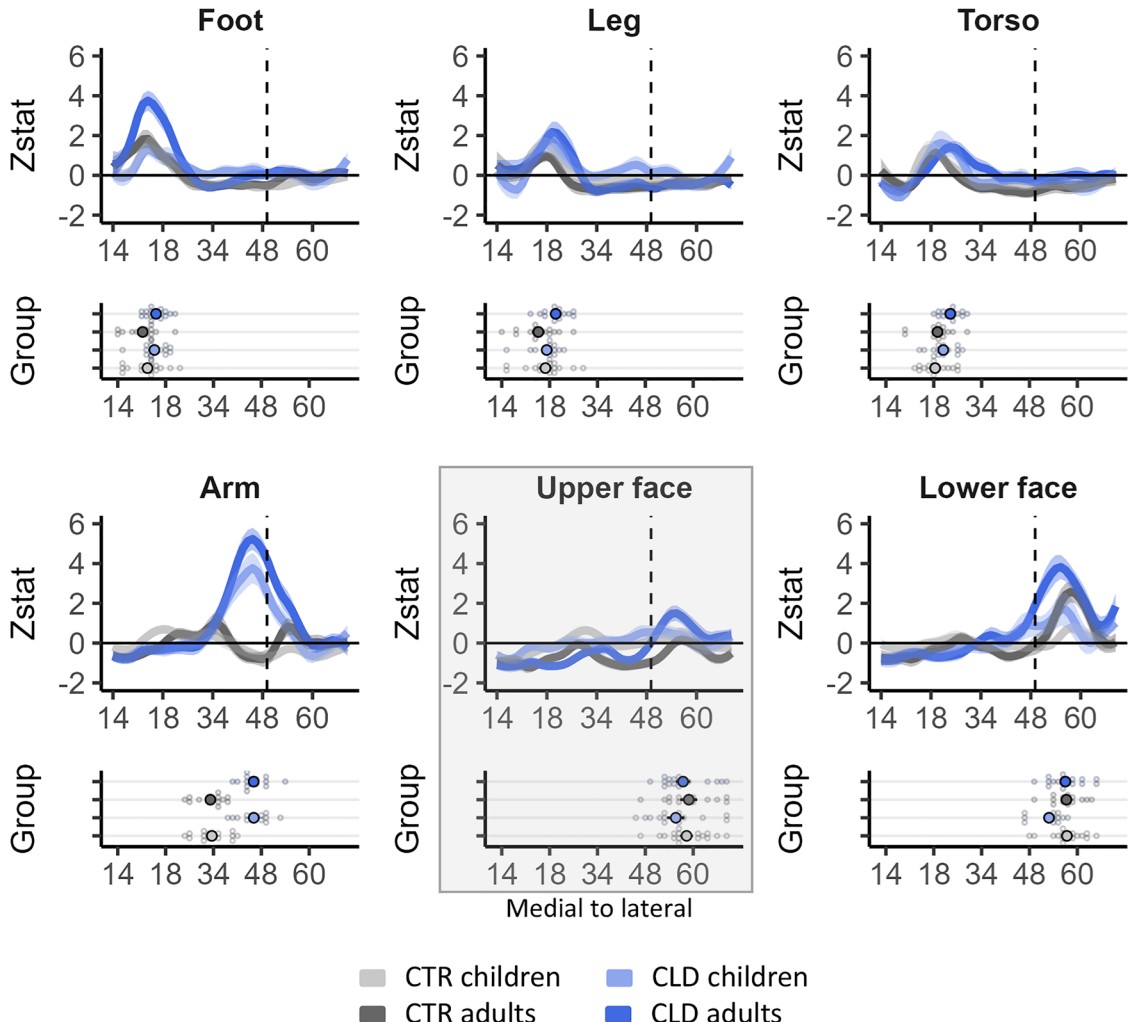

**Fig. 5 | Activity profile plots and peak position show consistent limb difference effects across body parts.** Group activity profile (top inset plots), extracted from the S1 BA1-2 contralateral to the missing/non-dominant hand (in CLD/CTR participants, respectively), shown in Fig. 4, for each of the tested body parts. Peak position for each body part and participant (bottom inset plots) was used for statistical comparison. All body parts showed significant shifts towards the missing hand position in the CLD groups compared to controls. This was found not only for the closer neighbours (residual arm's peak position shifted laterally, and the lower face shifted medially), but also for the lower body parts (feet, legs and torso), showing lateral shifts toward the missing hand area. This indicates that S1 remapping is pre- determined to a large degree before 5–7 years. Age-related interactions were revealed when considering aggregated effects across all primary body parts (omnibus analysis), indicating that factors in later development could further contribute to fine-tuning S1 topography. The upper face is shaded grey to indicate that it is a control body part and is not included in the main analysis unless otherwise specified. The x-axis values indicate MNI coordinates (medial to lateral). Shaded areas in (**B**) indicate the standard error of the mean. All other annotations are as indicated in the figures above. CLD children (*N* = 15), CLD adults (*N* = 16), CTR children (*N* = 20), CTR adults (*N* = 15). Source data are provided as a Source data file.

limb difference ($F_{1,58.50}$ = 30.78, *p* < 0.001). In the CLD groups, we see the residual arm's spatial position showing a *lateral* shift, whereas the lower face's spatial position shows a *medial* shift (relative to the CTR groups), both towards the (missing) hand area (arm: $F_{1,255.27}$ = 116.68, *p* < 0.001; lower face: $F_{1,255.27}$ = 5.61, *p* = 0.019; main effects of limb difference). We also see lateral peak shifts for the body parts which are further away from the deprived hand cortex towards the hand area— the feet ($F_{1,246.56}$ = 7.43, *p* = 0.007), legs ($F_{1,246.56}$ = 5.81, *p* = 0.017) and torso ($F_{1,246.56}$ = 7.84, *p* = 0.005; all main effects of limb difference). Crucially, the peak shifts did not compromise the overall S1 topo- graphy—the peaks were appropriately located in their relative topo- graphic positions, indicating that the local deprivation triggered extensive shifts which were found globally along the deprived S1.

With respect to developmental effects, we observed a main effect of age for the lower face only ($F_{1,255.54}$ = 4.30, *p* = 0.039), with no evi- dence that general development plays a role in the topographic dis- tribution of the other body parts along S1 (all *F* < 1.58, all *p* > 0.209 for

main effects of age). We found some indication for development- related remapping effects in the CLD groups, with the leg and lower face showing a significant interaction with age (leg: $F_{1,254.45}$ = 4.61, *p* = 0.033, lower face: $F_{1,255.54}$ = 4.68, *p* = 0.032). However, with the exception of the lower face ($t_{255.65}$ = 2.91, *p* = 0.004), none of the body parts showed significant differences between CLD children and adults. This finding indicates that the topographic shift may already be largely finalised in the children. That is, while CLD participants are different to controls in the topographic distribution of body parts, and CLD adults in particular, children and adults in each group mostly display con- sistent topography.

Interestingly, when adding the upper face in the model, which is the lateral neighbour of the deprived hand and is not widely used for compensatory behaviour, we found a weak trend towards a main effect of limb difference ($F_{1,317.97}$ = 2.99, *p* = 0.084), which remained ambig- uous following the Bayesian analysis (BF$_{10}$ = 0.52). While trend results should be interpreted with caution, this might indicate that the shifts

observed in children might not be exclusively shaped by behaviour in the earlier stages of development.

The omnibus analysis (across the 5 main body parts) found a strong main effect of limb difference ($F_{1, 58.50} = 30.78$, $p < 0.001$), but no main effects of age. The fact that limb difference, rather than age, strongly modulates spatial position suggests that S1 topography is predetermined to a large degree before the age of 5–7 years. Of interest, when combining across our primary body parts, we also find a strong trend towards an interaction between age and limb difference ($F_{1, 58.49} = 3.97$, $p = 0.051$, $BF_{10} = 1.69e^{168}$). This indicates that, although age effects are not robustly found for individual body parts, put together, there are also distinct developmental effects that determine global remapping in CLD individuals.

To complement this analysis of peak position, we also examined the *activity levels* at the peak of each individual participant's body part representations along S1. When comparing individual $z$ values for peak activity, the omnibus analysis reveals a significant interaction between age, limb difference and body part ($F_{4, 224.94} = 3.09$, $p = 0.02$), as well as main effects of limb difference ($F_{1, 58.88} = 20.91$, $p < 0.001$) and age ($F_{1, 58.81} = 4.88$, $p = 0.031$) (see Supplementary Results and Fig. S8 for full peak activity level analysis). In other words–overall, CLD adults show distinct effects to CLD children, which cannot be explained fully by the typical course of development, but this effect is not shared for all body parts.

Together, these results show that the representation of the entire homunculus is shifted in the CLD from an early age, and even in body parts that are represented further away from the deprived hand area. Nevertheless, some interactions with development for remapping in CLD individuals may hint at a potential modulatory role for behaviour.

## Homeostatic plasticity as a driver of global remapping across S1–a computational model

Our findings so far indicate that, while later-life determinants--such as compensatory behaviour--may contribute to brain topography, the most dramatic group differences are already determined in childhood. Given the unexpectedly broad effects of limb difference, we wanted to consider whether homeostatic plasticity is a plausible mechanism for topographic shifts spanning the entire somatosensory strip. To strengthen our conceptual understanding of the role of homeostatic plasticity, we implemented a biologically inspired computational model. Our model posits that when there is a change in the averaged input firing rate to a specific brain region (such as deprivation in the hand region), all weights of synaptic connections to that region are adjusted to maintain the overall/average local activity[14,30,31]. The primary goal of the model is to investigate how the global topography in S1 changes in CLD individuals. Rather than attempting to capture all biological complexity, the model operates at a high level and incorporates substantial simplifications and abstractions, in order to help us focus on core dynamics and underlying principles[32–34].

The model (shown in Fig. 6) simulates activity of the somatosensory homunculus in line with the typical topography of this region. Fifty internal nodes or neurons (Fig. 6A, top layer) represent the cortical regions that receive input from various body parts. For this implementation, we considered 10 main body parts: genitals, foot, leg, torso, arm, hand (including palm and fingers), face (including upper face and lower face) and throat (Fig. 6A, bottom layer). Each body part can be either at rest (stimulation value equal to zero) or stimulated (stimulation value equal to one) during a simulation. The activity of a neuron during stimulation of a single body part is determined by the connection weight between the stimulated body part and the corresponding internal neuron (as an example, see Fig. 6B where the torso is stimulated).

A crucial factor for synaptic scaling, which is driven by homeostatic plasticity is the average activity of neurons over time–referred to as *Mean Neural Activity* (MNA) in this model[13,14,31]. We calculate a

neuron's MNA as a sum of the products of the stimulation probabilities across body parts and their respective connection weights, as shown in Fig. 6C. The stimulation probability estimates how often a body part is typically used or stimulated. It is assumed that not all body parts receive the same amount of input during daily activity. In particular, hands typically experience more tactile stimulation than other body parts due to interaction with objects[35], as well as frequent self-touch of the other body parts, such as the face[36]. The connection weights reflect the node's body part selectivity. All cortical S1 regions receive input from all body parts' stimulations, but with different strength[37].

The essential feature of the model is the following: it assumes that the MNA value of each internal neuron over the long term (~hours-days) is preserved, regardless of outer conditions (e.g. complete loss of tactile stimuli from a body part). We called this value homeostatic MNA (or $MNA^{hom}$). This is what we consider homeostatic plasticity, which can be achieved through a synaptic scaling mechanism[14,30,31]. If there is a deviation from this homeostatic value, the connection weights are updated (i.e. multiplied by a factor) to restore the target value.

Our model allowed us to simulate the somatosensory homunculus activity when stimulating a specific body part, as well as deprivation effects by excluding or reducing inputs from a specific body part. More specifically, we considered the reduced input from the hand into the homuncular strip for CLD individuals. It is important to note that being born with a missing hand due to an upper limb developmental malformation does not completely eliminate primary input to the hand area, as some peripheral nerves that are genetically hard-wired to send information to the central nervous system remain intact. That is, nerve endings intended to innervate the hand densely populate the end of the residual limb, a phenomenon akin to peripheral reorganisation[38]. Consequently, when the residual arm is stimulated, either through use or passive stimulation, nerves associated with the arm are activated, in addition to residual nerves of the missing hand, although to a lesser extent. To account for peripheral reorganisation, the initial weights of the model from the hand in CLD individuals were only reduced and not set to zero, as can be seen in Fig. S9. To clarify that these represent residual nerves, in our model, we use different labels (i.e. Palm_nerves and Finger_nerves in Fig. S9 and onwards).

The difference in initial connectivity weights between the two groups leads to distinct MNA values for their corresponding neurons. This, in turn, results in different trajectories of homeostatic adjustment in each group. We simulated this mechanism with our model (see formulas in the Methods section). The model results are presented in Fig. 6D, which shows the activity profiles across the 50 nodes when a specific body part is stimulated (top plots) and the related shift of the position of the activity peaks (bottom plots). To investigate whether the synaptic scaling homeostatic mechanism can explain the observed activity patterns in CLD and control groups, we qualitatively compared the model activities with the activity patterns observed in humans (both CLD and CTR). More specifically, we compared the model's properties with four key empirical observations, as presented in Fig. 5 (see Methods for a more detailed explanation): (1) The activity peak during arm stimulation is higher and shifted laterally in CLD individuals compared to CTRs; (2) The activity peaks for the foot, leg, torso, upper face and lower face are closer to the hand area in CLD individuals than in CTRs; (3) The activity peaks are higher in CLD than in CTR individual; (4) Both groups show a decrease in activity peak magnitude from the foot to the leg and torso, and an increase from the upper face to the lower face. The model, which incorporates homeostatic adaptation, peripheral reorganisation, higher hand stimulation probability and stronger initial connection weights from hand to the internal nodes in CTRs, successfully replicates all four empirical findings (for descriptive comparisons, see Fig. 6D). That is, our key empirical observations can be explained using a key homeostatic plasticity mechanism–synaptic scaling.

To further confirm the role of homeostatic plasticity, we compared the results to a simplified, more rudimentary model based solely

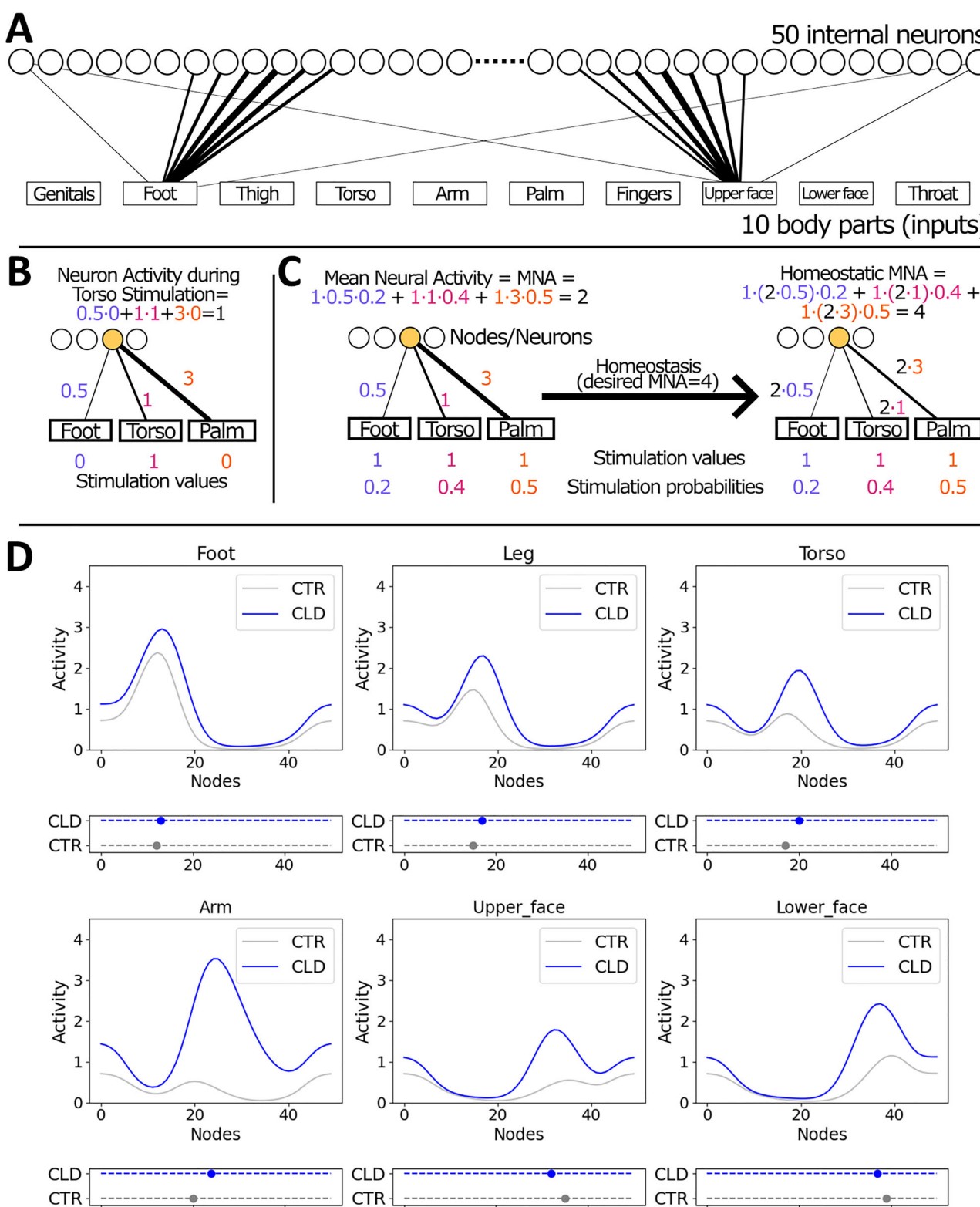

on the homeostatic mechanism, i.e. excluding peripheral reorganisation and assumptions of input inhomogeneity for the upper limbs. Despite its simplicity, the model largely aligns with the key findings related to differences between the two groups (observations 1–3). However, this rudimentary model did not replicate our 4th observation for the CLD group. This shows that our model is robust to variations in parameter choice (see also Supplementary Results). Notably, with our main model, we were able to replicate all four observations

without considering behavioural differences between the groups and further attempts to consider behavioural differences did not improve compatibility with the observed findings (see Supplementary Results for model variants with different parameter settings).

### The impact of behaviour on topographic shifts

While most of the topographic differences observed in the CLD groups appear to have been pre-determined prior to childhood (as evidenced

**Fig. 6 | A biologically inspired computational model demonstrates the explanatory power of homeostatic plasticity in shaping global remapping due to a congenital limb difference. A** The model, designed as a shallow neural network with 50 nodes representing the S1 homunculus, simulates how the brain adjusts synaptic connections to maintain overall activity despite changes in input firing rates, such as reduced input from an under-developed hand. Strength of connections from the input layer to the internal layer is schematically represented by the line width. For example, when the foot is stimulated, the 10th neuron from the left is activated more than the rest of the neurons, with decreasing graded strength. The model accounts for peripheral reorganisation in CLD individuals, where nerves intended for the hand innervate the residual limb, allowing partial arm input into the deprived cortical hand territory. **B** Illustration of a neuron activity calculation during stimulation of the torso. **C** Illustration of homeostatic plasticity through synaptic scaling. Initially, the Mean Neural Activity (MNA) of the selected (yellow) internal node/neuron is equal to 2. Assuming that the desired value of MNA is equal to 4, all connections of the neuron are multiplied by 2. Stimulation probabilities estimate how often individual body parts are stimulated. Stimulation values are equal to 1 for all body parts to simulate our study's SPA stimulation. **D** The model replicates several key empirical differences observed in CLD individuals relative to controls: higher activity peaks that are shifted towards the deprived cortex, not just for the residual arm but also more globally across S1, and patterns of peak magnitude changes across different body parts. These results suggest that homeostatic synaptic scaling mechanisms can account for some of the most striking effects observed in individuals with CLD, and adults in particular. The dots in the plots below the line plots indicate peak positions for each group. Empirical findings are presented in Fig. 5.

in the very strong main effects of limb difference across all analyses), we also identified evidence for divergent developmental trajectories in the CLD groups. In particular, in the S1 peak shift analysis, when combining across our primary body parts, we reported an interaction between age and limb difference ($F_{1, 58.49} = 3.97$, $p = 0.051$, $BF_{10} = 1.69e^{168}$). To directly examine whether the global S1 remapping reported above relates to behaviour, we analysed deviations in activity profiles and behavioural performance for each CLD individual relative to their peer control group. If behaviour links to brain remapping, then participants who used a specific body part more than their peers would also show larger peak shifts compared to their peers (Fig. 7A). In other words, larger deviations from the average peer group in behavioural usage would predict larger deviations in activity peak shifts relative to the average peer group shift.

We analysed the two CLD groups ($n = 15$ children and $n = 15$ adults) separately using permutation tests with 5000 iterations. The tests aimed to determine: (1) if the Spearman correlations between the behavioural and peak shift deviations were significantly greater than zero for both groups, and (2) if the children's group had a larger correlation than the adult group. Both tests yielded significant results (Fig. 7B–D). Specifically, the averaged Spearman correlation was 0.59 ($p < 0.001$) and 0.39 ($p = 0.003$) for the children and adults, respectively. Furthermore, the children's mean correlation was significantly larger than the adults' ($t = 2.25$, $p = 0.016$). These results were corroborated when using frequency as a measure to estimate compensatory behaviour (see the Supplementary Section 'Supportive analysis for the impact of behaviour on topographic shift'). These findings demonstrate that the unique behavioural adaptations that are favoured by individuals are also reflected in variations in S1 brain topography, particularly during early development.

## Discussion

The developmental trajectory of motor and sensorimotor skills is gradual and experience-dependent[39], and continues to evolve well beyond early childhood, extending into adolescence[40,41]. During this time, significant improvements occur in motor skills, coordination and complex motor learning[42], in which S1 plays a key role[43–46]. This prolonged developmental period allows us to consider how brain and behavioural adaptations interact in individuals with CLD.

Consistent with prior research[5,6], some of our strongest effects were observed for the cortical neighbourhood of the hand—namely for the residual arm and the lower face. These emphasise the key role of deprivation in driving remapping in CLD. Here, we also found that the effects of limb difference extend the spatial layout across the entire somatosensory homunculus, including cortically distant body parts such as the feet, legs and torso. While changes to local representation of these body parts could have been anticipated by the extensive compensatory behaviours demonstrated in our study, interestingly—across the body map—we see each of the body parts shifting *towards* the deprived hand area. This indicates a clear mechanistic role for sensory deprivation. Moreover, since these shifts

are also evident in the CLD children, our data allows us to establish that these primary sensory maps are configured early in life, with the resulting spatial organisation persisting into adulthood. This stability suggests that topographic organisation is mostly resilient to subsequent sensory experiences, underscoring a pre-established functional architecture that might be more resistant to change than previously understood[11]. As further support to this view, we find that some of the limb differences across S1 may be shared to a small extent (as expressed in trends towards significance) with the upper face, which was specifically included in our experimental paradigm as a control body part that is not relevant for compensatory behaviour. The early establishment of these dramatic remappings suggests a significant role for homeostatic plasticity processes: the deprived cortex may increase its responsiveness to surrounding or residual inputs, irrespective of behavioural relevance.

The primary function of homeostatic plasticity is to provide stabilising mechanisms that regulate changes in the system, preventing activity levels from becoming excessively high or low[14]. This can be achieved by network-level computational mechanisms such as divisive normalisation[47], where the neuron's response is modulated by the combined activity of a relevant population[48,49]. Electrophysiological research suggests that local remapping is primarily driven by the unmasking of silent inputs through disinhibition of homeostatic plasticity mechanisms[13]. Through our computational model, in the current study, we were able to provide a proof of concept that a simple plasticity rule predicts a much more global remapping pattern, causing topographic shifts across the entire homunculus. Moreover, this mechanism provides a powerful explanation to understand why CLD adults, who show *less* compensatory behaviour than children, show *increased* activity for these body parts relative to their peer groups, as demonstrated in the sporadic interactions across the results. This is because any reduction in the input from these body parts in later life, due to reduced diversity in compensatory strategies, will result in an upregulation of these synaptic inputs[15]. When placed in our experimental context, where stimulation in the fMRI study was matched across age groups, this will result in an increased S1 activity in CLD adults, above and beyond what could be explained by the main effects of age. Our rudimentary computational model did not include a Hebbian plasticity mechanism, and as such, it was not equipped to capture the likely more nuanced relationship between use- and deprivation-related plasticity. Together, our study highlights the crucial role of homeostatic plasticity in shaping the developing homunculus, through both deprivation and compensatory behaviours.

Nevertheless, we still find multiple pieces of evidence for topographic motifs being modulated across development. Although interaction effects with age were weaker, they provide conclusive evidence that the somatotopic maturation of CLD individuals follows a distinct trajectory from that observed in neurotypical development, opening up a potential role for their unique behaviours in refining the fine features of S1 topography. Indeed, homeostatic plasticity

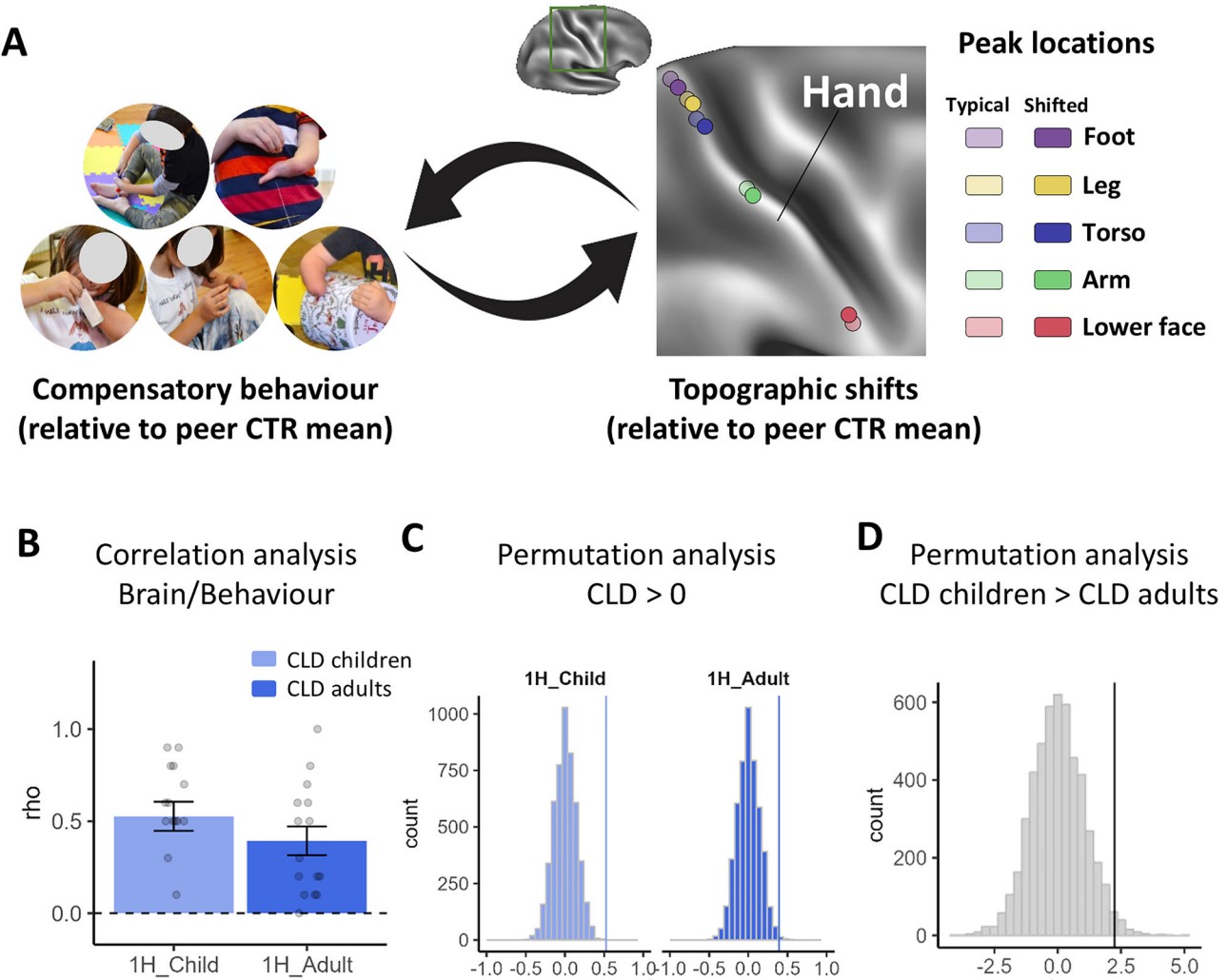

**Fig. 7 | Correlation analyses between compensatory behaviour and topographic shifts. A** For each CLD participant and each of the five body parts, compensatory behaviour was quantified as the differential percentage of time spent using alternative body parts, compared to the peer group norm. The corresponding brain change was estimated as the topographical shift in MNI coordinates, relative to the peer group norm. This approach yielded five paired values (compensatory behaviour and brain shift) per CLD participant. **B** These paired values were then used to compute a Spearman correlation coefficient for each individual, providing a participant-specific measure of the relationship between behavioural adaptations and changes in brain topography. To explore developmental differences, we calculated the mean Spearman correlation for each group (children and adults) separately. **C** To determine whether the group-averaged correlations were significantly greater than zero, we conducted a permutation analysis (5000 iterations), randomly shuffling the pairing between behavioural and brain data. The plot presents the null distributions for both groups, with the observed mean correlation values indicated by vertical lines. The test was positive in both groups, suggesting a meaningful relationship between compensatory behaviour and brain topographical changes. **D** Additionally, a second permutation analysis (5000 iterations) showed that the correlation in children was significantly stronger than in adults. These findings suggest that compensatory behavioural adaptations in CLD individuals are closely linked to changes in S1 brain topography, with this relationship being particularly robust during early development. Error bars in (**B**) indicate the standard error of the mean. CLD children ($N=15$), CLD adults ($N=15$). Source data are provided as a Source data file.

mechanisms are not isolated from the dynamic tuning of receptive fields by Hebbian learning[50]. We chose to study children aged 5–7 years, as they are developing fine motor skills, featuring a critical phase in motor control maturation[51]. This is also the age when children spend increasing time at school desks. This might be constraining for CLD children, considering their natural tendency for over-reliance on the lower limbs and feet, which we demonstrate here. Alternatively, it is tempting to speculate that the neural circuitry that enables the maturation of neurotypical hand control might provide the upper limbs with a natural dexterity advantage over the lower body. Regardless, between childhood and adulthood, individuals with CLD will abandon some of these unique compensatory behaviours and instead will learn to better rely on their upper limbs. This is reflected in the interaction effect we find in our body-part co-use index, indicating

a shift from a diverse compensatory repertoire to a more economical pattern. The changed compensatory repertoire across body parts will directly impact the natural statistics of the somatosensory activity pattern. The fact that we can capture these dynamic behavioural shifts between our children and adult CLD testing groups provides us with the unique opportunity to explore the role of adaptive behaviour on S1 organisation. Indeed, we find correlational clues for links between brain and behaviour—the body-part peak shifts of individuals with CLD along S1 reflect their distinct behavioural repertoire, with children showing greater correlations between brain and behaviour than adults. Together, our findings suggest that, while the cortical functional architecture of sensory cortex is established early, fine-tuning in response to CLDs and behavioural adaptations continues through late childhood or beyond.

Our study was motivated by related seminal research of remapping in the deprived sensory cortex of congenital blindness or deafness, where the roles of compensatory behaviour have also been considered as a driver of crossmodal remapping[1,52–54]. Such studies led to the strong notion of brain reorganisation—a qualitative switch in the functional identity or computational capacity of a given brain area—as a mechanism for driving physiological and behavioural change after various forms of insult to the nervous system. In a recent critical review, we did not find consistent strong evidence for the existence of functional pluripotency for the sensory cortex in any period of development[11]; instead, remapping effects in these deprived cortical areas reflect upregulation of pre-existing inputs, and as such, we suggested that the assignment of brain function to a given cortical structure is likely to be largely fixed at birth. The impressive compensatory behavioural abilities seen in these clinical cases, such as echolocation-based navigation or Braille reading, may be attributable to learning-related recruitment of higher-level cortical computations that can be functionally re-focused on residual primary area representations. Our current study confirms this perspective and adds to it by showing a lack of clear downstream readout of the S1 remapping effects by S2 (reported in Supplementary Materials and Fig. S14). Together, our findings demonstrate that deprivation effects can 'passively' contribute to extensive topographic remapping, above and beyond previous accounts.

Could these far more dramatic effects of deprivation- versus use-dependent plasticity observed in the current study be a mere consequence of our focus on sensory versus motor cortex? Indeed, homeostatic plasticity is a hallmark of S1 and other primary sensory cortices[55–57], whereas M1 is more dominantly associated with Hebbian and structural forms of plasticity[58,59]. Our focus on S1 was primarily derived from methodological considerations for the fMRI experiment. To ensure consistency across participants and groups, and minimise head movements, it was highly impractical to rely on the children to perform instructed movements. We were encouraged by previous research, both by ourselves and others, showing minimal differences in S1 organisation and remapping between active and passive somatotopy protocols[9,10,60]. Somatotopic maps derived from finger tapping (motor-driven fMRI) closely correspond with those evoked by direct microstimulation of the same S1 sites in tetraplegic patients undergoing intracortical implantation[61]. These studies provide direct validation that fMRI responses to motor tasks engage the same topographic architecture as tactile stimulation. We were also motivated by the tight functional similarity between primary sensory and motor cortex in controls[62] and individuals with CLD[5,6,10,23,29]. Indeed, given the key roles of S1 in motor planning[46], learning[63] and execution[64], as well as predictive coding frameworks[65], it is difficult to conceive a unidirectional functional architectural change in one node but not the other. In humans, tactile learning generalises in a use-dependent manner[66], and motor learning alters sensory feedback gains in S1[63], supporting the idea that functionally meaningful plasticity often emerges from co-dependent sensory-motor processes. In rodents, tactile information, including object texture and location, is computed in S1 through integration of motor and sensory signals[67]. Therefore, while mechanistic differences may exist at the microcircuit level, experience-dependent plasticity likely reflects across multiple levels of the sensorimotor hierarchy[68]. Here, it is also important to highlight that our local remapping findings are a replication of previous studies using active movements across both S1 and M1[5,6]; and that our present results are derived from the higher sub-divisions of S1 (BA1 and 2), which may be less closely associated with thalamic deprivation, relative to BA3A and 3B (see Supplementary Fig. S7). Taken together, these considerations suggest that the striking deprivation-driven plasticity we observe in S1 is unlikely to be an artefact of our sensory focus, but rather reflects deeply integrated sensorimotor processes that shape cortical organisation from early

development. Nevertheless, we cannot rule out the possibility that use-dependent plasticity plays a more prominent role in M1.

Our perspective, that S1 topography is mostly resistant to change, is also compatible with extensive research showing stable cortical body maps before and after arm amputation in adults[69]. Yet, it is important to also consider the potential role of developmental sensitive periods, which are thought to occur before plasticity brakes, which normally preserve homeostatic balance, such as inhibitory circuits and synaptic pruning, have fully matured[19]. This raises the obvious question—would we have been able to observe greater developmental effects if we had the means to scan even younger children? Given there is so little topographic mapping research in children with related sensory deprivation conditions[70–73], this is an open question. Previous research in children focused on sensory evoked potentials and fields, using EEG[28] or MEG[74,75], which lack the spatial specificity that we are focused on in the current study. Upper extremity development starts at ~4 weeks of gestation, with isolated arm movements observed as early as 8 weeks[76,77]. By full-term, the typical sensorimotor homunculus is already crudely organised[78]. Given the relatively rich somatosensory environment the uterus presents, it is likely that the main effects of deprivation we have observed are perinatal. Nevertheless, the impact of early motor development and the divergent developmental trajectories pursued by infants with CLD remain an open question for further research.

We aimed to capture how CLDs influence the evolution of the homunculus. We find that, while peripheral reorganisation, sensitive periods and use-dependent plasticity all play roles in shaping neurodevelopment, it is homeostatic plasticity mechanisms that predominantly drive the comprehensive topographical remapping observed in S1. This remapping, established early, remains robust into adulthood, indicating a largely pre-determined cortical architecture that is subtly refined by early life experiences, undergoing few alterations in later life.

The implications of these findings extend into clinical practice, particularly highlighting the limitations of behavioural interventions in shaping brain (re)mapping, particularly with respect to restoration of sensory input in populations with congenital or early life deprivation. Instead, our results highlight the importance of utilising and strengthening pre-existing neural pathways. Technologically, the implications of our findings advocate for the early adoption of neurotechnologies, suggesting integration during the earliest developmental stages could leverage sensitive periods of development to enhance functional outcomes.

## Methods
### Participants
We tested 16 CLD children (mean age: 6.69 ± 0.60; 7 left-handed; 10 female; see Table 1 for details) and 16 CLD adults (mean age: 41.19 ± 11.39; 5 left-handers; 12 female; see Table 2 for details) with an isolated congenital upper limb malformation—of either the entire limb or the hand-plate. CLD individuals were included if they have only one hand which can perform a pincer grip (i.e. the residual arm has either no digits or digits which do not provide a useful grip). CLD individuals with a single malformed but functional hand are excluded as 'too functional'; individuals with no functional hands are also excluded. One CLD child was excluded from the study due to having three residual fingers and a functional grip in the malformed limb. Control participants were recruited to match the other two groups in terms of age, gender and handedness (with respect to the intact hand). For the control group, we tested 21 children (mean age: 6.71 ± 1.10; 4 left-handed; 13 female) and 16 adults (mean age: 39.56 ± 9.85; 5 left-handers; 7 female). Not all participants (4 control children and 1 CLD adult) participated in the behavioural session, therefore the behavioural analyses were conducted on the following sample: 16 CLD children (mean age: 6.69 ± 0.60; 7 left-handed; 10 female); 15 CLD adults (mean

## Table 1 | Demographic details of children with congenital limb differences included in the study

| ID | Missing limb | Level | Prosthesis | Prosthesis use | Age range |
|---|---|---|---|---|---|
| 1 | Right | 2 | Yes | 5 | 5–7 |
| 2 | Right | 4, 0 digits | No | - | 5–7 |
| 3 | Left | 4, 0 digits | No | - | 5–7 |
| 4 | Left | 2 | No | - | 5–7 |
| 5 | Right | 4, 0 digits | Yes | 4 | 5–7 |
| 6 | Left | 4, 0 digits | Yes | 5 | 5–7 |
| 7 | Left | 2 | Yes | 5 | 5–7 |
| 8 | Left | 4, 1 digit | No | - | 5–7 |
| 9 | Left | 2 | Yes | 4 (cycling) | 5–7 |
| 10 | Left | 4, 0 digits | Yes | 3 (sport) | 5–7 |
| 11 | Right | 1 | Yes | 4 | 5–7 |
| 12 | Right | 1 | Yes | 5 | 5–7 |
| 13 | Right | 1 | Yes | 3 (cycling) | 5–7 |
| 14 | Left | 2 | Yes | 5 | 5–7 |
| 15 | Right | 1 | Yes | 5 | 5–7 |
| 16 | Left | 2 | | | 5–7 |

Missing limb refers to the side; Level refers to the level of the developmental arrest, as follows: 1 = above elbow, 2 = less than ½ forearm, 3 = more than ½ forearm, 4 = wrist or partial hand; prosthesis use refers to the typical frequency of prosthetic arm use, as follows: 1 = everyday, 2 = 5 days a week, 3 = weekly – less than 5 days a week, 4 = monthly, 5 = never/rarely, task-specific prosthetic use is noted where applicable.

## Table 2 | Demographic details of adults with congenital limb differences included in the study

| ID | Missing limb | Level | Prosthesis | Prosthesis use | Age range |
|---|---|---|---|---|---|
| 1 | Left | | | | 25–61 |
| 2 | Right | 4, 0 fingers | Yes | 1 | 25–61 |
| 3 | Left | 4, 0 fingers | No | 3 | 25–61 |
| 4 | Left | 4, 1 finger | Yes | 1 | 25–61 |
| 5 | Left | 4 | Yes | | 25–61 |
| 6 | Left | 4, 4 fingers | No | - | 25–61 |
| 7 | Left | 4, 0 fingers | No | - | 25–61 |
| 8 | Right | 2 | No | 5 | 25–61 |
| 9 | Left | 4, 0 fingers | No | 5 | 25–61 |
| 10 | Right | 4, 0 fingers | Yes | 3 | 25–61 |
| 11 | Right | 2 | Yes | | 25–61 |
| 12 | Left | 4, 0 fingers | No | 4 | 25–61 |
| 13 | Left | 4, 0 fingers | Yes | 3 | 25–61 |
| 14 | Left | 4, 0 fingers | Yes | 4 | 25–61 |
| 15 | Left | 4, 0 fingers | Yes | 3 | 25–61 |
| 16 | Right | 4, 0 fingers | No | - | 25–61 |

All annotations are as in Table 1.

age: 42.27 ± 10.91; 5 left-handers; 12 female); 17 control children (mean age: 6.59 ± 1.18; 4 left-handed; 10 female) and 16 control adults (mean age: 39.56 ± 9.85; 5 left-handers; 7 female). Not all participants (1 control child, 1 CLD child and 1 control adult) participated in the fMRI session, either because they decided to stop the experiment or for technical issues with the SPAs, therefore the fMRI analyses were conducted on the following sample: 15 CLD children (mean age:

6.73 ± 0.59; 7 left-handed; 9 female); 16 CLD adults (mean age: 41.19 ± 11.39; 5 left-handers; 12 female); 20 control children (mean age: 6.70 ± 1.13; 4 left-handed; 13 female) and 15 control adults (mean age: 39.67 ± 10.18; 4 left-handers; 7 female). Ethics approval for this study was granted by the University College of London research ethics committees (17205/001). Informed consent and consent to publish were obtained in accordance with ethical standards set out by the Declaration of Helsinki. All participants met local MRI safety guidelines and provided written consent. For child participants, a parent provided written consent, and the child gave verbal assent. Adult participants were compensated for their participation and had the option of donating all or part of their compensation to a charity (either REACH or LimBO). Child participants received an Amazon voucher, and a fixed amount was also donated to a charity.

To improve access to CLD participants, the study was conducted from two locations: the Birkbeck UCL Centre for Neurological Imaging (BUCNI) in London and the MRC Cognition and Brain Sciences Unit (CBU) in Cambridge. 21 CLD participants (10 children and 11 adults) were tested at BUCNI, while 12 CLD participants (7 children and 5 adults) were tested at CBU. Additionally, nine control children were tested at BUCNI, and 12 control children were tested at CBU. All control adults ($n = 16$) were tested at CBU.

### Behavioural session

**Procedure and task.** To evaluate how CLD participants compensate for their missing limb during daily activities, a standardised object manipulation task ('Surprise Suitcase') designed to elicit bimanual coordination was employed. This task involved a series of everyday objects placed inside a large, lightweight suitcase locked with a keyed padlock (see complete task and object list in Supplementary Table 1). This task was performed by all four participant groups (including adults). Participants were asked to open or manipulate these objects (e.g. open a plastic salad container, unwrap candy) without any guidelines or restrictions regarding the body parts to employ. The task was conducted in a separate room prior to the scanning session. Participants were seated on a soft play mat, and the session was video recorded for later analysis. The experimenter sat directly in front of the participant, providing instructions for task completion and encouragement. For children, a parent typically remained in the room for comfort and support. The task began with the participant seated on the mat with the closed suitcase in front of them. The first instruction involved using the key to unlock the padlock. Following this, the experimenter presented each object in a pre-determined order, instructing the participant on how to manipulate it. For instance, an instruction might sound like: 'Could you please open the salad box?'. Generally, the execution time varied by age group. For children, completing the task typically took 10–15 min (including all conditions), whereas adults generally finished within 5 min. While CLD participants were given the option to wear a prosthetic to complete the task, none of the children did. One CLD adult preferred to complete the task with the prostheses, and five CLD adults did the task with and without the prostheses. Analyses were conducted with their preferred and more comfortable method.

### Compensatory behaviour analysis

**Video coding.** Compensatory motor behaviour was coded offline from video recordings of the behavioural tasks. Behaviour was characterised as the active use of the following body parts: intact or dominant limb, residual or non-dominant limb, torso, lower face, legs and feet. Behavioural coding was completed by four independent raters. To establish the reliability of the coding, the ratings of the three additional raters were compared against those of the main rater for 20% of videos. Raters demonstrated almost perfect agreement on body part used (Cohen's kappa = 0.972) and excellent reliability for body part usage duration (ICC = 0.989). As agreement rates were almost indistinguishable, coding was left as per the secondary raters' rating.

**Behaviour indices**. To assess everyday compensatory behaviours in CLD and control participants, two primary measures were calculated for each sub-task: (1) Co-use: The co-occurrence of body part usage during tasks, and (2) Time Usage: The percentage of time spent utilising each body part relative to the total task duration. More specifically, to calculate the percentage of use metric, in each sub-task, we summed the durations of all bouts using a particular body part and expressed that as a percentage of the duration of the sub-task time. These percentages were then averaged across sub-tasks. Furthermore, we derived two outcome measures from the co-use measure. The first co-use measure is an estimated index that reflects both the number of body parts used and how consistently they were used, with higher values indicating more usage or more consistent usage (see Supplementary Methods for details on how this index was calculated). The second co-use measure allowed us to compare compensatory co-use with brain measures. A co-use matrix ($5 \times 5$) was generated for each participant. This matrix captured the frequency of co-occurrence for each pair of body parts across all tasks within the session. The value at each cell in the co-use matrix represented the number of sub-tasks where those two body parts were co-used during the entire session.

## fMRI session

**Soft pneumatic actuator (SPA) skin stimulator and the controller setup.** Participants were stimulated during the scanning session with a modified version of the SPA-skin interface[79–81]. Owing to its soft and compliant materials, this technology was previously demonstrated to provide reproducible, localised and high-fidelity tactile stimulation over various parts of the human body, with stiffness ranging from 0.5 to 2 megapascal (MPa).

The complete apparatus comprised an actuator, an electronic controller and a pneumatic supply system. The silicon-fabricated actuator (1 mm thickness) consists of two layers of thin silicone sandwiching a masking tape layer (Black tape Fig. 2A) that determines the shape of the active area of actuation. Upon inflation through pneumatic air input of up to 100 kPa, the actuator is able to provide a force of 3 N with 10 mm² active area[79,80]. In our study, we used a round-shaped actuator with a 3.75 mm radius and an active area of 4.4 mm² for all body parts, except for the thumb, where we applied a smaller round-shaped actuator with a 3.2 mm radius and a 3.25 mm² active area to ensure a better fit. By controlling the rate at which the air is sent in and extracted out, a sense of vibratory feeling is generated through the electronic controller unit, which is the second component of the apparatus. The electronic controller is connected to a computer via USB cable, receiving instructions at 50 times per second to activate the respective valve at the respective actuation amplitude and frequency. An air compressor in the control room of the fMRI scanner was used to supply input pressure to a pneumatic pressure regulator (ITV1050, SMC Corp., USA) holding stable output pressure up to 2 bars or 200 kPa. The complete system was designed in EPFL, Switzerland[79,80], and a homologous system was created for the CBU testing site. The system components have been extensively tested for artefacts they may introduce in the fMRI environment[81].

**Scanning procedure and fMRI task.** After the behavioural session, participants took part in the MRI session. To familiarise the children with the scanner setting before starting the experiment, they were taken with their parents to see either the actual scanner (at BUCNI) or a mock scanner (at CBU). Children were invited to enter the scanner to ensure they were not scared of the machine and helped the researcher place a puppet on the scanner bed. This procedure aimed to make the children comfortable and excited about the study. Additionally, the noise of the scanner was presented to the participants at this stage to avoid any surprise during the actual scanning. Furthermore, to ensure participation in the experiment and to make sense of the sensory stimulation provided in the scanner, children were introduced to a narrative about an enchanted forest where butterflies became invisible. Participants were asked to help make them visible again by wearing special devices (the soft vibrotactors) that could capture these invisible butterflies. The vibrotactors delivered gentle vibrotactile stimulation to the participants' skin, mimicking the sensations of butterfly wings. Participants were instructed to simply be aware of these sensations during the scan, with no further action required. They were also informed that they would only need to watch videos throughout the session (Pixar short films).

After this first familiarisation phase, the 7 soft vibrotactors were attached to 7 predefined body parts (see Fig. 2B) using medical tape and self-adhesive sports bandages. The predefined body parts for the CLD participants (children and adults) were: (1) Chin (just below the lower lip, close to the tip; lower face throughout the text); (2) Residual arm (palmar side); (3) Torso (just below the rib cage); (4) Leg (end section, just above the knee); (5) Foot (internal section, just above the midsection of the foot arch); (6) Thumb (on the palmar side distal phalanx) of the intact or dominant hand; (7) Forehead (around the midline; upper face throughout the text). Note that body parts 3, 4 and 5 were ipsilateral to the residual arm (missing-hand side). These body parts were chosen because they are the alternative effectors that congenital children tend to use for their daily activities (with the exception of the upper face, which was included as a control condition). For the control children and adults, a similar configuration was used, except that the actuator on body part 2 (i.e. the residual arm for the experimental group) was placed on the palmar side of the non-dominant wrist. For the control group, body parts 3, 4 and 5 were ipsilateral to the non-dominant hand.

After all the actuators were placed on the participant's body, they were taken to the scanner room to start the actual experiment and were invited to lie on the scanner bed. A few exemplar stimulations were delivered to each body part to ensure that the correct body part was stimulated and that the participant was able to clearly feel each stimulation. After this second familiarisation phase, the actual experiment could start. For the group of children, one parent sat with the experimenters in the control room (at BUCNI) or in the scanner room while wearing protective headphones (at CBU) to comfort the children during the session.

The MRI session consisted of three functional runs, one structural run, and a fieldmap run. The fieldmap was acquired after the first functional run. The remaining two functional runs and the structural scan followed in that order. The protocol was flexible and could be adjusted based on each participant's needs.

After each functional run, participants were asked if they felt the SPA stimulation (butterflies) throughout the experiment. Positive reinforcement was provided by mentioning how many butterflies were caught with their help. Before the final structural run, where no tactile stimulation occurred, participants were informed that all butterflies had been captured, but an additional scan was necessary to confirm there were none remaining.

To ensure that participants remained still and comfortable throughout the scanning process, cartoons were projected and visible to participants through a mirror throughout the session. The passive nature of watching cartoons helped the children keep still, essential for acquiring high-quality imaging data over the multiple scans. We confirmed that the cartoons minimally affected S1 topography mapping prior to the start of the study using several pilot sessions, where we compared selectivity patterns across body parts with and without cartoon watching. Moreover, any age-related differences that might be causing attentional changes to the vibrotactor stimulation were accounted for by the age-matched control groups. Additionally, avoiding a motor task eliminated head movements and other potential performance differences between groups, especially between younger children and older children, or relative to adults.

**MRI data acquisition.** All MRI measurements were acquired using a Siemens 3 T Magnetom Prisma scanner. At the BUCNI, a 30-channel head coil provided a clearer view of the screen due to the removal of the two frontal coils. At the MRC Cognition and Brain Sciences Unit (CBU), a 32-channel head coil was used. Signal-to-noise analysis confirmed no significant difference between the data collected at the two sites, and activity profiles were also comparable. Task fMRI data were acquired in both sites using a multiband GE echo planar imaging sequence with an acceleration factor of 4. The field-of-view (FOV) consisted of 72 slices (TR: 1450 ms, TE: 35 ms, FA: 70°) with a spatial resolution of 2 mm isotropic. A whole brain anatomical T1-weighted (MPRAGE) image was also collected with a 1 mm isotropic spatial resolution (FOV: 192 × 192 × 176, TR: 2530 ms, TE: 3.34 ms, FA: 7°, TI: 1100 ms, GRAPPA factor: 2).

Three separate sensory stimulation sequences were created using a custom-made Matlab script using a block design. Each run consisted of 5 block repetitions for each of the 7 tested body part conditions, resulting in a total of 15 block repetitions per body part across the three runs. All analysed participants completed the three runs. As achieving perfect first-order counterbalancing wasn't feasible due to the specific number of blocks and repetitions, we designed an optimal first-order counterbalancing across the three functional runs. This ensured that each body part was stimulated roughly the same number of times after each other body part throughout the entire experiment. Notably, all participants experienced the same three sequences, but the run order was randomised for each individual.

Tactile stimulation within each 9 s block began 1 s after the block started to allow for pressure stabilisation and lasted for 7.5 s, followed by 0.5 s of no stimulation before the next block began. Each stimulation involved triplets of frequencies (5 Hz, 15 Hz, 30 Hz) repeated five times. The frequency changed every 400 ms with a 100-ms gap between frequencies. This variation in frequency aimed to prevent peripheral or central adaptation[81–84], which could result in reduced sensory processing.

The tactile blocks were interleaved with six 9 s 'null blocks' interspersed throughout the run, to allow for BOLD signal relaxation for baseline estimation. These null blocks were placed randomly with the following restrictions: No null block could occur at the beginning or end of the run. Null blocks were grouped in chunks of either one or two consecutive blocks. Chunks of two null blocks had to be separated by at least two experimental blocks. In addition, two 16 s null blocks were added to the beginning and end of each run. The total run time was 401.6 s (277 volumes with a TR of 1.45 s).

**MRI analysis.** For each individual, cortical surfaces were estimated from the structural images using FreeSurfer 7.4.0[85,86]. We used the Connectome Workbench software (https://www.humanconnectome.org/software/connectome-workbench) to visualise the individual (native space) and group (standard space) topographical maps projected on the surfaces and to ensure accurate spatial registration between the structural and functional volumes, as well as to verify the precise alignment of the ROIs. Connectome Workbench was also used to map the volumetric maps to the surface space.

**fMRI pre-processing and first-level analysis.** All MRI data pre-processing and analysis were carried out using FMRIB Software Library[87] (FSL version 6.0) as well as scripts written in MATLAB (version R2020b) and R (4.2.0), which were developed in-house.

To ensure that the functional scans were well aligned for each participant, we calculated a midspace between the three runs. This midspace represents the average space where the images are minimally reoriented. Each scan was then aligned to this midspace using FMRIB's Linear Image Registration Tool or FLIRT[88,89]. Finally, the midspace and functional images were then aligned to the high-resolution anatomy image using SPM12. fMRI data were subject to standard pre-processing, which included motion correction using FMRIB's MCFLIRT[89], brain extraction using Brain Extraction Tool[90], temporal high-pass filtering, with a cutoff of 90 s and spatial smoothing. Signal distortions caused by B0 field inhomogeneities were corrected using B0 field maps. The fieldmap magnitude and phase images were processed with FSL's *fsl_prepare_fieldmap*. No slice-timing correction was applied because of the relatively short TR (1450 ms). A 4 mm (2× voxel resolution) full-width at half maximum (FWHM) kernel was used for smoothing data in the univariate analyses, while a 2 mm FWHM kernel was used for multivariate analyses.

To localise brain regions activated during tactile stimulation, we used a voxel-based General Linear Model as implemented in FEAT (FMRI Expert Analysis Tool). For each run, seven regressors were created, one for each body part condition. These regressors were convoluted with a double-gamma function to account for the delayed BOLD response. Additional first-derivative regressors were included to capture temporal variations in the BOLD signal and estimated head movements (six parameters). Finally, volumes with excessive motion or unusual signal intensity (frame-to-frame displacement larger than 0.9 mm) were flagged for exclusion (using the FSL function *fsl_motion_outliers*) and included in the model as additional regressors of no interest (one for each outlier). The 0.9 mm threshold was selected based on previous methodological work by Siegel et al.[91]. For children, across the three runs (totalling 99 runs for 33 children), the vast majority of the data (85% of all runs) contained 10 or fewer identified volume outliers which is 4.63% or less of the total volumes with stimulation (around 215 volumes out of 277 total volumes per run) were flagged as outliers. This low proportion of outlier data suggests that children generally remained still during the stimulation. Importantly, when comparing the distribution of outliers between the two groups of children using a chi-square test, we observed no significant difference ($X^2 = 23.41$, df = 23, $p$ value = 0.437). Nine children showed a larger number of outliers in at least one of the three runs (min = 11 and max = 69, corresponding to 5.09% and 31.94% of the total volumes with stimulation). These children were visually inspected to ensure the estimated maps were as expected. Since the maps were consistent, we decided to keep these datasets. For adults, 97% of the outliers across the three runs were less than or equal to 1 (0.46% of the total volumes with stimulation).

**Second-level and group-level analyses.** For each participant, the three first-level contrast images of each body part obtained from the first-level analysis were further processed with a second-level analysis using a fixed-effect model in FSL. The resulting images were normalised to MNI space and used for the line analyses (Figs. 3–5).

Before conducting the group-level analysis, we flipped these second-level images along the $x$-axis for CLD participants missing the right hand and for control participants who were left-hand dominant. This standardised the left hemisphere as the 'deprived/non-dominant' hand hemisphere across all participants. This step was necessary, as the tactile stimulators were aligned with the missing/non-dominant side. We then conducted two types of group analysis using these second-level contrast images. First, we estimated group-level activation to identify regions engaged during stimulation of each body part for visualisation purposes (Figs. 2C and S4). Second, we contrasted body part maps between the CLD and control groups to determine differential activity in response to stimulation (Fig. S6). Both analyses employed a random-effect model in FSL (FLAME 1) and a cluster correction to control for multiple comparisons with the following parameters: *Z threshold* = 2.58 (corresponds to a two-tailed $p$ value of approximately 0.01) and *cluster p threshold* = 0.05. The resulting thresholded maps were then projected on a standardised surface for visualisation purposes (Figs. 2C, S4 and S5-Arm). Where statistical significance was not reached, uncorrected maps ($p < 0.01$) are presented (Supplementary Fig. S6, except for Arm).

## ROIs definition

As we were interested in studying both global (i.e. along S1) and local (i.e. hand territory) changes due to age and/or deprivation, we defined two different types of ROIs within S1. Specifically, we defined a group ROI for the global analysis and individual ROIs for the local analysis. The group ROI, derived from Glasser atlas[92] areas 1 and 2 in MNI space, provided an unbiased and efficient means of capturing global changes across the somatosensory cortex. In contrast, the individual ROIs, defined based on thumb peak locations, allowed for more accurate and sensitive detection of local changes within the hand territory, given its known anatomical variability[93]. We also defined an ROI to study activity changes in S2 (see Supplementary Results). The ROIs were focused on the deprived hemisphere in individuals with CLD and compared with the non-dominant hand hemisphere in controls. In addition, homologous ROIs in the intact/dominant hand hemisphere (in CLD/controls, respectively) were constructed for control analyses involving the intact/dominant-hand thumb.

**Global S1 analysis (bins-ROI).** We used anatomical atlases to investigate global changes for all body parts across the entire postcentral gyrus strip. The ROI was defined in each hemisphere on the standard flat maps (FS_LR 32k) of the Human Connectome Project. Hemisphere-specific rotations (15° counter-clockwise for the left and clockwise for the right hemisphere) were applied to the flat maps, aligning the central sulcus perpendicular to the horizontal plane using the fundus of the central sulcus (defined by the lowest point of BA3b based on the FreeSurfer atlas) as the pivot (index 133 and 130 for left and right hemispheres, respectively; these points correspond to the origin flat map coordinate). We first created a mask for the postcentral gyrus using Glasser's multimodal parcellation encompassing BA1 and BA2[92]. Since this combined ROI did not fully cover the medial aspect of the central sulcus, we also added region BA5m, which corresponds with the medial aspect of FreeSurfer's S1 and also overlapped with the foot and leg activity in our cohort (see Figs. 2C and S3). Within the highest and lowest points of the defined mask on the flat map, fifty horizontal lines equally spaced were defined, resulting in 49 smaller rectangles with their longer axis oriented with the anterior-posterior axis. These rectangles were subsequently mapped onto cortical surface vertices, resulting in 49 left and 48 right ROIs spanning the medial-to-lateral extent of S1. The different number of bins between the two hemispheres is due to small differences in the geometry of the two surfaces, resulting in zero vertices enclosed in the most medial rectangle. This procedure yielded surface ROIs that were subsequently transformed into volumetric ROIs. For comparison, we also extracted an equivalent ROI based on BA3b, as presented in the Supplementary Materials (Fig. S7).

**Functionally-defined hand ROIs.** To create a hand-specific ROI for each participant, we employed the functional second-level contrast of thumb vs baseline from the intact and dominant hands for the CLD and control groups, respectively. For each individual, the functional response (z-statistic) elicited in the hemisphere contralateral to the dominant/intact hand during thumb stimulation was averaged within each bin defined in the aforementioned S1 global ROI (Figs. 2C and S3). For each participant, the bin with the peak activity (highest z-statistic) was identified, and its corresponding MNI coordinate was used to locate the homologous bin in the hemisphere ipsilateral to the dominant/intact hand (i.e. the deprived/non-dominant hemisphere) to estimate the location of the missing/non-dominant thumb. We have previously extensively validated this procedure[5,94]. Finally, since the thumb defines the lateral boundary of the hand area[95–97], the hand ROI was defined by combining the peak bin with eight bins medial to it and one bin lateral to it, encompassing a total of ten bins. We also defined an S2 ROI according to Glasser's parcellation (see Supplementary Methods for full details).

## ROI analysis

**Local remapping: univariate analysis.** We investigated *local plasticity* effects specifically within the deprived (missing) hand ROI. We averaged the contrast values (z-statistic) from the second-level analysis within the ROI for each body part compared to baseline, for each participant. These averaged values were then used for the group analysis. This analysis provided a basic measure of activity changes within the hand representation, as measured in prior studies[5,6]. The same approach was further utilised for studying S2 (Fig. S14).

**Local remapping: representational similarity analysis (RSA).** To confirm that activity levels for the different body parts in the S1 (missing) hand area provide functional information, and not just noise, we employed RSA[98]. For each participant and run, we extracted the beta weights (representing brain activity) estimated with FEAT from each ROI. We then calculated the pairwise distance between these beta patterns across all body parts (excluding the thumb), using the cross-validated Mahalanobis distance[99]. This distance reflects the dissimilarity between brain activity patterns evoked by different body parts. To improve the reliability of these distance estimates, we applied multidimensional noise normalisation. This approach down-weights the influence of noisy voxels within the ROI, focusing on the more reliable signal. The Mahalanobis distance offers two key advantages: (1) Multivariate noise normalisation removes noise that might be correlated across voxels, leading to a more accurate measure of dissimilarity between activity patterns[99]. (2) Cross-validation ensures that random noise alone won't inflate the distance between patterns. If the only difference between two patterns is noise, their average dissimilarity will be close to zero. Consequently, the dissimilarity measure can even be negative[100]. Importantly, dissimilarities that are significantly greater than zero provide evidence that the ROIs contain distinct, task-related information (e.g. separate representations for different body parts). The cross-validated Mahalanobis distance calculations were performed using the Python library for RSA, rsatoolbox version 0.0.4 (https://github.com/rsagroup/rsatoolbox). The same approach was further utilised for studying S2 (Fig. S14).

## Global S1 remapping

A two-dimensional analysis procedure was employed to quantify functional activity elicited in the deprived hand hemisphere during tactile stimulation of each body part (foot, leg, torso, arm, upper face, lower face). First, for each individual, the second-level estimated contrast (z-statistic) related to each body part (versus baseline) elicited in the deprived/non-dominant hand hemisphere (or the hemisphere contralateral to the intact/dominant hand for thumb-related analyses) was averaged within each bin, defined in the global ROI. This resulted in a medial-to-lateral line plot depicting activity across S1 (Figs. 3–5). For visualisation purposes (Figs. 3 and 4), these activity plots were averaged across participants within each group, conveying a distribution of peak activity along S1. Note that for two participants, a clear peak activity (z-statistic exceeding zero, indicating a statistically significant positive activation) was not identifiable for the torso, and therefore these two contrasts (out of 462 total contrasts, i.e. $7 \times 66$ participants) were excluded from further analysis. Additionally, double-peaked activity patterns were observed in a subset of participants ($n = 20$) and contrasts. In these cases, the peak closer to the group average for that specific contrast, which typically aligned with the expected location based on the classical homunculus model, was chosen manually. Most (15/20) of the double peaks were observed in control adults, particularly when stimulating the arm (7/15) and leg (4/15). Contrasts that were excluded and with double peaks are reported in Supplementary Table 12 for further reference. Following data cleaning, based on the peak activity identified for each individual participant and body part, we conducted a position analysis. This analysis investigated the location (bin) of the activity peak within the

activity line profiles to evaluate changes in the topographic position of body part representations. To complement this, we also conducted an activity analysis (see Supplementary Materials and Fig. S8). This analysis examined peak values (z-statistic) identified in the activity line profiles to assess changes in activity for body part representation.

## Brain-behaviour correlation analysis

For each CLD participant, we conducted an analysis to examine the correlation between behavioural deviance and brain position deviance across different body parts. This analysis involved several steps: First, we quantified the behavioural deviance for each body part by calculating the difference in the percentage of usage of each body part between the individual participant and their age-matched peer group (Fig. 7A). We thus obtained for each participant five behavioural deviance measures, one for each body part. These measures indicate an increase/decrease in body part use by the participant, compared to the control peer group.

Second, for both control and CLD participants, we measured the bin distance between each body part peak position (i.e. bin location) and the participant's functional thumb position. Then, for the CLD participants only and each body part, a deprivation-related position deviance was defined as the ratio of the participant's body-part-from-hand distance to the average body-part-from-hand distance of the control peer group. We thus obtained for each participant five brain-related deviance measures, one for each body part. The resulting ratio reflects the extent of deviation from the typical bin location of each body part.

Finally, to evaluate the relationship between behavioural changes and brain changes, we calculated the Spearman correlation coefficient for each participant between their behavioural deviance measures and brain position deviance measures (Fig. 7B). This correlation provided an individual-specific measure of how deviations in body part usage were related to deviations in brain activity locations. Through this approach, we aimed to uncover the nuanced relationship between compensatory behaviour and corresponding brain changes in individuals with CLD. For information about the permutational analyses conducted to verify the statistical significance of the Spearman correlation coefficients, see the 'Statistical analysis' section.

## Model

We used a computational model based on a key homeostatic stabilisation mechanism, synaptic scaling[31] (see ref. [14] for a review), to investigate whether our empirical findings could be explained by this mechanism. The model is available here: https://colab.research.google.com/drive/1L9Bv3mbHKG1wh2oF2VHgOhZljOiDjsmH?usp=sharing.

**Model description.** The model (Fig. 6) is designed as a shallow neural network with 50 internal neurons, each representing regions that receive input from ten body parts (genitals, foot, leg, torso, arm, hand (including palm and fingers or corresponding residual nerves in CLD individuals), face (including upper face and lower face) and throat). The internal neurons mimic the activity along the S1 strip $(Y_1, Y_2, \cdots, Y_{50})$. The activity of a neuron $Y_j$ is calculated as $Y_j = \sum_{i=1,2,\cdots,10} W_{j,BP_i} \cdot X_{BP_i}$, where $W_{j,BP_i}$ is the weight of the connection between body part $BP_i$ and internal neuron $j$, and $X_{BP_i}$ are stimulation values ($X_{BP_i} = 1$ for the stimulated body part, $X_{BP_i} = 0$ for the non-stimulated body part). If only one body part $BP_l$ is stimulated ($X_{BP_{j\neq l}} = 0$), the activation of the neuron $Y_j$ is simply calculated as $Y_j = W_{j,BP_l} \cdot 1 = W_{j,BP_l}$.

As mentioned in the results section, the aim of the model is to simulate homeostatic plasticity via a synaptic scaling mechanism[14,30,31]. This is achieved through two main steps. First, an initial Mean Neural Activity (MNA) is calculated for a neuron $j$:

$$MNA_j^{init} = \sum_{i=1,2,\cdots,10} W_{j,BP_i}^{init} \cdot P_{BP_i} \qquad (1)$$

where $W_{j,BP_i}^{init}$ is the initial (seed) weight (see below) between the body part $BP_i$ and internal neuron $j$, $P_{BP_i}$ is the stimulation probability of a body part. We considered equal probability ($P = 0.05$) for all body parts except for the hand (both palm and fingers), which were assumed to have a probability of 0.15 (which means that the hand is stimulated 15% of the time). For a detailed analysis, please see the section 'Justification of stimulation probability settings' in the Supplementary Materials. Second, all weights of the neuron $j$ are updated according to the following formula:

$$W_{j,BP_i}^{hom} = MNA^{hom} \cdot \frac{W_{j,BP_i}^{init}}{MNA_j^{init}} \qquad (2)$$

After the weight update, $MNA_j$ value is equal to the homeostatic value $MNA^{hom}$. For this implementation, we used an $MNA^{hom}$ equal to 0.5.

Note that for the CLD group, the hand will be referred to as *hand nerves* (including palm nerves and fingers' nerves) to consider residual hand input from the residual arm (see Results section). During model evaluation, when arm stimulation occurs, it is assumed that the palm/finger nerves are also stimulated. The activity of an internal neuron $j$ is calculated as $Y_j = W_{j,Arm} \cdot 1 + W_{j,Palmnerve} \cdot 1 + W_{j,Fingersnerve} \cdot 1$. Peripheral reorganisation is also considered during the homeostatic adjustment of weights by considering the stimulation of the residual nerves. To study remapping without behavioural differences between the groups, the stimulation probabilities of arm and palm/fingers residual nerves of CLD individuals remain the same as the arm and palm/fingers probabilities of CTRs.

**Initial weights of the model.** Our model posits that there is a genetically pre-determined pattern of connections between the body parts and internal neurons (see Fig. S9). These initial connection weights are modified by homeostatic mechanisms. The initial weights are represented by bell-shaped curves, with their peak positions uniformly distributed across the S1 homunculus strip and in line with the typical topography of this region (please see the Supplementary Section 'Setting of initial connection weights' for a detailed description of how these weights are calculated). To ensure a weak connection of all body parts with all neurons[15], the bell-shaped weights are vertically offset by a small constant value equal to 0.05. The density of tactile nerve fibres in the skin is not uniform; it varies significantly across different parts of the body. Notably, the hand (including the palm and fingers) has a high concentration of Aβ myelinated tactile fibres[101]. This dense innervation is related to greater tactile sensitivity[102] and a larger representation of the hand in the brain[103]. In the model, this is reflected by assigning stronger connections between the hand and the somatosensory homunculus compared with the connections related to other body parts (Fig. S9, left plot). For the CLD group, to reflect the reduced influence of residual nerves, the original weights associated with palm and fingers are multiplied by a factor of 0.15 (see Fig. S9, right plot). This adjustment reflects the weaker connection between residual nerves and internal neurons. Notice that homeostatic adaptation modifies the strength of the individual connections but does not change the relative proportions of the weights between body parts and a given neuron.

**Key observations to be explained by the model.** To evaluate the explanatory power of our model, we identified four key empirical findings from the global S1 remapping analysis and investigated whether the model replicates these findings, which is a standard approach in assessing computational models[32–34,104]. The model must account for both the differences between the CLD and CTR groups, as well as the underlying patterns common to both groups.

**Observation 1.** The activity peak during arm stimulation is both higher (main effect of limb difference for activity: $F_{1,\ 226.50} = 71.70$, $p < 0.0001$) and laterally shifted (main effect of limb difference for position: $F_{1,255.27} = 116.68$, $p < 0.001$) in the CLD group compared to the CTR group, as shown in Figs. 5-Arm and S8-Arm, respectively. Notably, this residual arm stimulation also activates a large portion of the hand area (approximate position indicated by the dashed line). This is likely due to the dense concentration of nerves from the missing hand that are anchored at the tip of the residual arm, which still respond to stimulation.

**Observation 2.** In the CLD group, the activity peaks for the foot ($F_{1,246.56} = 7.43$, $p = 0.007$), leg ($F_{1,246.56} = 5.81$, $p = 0.017$), torso ($F_{1,246.56} = 7.84$, $p = 0.005$; all main effects of limb difference), lower face (although, this was only significant for children, $F_{1,255.54} = 4.68$, $p = 0.032$) and upper face ($F_{1,\ 317.97} = 2.99$, $p = 0.084$, $BF_{10} = 0.52$) are shifted towards the hand area compared to the CTR group, as shown in the corresponding subplots of Fig. 5. This suggests that a local deprivation triggers a global cortical remapping of other body parts to shift towards the area corresponding to the missing hand—laterally for the foot, leg and torso, and medially for the upper and lower face. This shift highlights the brain's adaptability in reallocating cortical space following sensory deprivation.

**Observation 3.** The activity peaks were overall significantly higher in the CLD group than in the CTR group (main effect of limb difference for activity: $F_{1,\ 57.86} = 21.38$, $p < 0.001$). Alongside the shifts described in Observation 2, this increase in activity suggests that the deprivation of the hand leads not only to a spatial reorganisation but also to a general increase in cortical activity across the entire S1 region. This heightened activity may further impact remapping matrices that are commonly used (such as the activity increase in the deprived cortex, which we replicate in the current study).

**Observation 4.** CLD and CTR groups exhibit a decrease from the foot to the leg and torso (linear trend: $F_{1,56.45} = 18.30$, $p < 0.001$), followed by an increase from the upper face to the lower face (linear trend: $F_{1,59.42} = 34.39$, $p < 0.001$). This observation serves as a critical benchmark for our model.

### Statistical analysis

Statistical analyses were performed using custom scripts written in Matlab R2020b (MathWorks), R version 4.1.3 (R Core Team, 2022) with RStudio (2021.09.0 Build 351), and Python 3.10.6 with Spyder 5.3.3. Behavioural and fMRI data were analysed using Linear Mixed Models implemented in the R package *lmerTest*[105], based on the *lme4* package[106]. Model parameters were estimated, and variance partitions were derived using the *anova* function in R, which are the results reported throughout the text.

In all models, participants were treated as random effects. Additionally, tasks in the behavioural session, body parts in both behavioural and fMRI sessions (for univariate analysis), and pairwise labels (for distance analysis) were also considered random effects. The predictor variables generally included age (children/adults) and limb difference (CLD/CTR), unless otherwise indicated. Body parts were included in models predicting percentage of usage and functional activity, with the exception of the lower face in brain analysis, as it was designed as a control body part. For completeness, a model including the lower face was also run. In the behavioural analysis, face usage was analysed separately using a chi-square test to examine distribution differences across the four groups.

In the behavioural analysis, face usage was separately examined using a chi-square test to determine if there was a different distribution of use across the four groups. For evaluating differences in face usage among the 10 CLD participants who showed face usage (five children and five adults), a bootstrap analysis with 5000 iterations was employed. At each iteration, participants were sampled with replacement and divided into two groups mirroring the original group sizes to create a null distribution, estimating the $p$ value as the proportion of bootstrap mean differences that were more extreme than the observed mean difference.

To test the significance of the correlation between brain measures and compensatory behaviour, a permutation test was conducted. This involved 5000 iterations where behavioural and brain deviance data were shuffled within each participant, maintaining the original structure but randomising pairings of deviance measures. Spearman correlation coefficients between behavioural and brain deviance measures were calculated for each participant in each iteration and averaged for each group (children and adults) to form a null distribution of mean correlation coefficients. The observed mean correlation coefficient was compared to this distribution to determine the empirical $p$ value, representing the proportion of permuted datasets producing a correlation coefficient as extreme or more extreme than the observed coefficient. This analysis assessed the significance of observed brain-behaviour correlations, shedding light on the relationship between behavioural changes and corresponding brain changes in CLD individuals.

Contrasts of interest were estimated using the R package *emmeans 1.10.1*[107]. For trending results of interest, we reported the corresponding Bayes factor ($BF_{10}$), defined as the relative support for the alternative hypothesis. While it is generally agreed that it is difficult to establish a cutoff for what consists sufficient evidence, we used the threshold of $BF < 1/3$ as sufficient evidence in support of the null, $BF > 3$ as sufficient evidence in support of the alternative hypothesis, and $1/3 < BF < 3$ as inconclusive evidence, consistent with others in the field[108,109].

### Reporting summary

Further information on research design is available in the Nature Portfolio Reporting Summary linked to this article.

## Data availability

The processed data generated in this study, as well as the raw behavioural data, have been deposited in the OSF database [https://osf.io/npzxb/] (https://doi.org/10.17605/OSF.IO/NPZXB). Source data are provided with this paper.

## Code availability

The code used to implement the computational model used in this work can be found here: [Model_Manuscript_public_v2.ipynb] (https://colab.research.google.com/drive/1L9Bv3mbHKG1wh2oF2VHgOhZljOiDjsmH?usp=sharing).

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

## Acknowledgements

Funding was provided by the Wellcome Trust (215575/Z/19/Z) and the Medical Research Council (MC_UU_00030/10), awarded to T.R.M. Z.S. and M.H. were supported by the Czech Science Foundation (GA ČR), project no. 20-24186X. The authors thank the participants for their help and commitment. The authors thank REACH and Opcare for crucial help with participant recruitment. The authors thank Dollyane Muret for project development and piloting. The authors thank Allie Williams for helping with data collection and Isabel Castelow, Emerald Grimshaw, and Nefeli Strongylaki for helping with data analysis. The authors also thank Tessa Dekker and Jörn Diedrichsen for insightful comments. For the purpose of open access, the author has applied a Creative Commons Attribution (CC BY) licence to any Author Accepted Manuscript version arising from this submission.

## Author contributions

R.T., Do.C. and T.R.M. designed the research; R.T., L.B., M.S. and M.K. performed the experimental work; R.T., L.B. and M.S. analysed the data; Z.S., M.H. and R.T. performed the computational analysis; H.A.S., Da.C. and J.P. contributed to hardware development; R.T. and T.R.M. wrote the first draft of the manuscript; R.T., L.B., Z.S., M.S., M.H., Do.C. and T.R.M. edited the manuscript.

## Competing interests

The authors declare no competing interests.

## Declaration of generative AI and AI-assisted technologies in the writing process

During the preparation of this work, the author(s) used ChatGPT in order to light-touch proofreading. After using this tool/service, the author(s) reviewed and edited the content as needed and take(s) full responsibility for the content of the publication.
