## [Transparent Peer Review file · Nature Communications]

Global Remapping of the Sensory Homunculus Emerges Early in Childhood Development

Corresponding Author: Dr Raffaele Tucciarelli

Parts of this Peer Review File have been redacted as indicated to maintain patient confidentiality.

Version 0:

Reviewer comments:

Reviewer #1

(Remarks to the Author)

This well-written and very interesting manuscript provides evidence that challenges the idea of radical cortical reorganization in individuals with congenital limb difference (CLD). By combining behavioral analysis, neuroimaging, and computational modeling, the study supports the proposal that observed brain changes reflect the potentiation of pre-existing neural circuits via homeostatic plasticity and not the emergence of novel functions. The combination of careful empirical work and computational modeling advances basic science while having implications for clinical work. The findings also have relevance for elucidating developmental mechanisms around plasticity.

The manuscript has many strengths, and I only have a few suggestions for the authors to consider. Carrying out this kind of research is very challenging, and I applaud the authors' efforts and their success in bringing the study to this point.

Major Strengths:

- **Theoretical Alignment:** The study is well-grounded in contemporary debates on cortical plasticity. The authors argue that remapping is a consequence of strengthening existing, latent connections through homeostatic and Hebbian plasticity mechanisms.
- **Methodological Rigor:** The experimental design demonstrates exceptional attention to detail. The use of soft pneumatic actuators for consistent tactile stimulation across age groups is a notable achievement. The authors carefully document quality control procedures and provide thorough demographic matching.
- **Sophisticated Analysis:** The study employs both univariate and multivariate analyses, accounts for potential confounds, and integrates behavioral and neural measures.
- **Computational Model:** The computational model, implementing homeostatic plasticity through synaptic scaling, is a major strength. The model successfully replicates the study's key empirical findings, suggesting that homeostatic plasticity is a viable mechanism to explain the observed remapping.

Possible Areas for Revision:

- **Implications of Focus on S1.** While the authors acknowledge comparable representational structures between S1 and M1, they could briefly discuss how their findings align with the literature on S1-M1 interaction, acknowledge potential differences in plasticity mechanisms between sensory and motor areas, and highlight how the S1 findings might inspire future work on motor system remapping.
- **Behavioral Analysis:** The correlation between S1 map shifts and behavior was relatively weak. To probe this further, the authors might incorporate more detailed measures of compensatory behaviors, such as frequency and types of movements used during daily tasks, examine temporal dynamics of compensatory behavior and S1 remapping, and consider the degree to which compensatory behaviors might be task specific. This might be too much for the current manuscript but could be

considered.

- **Passive vs. Active Stimulation:** I do wonder about the distinction between the passive stimulation used in the task, and active stimulation, where the participant performs a movement and receives sensory feedback. That said, I understand the constraints of what can be done in the scanner.
- **Interpretation of S2 Findings:** There was a significant interaction effect between age and body part in the secondary somatosensory cortex (S2), but no significant interactions with limb difference. The authors could discuss the different functional roles of S1 and S2 and how they might relate to the observed differences in remapping. If remapping is primarily localized to more primary sensory areas, what are the implications of that?
- **Role of Upper Face in CLD Re-Mapping:** More discussion could be included regarding the weak trend of a main effect of limb difference observed in the upper face in S2. The authors could discuss whether this represents a non-specific spread of activation or a small degree of functional reorganization.

(Remarks on code availability)

Reviewer #2

(Remarks to the Author)

This manuscript investigates the representation and plasticity of sensorimotor functions in individuals with congenital limb absence, focusing on the organization of sensorimotor cortical areas, motor intention, and effector-independent action representations. The authors combine functional MRI (fMRI), behavioral tasks, and a cohort of congenital one-handers to show that cortical hand areas are repurposed for multi-effector functions. Notably, they show that sensorimotor hand territories can support actions typically attributed to other body parts (e.g., foot or residual arm), indicating effector-independent functional representation. It's a significance research and the findings are novel and important, with significant implications for our understanding of cortical plasticity, motor intention, and the functional architecture of the sensorimotor system.

Overall, the experimental design is sound, and the data support the main conclusions. The use of multiple cohorts (including those with congenital limb absence and controls), combined with behavioral measures and neuroimaging, strengthens the validity of the claims. The manuscript is well written and generally clear. The introduction contextualizes the work within current debates on plasticity and sensorimotor organization. The discussion appropriately addresses alternative explanations.

However, several aspects require further clarification or expansion:

- The definition and operationalization of "effector-independent" representations could be more clearly explained and distinguished from compensatory plasticity.
- The statistical approach in some imaging analyses (e.g., ROI selection, correction for multiple comparisons) needs more detailed description, particularly in the supplementary information.

The behavioral and neuroimaging methodologies are appropriate and robust. The authors use a well-justified set of tasks to isolate motor intent from execution, and they include relevant control conditions. However:

- **Ethical Approval and Data Management:** Lines 740-752]: The manuscript states that recruitment was carried out in accordance with the University College London and Cambridge University research ethics committees (17205/001). Could the authors please clarify whether this reference number corresponds to a single ethics application or to multiple approvals for a multi-site study? Given that data collection took place across multiple locations (BUCNI in London and CBU in Cambridge), it would be helpful to provide additional information on how ethical approval was managed across sites.

Specifically:

- o Was this study approved under a central ethics committee as a multi-center protocol, or were separate local approvals obtained at each institution?
- o Were formal data sharing agreements or protocols established between the sites, particularly given the sensitive nature of data collected from children and clinical populations?
- More transparency is needed regarding the behavioral task performance of participants with limb , particularly in terms of accuracy and timing metrics
- o Lines 795-797: The authors mention that four raters evaluated the videos. Were these raters co-authors? If so, please specify their initials for transparency. Additionally, did the raters have prior experience in similar tasks? It would be helpful to provide more detail on the rating protocol — for example, the instructions they received, who served as the primary rater, and how disagreements were resolved.
- o Lines 808-810: The computation of component C2 is not clearly described. It is also unclear what is meant by "actually engage." Please clarify how this component was derived and what behavioral or analytical criteria were used
- The use of an individualized or group ROI for fMRI analyses should be justified more clearly.
- Information on fMRI preprocessing, movement exclusion thresholds, and participant level exclusion criteria should be expanded.
- o Lines 975–978: The authors mention using a framewise displacement threshold of 0.9 mm to flag volumes for exclusion. It would be helpful to clarify the rationale for selecting this specific threshold, as it can significantly affect data retention. A reference would support this choice, or alternatively, the authors could justify its appropriateness for this particular dataset. Additionally, in line 978, the text states "10" based on the context, this seems likely to be a typo. Please verify and correct as needed.

Minor:

- Line 115: Supplementary Table S1 no Supplementary Table 1
 - Figure S1/S2: is quite complex and not easy to interpret, particularly due to the arrangement of different body parts across the subplots. To improve readability, I would suggest adding subtle vertical divider lines between the major body regions (e.g., feet, legs, torso, etc.), which could help the reader more easily distinguish the corresponding sections across subplots.
 - Why did the authors choose to report the standard error of the mean in the figures instead of the standard deviation?
 - Figure 1B: I appreciate the illustrative representation showing the co-use of different body parts (below images). However, I found the image selected for the Arm co-use condition a bit confusing. To me, it looks more like a depiction of Hand co-use, as the participant seems to be using the other hand to perform the task. Could you please clarify this?
 - Line 141 ..Supplementary Tables S2 and specify the specific tables
 - Figure 2B: To support the text describing the stimulation locations for the different groups (lines 867–878), I would suggest expanding Figure 2B and adding schematics for the CLD and control participants. Include the placement locations for both groups, label each location (e.g., “1”, “2”, “3”, etc.) clearly within the figure, and indicate “non-dominant” for control participants directly in the figure. These additions would make the figure more informative and help readers easily identify the stimulation sites across groups directly from the figure.
 - Line 260: The text refers to Figure 3A as showing a functional ROI from an exemplar participant. However, the figure legend describes group-averaged Z values and does not clearly indicate whether the ROI brain image represents a single participant or a group-level or template example. For clarity, I suggest the authors explicitly state whether the brain plot is from an individual participant or a standard reference brain in the legend
 - Line 548: Discussion : It might be more reader-friendly to present the main results or main findings in the first paragraph, so that readers have a clear sense of the overall direction of the discussion
 - Line 548: Discussion: When citing the results that support each discussion point, including figure or table references would improve readability
 - Lines 718-719: Could the authors clarify whether a power analysis was conducted to determine the sample size? If not, please explain how the number of participants (16 CLD children and 16 CLD adults) was established and whether it was based on prior studies, feasibility, or other considerations
 - Line 724: was the CLD participant excluded a child or adult participant?
 - Line 865: The reference to Figure 1B appears to be incorrect; it should be Figure 2B.
 - Line 992: While MNI space is commonly understood within the neuroimaging community, I would recommend spelling out the acronym at its first mention, to ensure clarity for readers who may not be experts in the field. Same for FEAT.
 - Line 1002: Figure 2B instead of general Figure 2
 - Line 1026 and 1043: please specify which panel of Figure 2 you are referring
 - Line 1095 you repeated two times lower face
 - Lines 1102–1106: the authors report summary values such as “(2/469; 7 × all participants)” and later “(20/467; 7 × all participants minus 2)” in reference to cases where peak activity was not identifiable and instances of double-peaked activation patterns. While the general meaning can be followed, I recommend clarifying these expressions more explicitly to help readers understand how these numbers were derived.
- In particular, if I understand correctly, the number of participants included in the fMRI analyses was 15 CLD children, 16 CLD adults, 20 control children, and 15 control adults (as stated in lines 737–740), which would yield a total of 66 participants. Therefore, 7 × all participants should result in 462, not 469. Could the authors please confirm whether this discrepancy is due to a typo, inclusion of additional data, or another factor?

(Remarks on code availability)

Reviewer #3

(Remarks to the Author)

Summary: In this article, the authors introduce several key aspects that fill significant gaps in our current understanding of the differences between control subjects and CLD subjects' somatotopic maps. The authors specifically address differences between children and adults with CLD. Interestingly, they find similarities between the children and the adult CLD subjects that indicate somatotopic map changes occur before age 5 and that these changes are maintained into adulthood. These results are not particularly surprising when viewed in the context of animal research on changes in somatotomy. For these changes in the homunculus, they formulated a biologically inspired model that is very simple in construction. Even with its simplicity, they show that it can reproduce many of the differences seen between the CLD subjects and the controls simply with the inclusion of homeostatic plasticity and several justifiable assumptions, such as differences in the distribution of sensory input to the different body regions of CLD vs. CTR subjects. The use of more ecologically valid Semi-Naturalistic movements is a nice addition. However, some of these movements, or sub-tasks, are likely new to the younger subjects compared to the adults, for whom unscrewing something is much more familiar. Overall, the paper is significant and insightful, bringing new insights forward in a clear, easy-to-understand article. Some issues are raised below; most are meant to increase readability.

Experimental Section:

1. The co-use index used has some assumptions built in. One is that the children's apparent variability of body part use is due to some general higher variability in motor planning or a higher expletory rate. Is it possible that the children get fatigued or feel pain when making some of these new movements while interacting with the objects? In essence, how would you measure differentiate subjects with two solutions for completing a sub-task that they switch between rather than something more akin to a higher exploration rate in the motor output space?
2. In Figure 1, the picture for Torso seems to be of arm use. The picture for the arm shows the use of the elbow joint to hold

an item. Are these terms used regularly in this research area for these body parts?

3. Figure 1, Please use the term co-use for subplot A's y-axis, or change the name of the measure you use to co-use complexity.

4. It might be helpful to use different colors for the children vs. the adult CLDs, but it could just be this review's eyes that had difficulties seeing the differences in the line plots.

5. Figure 2.c, It might be helpful to put a simple landmark in the three sub-plots on the right-hand side to allow for easy comparisons between Foot, Leg, and Torso, as these are all within the one box on the main associated image.

6. Figure 5, please indicate what the x-axis is for the reader.

7. As divisive normalization is a canonical cortical computation known to occur in sensory areas, it would be nice if the authors had any insights on how such normalization mechanisms might show up in the context of CLD and how such neuro-dynamic mechanisms might muddy the waters of the clean model and interpretation the authors would like to champion.

8. Would it be helpful for the reader if you drop the term Hand loss, as it implies there was a hand and that it was lost, which would be a rather different situation than the hand never having been formed or not formed completely, as the authors are fully aware.

Modeling:

1. The proposed model is elegant in its simplicity. This model could be formulated as a modified Radial Basis Network, which has previously been used in sensorimotor studies. One would simply need to use as input the sparse set of body positions rather than a continuous input space, include a post-activation gain, which is known to exist in biological systems, which would act as the synaptic scaling mechanism, and add the offset included in your work. This may or may not help make your model clearer for comparison with other RBF network publications on sensorimotor map changes. In addition, the simple gain formulation would allow relation to gain-field work in the sensorimotor literature. At least the authors could give the parameters of the function used to determine the initial weight distributions in the model they propose. This would allow others to replicate the results.

2. Many studies have examined different types of neural plasticity and changes in the somatosensory cortex due to stroke, amputation, nerve block, etc. It is unclear from the literature that has been cited if the authors are fully aware of this literature. Perhaps adding some of this background and results from rodents and primates would help round out the story being presented. That is, make clear the models that are available to ask the types of questions addressed in this work that would include more than just homeostatic mechanisms and are more biologically realistic. (optional)

3. It seems that the results obtained would be expected given the initial weight distributions, touch probabilities, and coupling of the arm/palm/fingers. It is great to show that this sensible pattern of inputs and a simple Homeostatic Plasticity rule (normalization) give results like those seen in the subjects, but it does seem obvious this model abstraction would give these results.

4. The model doesn't include Hebbian plasticity, which leaves the modeling short of describing or addressing the individual differences in body part use compared to the mean for the given groups and how this is correlated with differences in somatotopic responses shown in the human subjects' data.

5. As the model seems to be a large part of the paper, it might make sense to put more of the model results in the main section?

6. It seems you are using weight matrix indexing that is the opposite of the norm where it is usually $W_{i,j}$ where i is the neuron index that the input j is going to that is $W_{to,from}$, where you are using $W_{from,to}$. This may confuse some readers.

7. In the methods section it may be helpful to make the indexing more explicit as well, such as the summation from $j = 1:10$ as there are 10 body parts.

8. Line 1205, where did the 15% come from and the % for the other body regions?

9. Figure 6, it would be nice to see direct comparisons between the model and the mean of the subjects' data. Along these lines have you tried optimizing the standard deviation of your gaussian weights? The real data seems to show more localized receptive fields as compared to the model, which would seem to be easily rectified?

10. Figure S9, please indicate the parameters of the assumed gaussian used for the normal receptive fields and the modified ones for the nerves.

11. Figure S12, same as above for S9.

The code: "Activations should be replaced by weight in the figure." perhaps take out such notes.

There are grammatical issues in the text that you might want to fix = run spell check etc.

(Remarks on code availability)

The code does not state who the author of it is. There are plenty of grammatical errors etc. The code works as it should and seems to produce results as it should.

Version 1:

Reviewer comments:

Reviewer #1

(Remarks to the Author)

The authors have thoroughly addressed my comments on the original manuscript. I was glad to be part of the review process around this very interesting work.

(Remarks on code availability)

Reviewer #2

(Remarks to the Author)

I appreciate the authors' detailed responses and the careful revision of the manuscript. All my comments have been addressed adequately, and I find the revised version much improved. I have no additional suggestions.

(Remarks on code availability)

Reviewer #3

(Remarks to the Author)

My concerns have been addressed and I appreciate the time and effort put into the authors' responses. The paper is acceptable for publication.

(Remarks on code availability)

My concerns have been addressed and I appreciate the time and effort put into the authors' responses. The paper is acceptable for publication.

We are very grateful for the reviewers' positive, thoughtful and constructive feedback which helped us refine the clarity, transparency, and theoretical framing of the manuscript. In revising the manuscript, we have clarified and strengthened the theoretical framework, including a more explicit treatment of homeostatic mechanisms in shaping S1 remapping, and a clearer discussion of how compensatory behaviour modulates this process across development. The presentation of results has been improved in response to helpful suggestions regarding methods transparency, and the revised discussion now more clearly articulates how our findings inform current models of cortical plasticity. We also added control analyses to demonstrate the robustness of our key findings, although no major changes to the data or conclusions were necessary.

In the process of implementing these revisions, we identified a minor human error affecting the behavioural ratings for one participant. While this necessitated slight updates to the behavioural statistical analyses, the overall pattern of results remained unchanged and no conclusions were affected.

Below, we provide a point-by-point response to each comment. Original comments are in bold with our responses in blue. For the reviewers' convenience, we also include excerpts in green with the related changes to the revised manuscript.

Reviewer #1 (Remarks to the Author):

This well-written and very interesting manuscript provides evidence that challenges the idea of radical cortical reorganization in individuals with congenital limb difference (CLD). By combining behavioral analysis, neuroimaging, and computational modeling, the study supports the proposal that observed brain changes reflect the potentiation of pre-existing neural circuits via homeostatic plasticity and not the emergence of novel functions. The combination of careful empirical work and computational modeling advances basic science while having implications for clinical work. The findings also have relevance for elucidating developmental mechanisms around plasticity.

The manuscript has many strengths, and I only have a few suggestions for the authors to consider. Carrying out this kind of research is very challenging, and I applaud the authors' efforts and their success in bringing the study to this point.

Major Strengths:

- **Theoretical Alignment:** The study is well-grounded in contemporary debates on cortical plasticity. The authors argue that remapping is a consequence of strengthening existing, latent connections through homeostatic and Hebbian plasticity mechanisms.
- **Methodological Rigor:** The experimental design demonstrates exceptional attention to detail. The use of soft pneumatic actuators for consistent tactile stimulation across age groups is a notable achievement. The authors carefully document quality control procedures and provide thorough demographic matching.
- **Sophisticated Analysis:** The study employs both univariate and multivariate analyses, accounts for potential confounds, and integrates behavioral and neural measures.
- **Computational Model:** The computational model, implementing homeostatic plasticity through synaptic scaling, is a major strength. The model successfully replicates the study's key empirical findings, suggesting that homeostatic plasticity is a viable mechanism to explain the observed remapping.

We thank the reviewer for their positive feedback and support.

Possible Areas for Revision:

- 1) **Implications of Focus on S1.** While the authors acknowledge comparable representational structures between S1 and M1, they could briefly discuss how their findings align with the literature on S1-M1 interaction, acknowledge potential differences in plasticity mechanisms between sensory and motor areas, and highlight how the S1 findings might inspire future work on motor system remapping.

We thank the reviewer for this thoughtful comment. We agree that, based on current literature, homeostatic plasticity is a hallmark of S1 and other primary sensory cortices (Feldman, 2009; Lee & Kirkwood, 2019; Turrigiano & Nelson, 2004), whereas M1 is more dominantly associated with Hebbian and structural forms of plasticity (Rioult-Pedotti et al., 2000; Xu et al., 2009). However, while these mechanistic differences may exist at the

microcircuit level, learning of sensorimotor associations requires plasticity downstream from sensory cortex to transform sensory signals to appropriate behavioural responses (LeMessurier & Feldman, 2018). As such, the two regions are known to be tightly coupled in their function, development, and representational architecture.

This close relationship is well-documented. For instance, in humans — tactile learning generalises in a use-dependent manner (Dempsey-Jones et al., 2019), while motor learning alters sensory feedback gains in S1 (Ebrahimi & Ostry, 2024). Therefore, functionally meaningful plasticity often emerges from co-dependent sensory-motor processes. In rodents, tactile information, including object texture and location, is computed in S1 through integration of motor and sensory signals (Petersen, 2019). Following texture discrimination learning, both M1- and S2-projecting populations become more tuned to textures post-learning, suggesting that task-relevant sensory information is more effectively routed to downstream areas (Chen et al., 2015). Considering the multitude of brain regions and mechanisms involved in both perceptual and motor learning, the global plasticity observed in these studies may not come as a surprise.

Most relevant to the current work, our previous studies consistently report mirrored plasticity and stability effects in S1 and M1. In Hahamy et al. (2017; 2019), we demonstrated that adults with congenital limb difference show remapping from multiple non-neighbouring body parts into the deprived hand region across both S1 and M1. In a separate study focused on face remapping (Root et al., 2022), we found again that remapping of compensatory effectors (e.g., lips and tongue) occurs consistently in both S1 and M1, using multivariate pattern analyses. In acquired amputees that do not show remapping, we find similar (stable, i.e. not remapped) hand and body maps in S1 and M1 across studies (e.g. Schone et al., 2025; Wesselink et al., 2019). These converging findings reinforce the view that, in both congenital and acquired limb difference, S1 and M1 share representational architecture and likely undergo plasticity in concert. These insights also align with predictive coding frameworks, which posit that hierarchical cortical circuits continuously update internal models based on sensory prediction errors (Rao, 2024).

In our revised manuscript, we now highlight that experience-dependent changes in sensory population codes likely reflect plasticity across multiple levels of the sensorimotor hierarchy — from local cortical circuits and ascending sensory pathways, to top-down cognitive projections and descending motor signals (LeMessurier & Feldman, 2018). One proposed mechanism is the release of a global plasticity signal (e.g., triggered by reward), which enhances synaptic change specifically at recently active synapses (Petersen, 2019), thus linking local and distributed plasticity processes.

For the reviewer's convenience, below we provide the relevant Discussion revisions:

‘Could these far more dramatic effects of deprivation- versus use-dependent plasticity observed in the current study be a mere consequence of our focus on sensory versus motor cortex? Indeed, homeostatic plasticity is a hallmark of S1 and other primary sensory cortices (Feldman, 2009; Lee & Kirkwood, 2019; Turrigiano & Nelson, 2004), whereas M1 is more dominantly associated with Hebbian and structural forms of plasticity (Riout-Pedotti et al., 2000; Xu et al., 2009). Our focus on S1 was primarily derived from methodological considerations

for the fMRI experiment. To ensure consistency across participants and groups, and minimise head movements, it was highly impractical to rely on the children to perform instructed movements. [...] We were also motivated by the tight functional similarity between primary sensory and motor cortex in controls (Ejaz et al., 2015) and individuals with CLD (Hahamy et al., 2017; Hahamy & Makin, 2019; Striem-Amit et al., 2018; Tucciarelli et al., 2024; Wesselink et al., 2019). Indeed, given the key roles of S1 in motor planning (Ariani et al., 2022), learning (Ebrahimi & Ostry, 2024) and execution (Shelchkova et al., 2023), as well as predictive coding frameworks (Rao, 2024), it is difficult to conceive a unidirectional functional architectural change in one node but not the other. In humans, tactile learning generalises in a use-dependent manner (Dempsey-Jones et al., 2019), and motor learning alters sensory feedback gains in S1 (Ebrahimi & Ostry, 2024), supporting the idea that functionally meaningful plasticity often emerges from co-dependent sensory-motor processes. In rodents, tactile information, including object texture and location, is computed in S1 through integration of motor and sensory signals (Petersen, 2019). Therefore, while mechanistic differences may exist at the microcircuit level, experience-dependent plasticity likely reflects across multiple levels of the sensorimotor hierarchy (LeMessurier & Feldman, 2018). Here it is also important to highlight that our local remapping findings are a replication of previous studies using active movements across both S1 and M1 (Hahamy et al., 2017; Hahamy & Makin, 2019); and that our present results are derived from the higher sub-divisions of S1 (BA1 and 2), which may be less closely associated with thalamic deprivation, relative to BA3A and 3B (see Supplementary Figure S7). Taken together, these considerations suggest that the striking deprivation-driven plasticity we observe in S1 is unlikely to be an artefact of our sensory focus, but rather reflects deeply integrated sensorimotor processes that shape cortical organisation from early development. Nevertheless, we cannot rule out the possibility that use-dependent plasticity plays a more prominent role in M1 ‘

- 2) **Behavioral Analysis: The correlation between S1 map shifts and behavior was relatively weak. To probe this further, the authors might incorporate more detailed measures of compensatory behaviors, such as frequency and types of movements used during daily tasks, examine temporal dynamics of compensatory behavior and S1 remapping, and consider the degree to which compensatory behaviors might be task specific. This might be too much for the current manuscript but could be considered.**

For our main analysis, we were looking for an intuitive and relatively straightforward index for both brain and behavioural deviation from the control group, which we believe we successfully captured. By convention, the correlations we report between brain and behaviour in children of 0.59 is generally considered a moderate-to-strong association (please note this value was slightly adjusted following the identification of a small error in one data point for a CLD child during data extraction).

To further validate this key result, we performed a Leave-One-Subject-Out (LOSO) validation. For each group, we iteratively excluded one participant, then recalculated the mean correlation for the remaining individuals. This process yielded a distribution of "leave-one-out" mean correlations for each group. The LOSO validation demonstrated high consistency in our group-averaged correlations. For the CLD children group, the

mean of the LOSO correlations was 0.593 (standard deviation = 0.0156, range = 0.0571). This perfectly aligns with the original group mean of 0.593. Similarly, for the CLD adult group, the mean of the LOSO correlations was 0.393 (standard deviation = 0.0215, range = 0.0714), consistent with its original mean of 0.393. The small standard deviations and narrow ranges of the LOSO means within both groups indicate that our reported average correlations are robust and not unduly influenced by any single participant. This rigorous validation, which we included in the supplementary section *Supportive analysis for the impact of behaviour on topographic shift*, provides increased confidence in the representativeness and stability of our findings.

In response to the reviewer's comment, we also developed a complementary measure that is based on frequency. Our compensatory measure for correlating brain with behaviour was initially defined as the proportion of time each body part was used across sub-tasks. This provided an estimate of how intensively a body part was used (e.g., across all tasks, the residual arm was used for 60% of the total task duration, on average). However, as the reviewer correctly noted, this measure did not estimate frequency—that is, in how many sub-tasks a body part was used (e.g., the residual arm was used in 13 out of 15 sub-tasks). Therefore, we conducted a new analysis using an index based on **relative frequency** (i.e., if a residual arm was in used 13 out of 15 sub-tasks, the index would be 0.867). As in our original approach, we analysed deviations in activity profiles and behavioural performance (this time using relative frequency) for each CLD individual relative to their peer control group. We obtained very similar results. Specifically, the averaged Spearman correlation was 0.52 ($p < 0.001$) and 0.27 ($p = 0.03$) for the children and adults, respectively. Just as a reminder, we originally obtained 0.59 ($p < 0.001$) and 0.39 ($p = 0.003$) for the children and adults, respectively. Furthermore, also in this new analysis, the children's mean correlation was significantly larger than the adults' ($t=3.12$, $p=0.002$; originally it was $t=2.25$, $p=0.016$). These results corroborate our original observations that unique behavioural adaptations that are favoured by individuals are also reflected in variations in S1 brain topography, particularly during early development. This additional analysis is reported in the also reported as supplementary materials as follows:

To further validate our correlation analysis, we conducted an additional permutation test. This time, our compensatory behaviour measure was based on the relative frequency each body part was used across all sub-tasks (e.g., if the residual arm was used in 13 out of 15 sub-tasks, the index was 0.867). Similar to our original approach, we analysed deviations in activity profiles and behavioural performance for each CLD individual relative to their peer control group. We observed very consistent results: The averaged Spearman correlation was 0.52 ($p < 0.001$) for children and 0.27 ($p = 0.003$) for adults. Moreover, children's mean correlation remained significantly larger than adults' ($t=3.12$, $p=0.002$). These findings strongly corroborate our original observations: individual behavioural adaptations are indeed reflected in variations in S1 brain topography, particularly during early development.

To further validate our key result, we conducted a Leave-One-Subject-Out (LOSO) validation. For each group, we iteratively removed one participant and then recalculated the mean correlation for the remaining individuals. This process

created a distribution of "leave-one-out" mean correlations for each group. The LOSO validation confirmed high consistency in our group-averaged correlations. For the CLD children, the mean of these LOSO correlations was 0.593 (standard deviation = 0.0156, range = 0.0571), perfectly matching their original group mean of 0.593. Similarly, for the CLD adults, the LOSO mean was 0.393 (standard deviation = 0.0215, range = 0.0714), consistent with their original mean of 0.393. The small standard deviations and narrow ranges observed within both groups indicate that our reported average correlations are robust and not disproportionately affected by any single participant.

3) Passive vs. Active Stimulation: I do wonder about the distinction between the passive stimulation used in the task, and active stimulation, where the participant performs a movement and receives sensory feedback. That said, I understand the constraints of what can be done in the scanner.

We agree with the reviewer that this issue merits careful consideration. Active stimulation is likely to provide a richer and more naturalistic input to S1, engaging not only multiple sources of afferent input (skin, deeper tissue, proprioceptors), but also efferent signals (e.g. efference signals).

To evaluate the impact of using passive stimulation in the current study, we first note that the local remapping effects we report in S1 closely replicate our earlier findings in adult congenital one-handers using active paradigms (Hahamy et al., 2017; Hahamy & Makin, 2019), as well as those of others. Importantly, Striem-Amit et al. (2018) also reported comparable remapping using passive and active stimulation, suggesting that the phenomenon is not strictly contingent on stimulation mode.

These observations also align with our previous systematic comparison of passive and active somatotopy in S1 (Sanders et al., 2023), in which, crucially, we found no significant differences in multivariate representational structure and spatial map layout between the two paradigms, using 7T neuroimaging. This work directly validates the use of passive stimulation for probing functionally relevant topography and plasticity in S1. Further converging evidence comes from a recent study in tetraplegic patients undergoing intracortical electrode implantation, where somatotopic maps derived from finger tapping (motor-driven fMRI) were found to closely correspond with those evoked by direct microstimulation of the same S1 sites (Downey et al., 2024). This provides further direct validation that fMRI responses to motor tasks engage the same topographic architecture as tactile stimulation. Third, in our face remapping study (Root et al., 2022), although we used an active motor paradigm, we included some supplementary analysis using passive stimulation to validate our methodology. Taken together, these findings indicate that remapping in S1 is robust across active and passive stimulation modes.

In combination with our response to Comment 1 — noting that behaviourally relevant plasticity is likely to shape the entire sensorimotor hierarchy — this convergence of evidence gives us confidence that the passive paradigm used here provides a valid and reliable window into S1 plasticity, particularly in paediatric populations where active paradigms are less feasible due to compliance limitations.

In the revised manuscript we bolstered our original discussion on this issue as follows:

‘We were encouraged by previous research both by ourselves and others showing minimal differences in S1 organisation and remapping between active and passive somatotopy protocols (Root et al., 2022; Sanders et al., 2023; Striem-Amit et al., 2018). Somatotopic maps derived from finger tapping (motor-driven fMRI) closely correspond with those evoked by direct microstimulation of the same S1 sites in tetraplegic patients undergoing intracortical implantation (Downey et al., 2024). These studies provide direct validation that fMRI responses to motor tasks engage the same topographic architecture as tactile stimulation.’

- 4) **Interpretation of S2 Findings: There was a significant interaction effect between age and body part in the secondary somatosensory cortex (S2), but no significant interactions with limb difference. The authors could discuss the different functional roles of S1 and S2 and how they might relate to the observed differences in remapping. If remapping is primarily localized to more primary sensory areas, what are the implications of that?**

We appreciate the reviewer’s interest in our S2 findings. Supporting this interpretation, we observed significant correlations between representational dissimilarity patterns in S1 and S2 across participants. These correlations were present across all groups, confirming that S2 carries similar broad-scale somatotopic information. However, the strength of these correlations was not modulated by limb difference, and tended to be weaker in children than in adults, consistent with developmental refinement rather than experience-dependent remapping (Nevalainen et al., 2014). This further supports the view that S2 reflects core body part distinctions but may be less sensitive to the specific plasticity effects observed in S1.

However, we encourage a cautious interpretation of these findings for conceptual and methodological reasons. Relative to S1, S2 representations are thought to be less spatially distinct (Ruben et al., 2001) and more influenced by task demands (e.g. object-related features (Reed et al., 2004), multisensory integration (Gentile et al., 2013)). Therefore, the current paradigm (abstract stimulation across individual body parts) may not have provided the ideal task to study S2 representation. Consequently, while activity levels were relatively high, the representational dissimilarity values observed in our dataset were correspondingly reduced. This inherently limits the sensitivity of our analysis and makes it more difficult to detect subtle effects such as remapping.

Second, in the absence of a dedicated individual-level functional localiser, our S2 ROI was defined using an atlas-based anatomical label constrained by group-level functional activation. This approach, while conservative and consistent with prior work, may not precisely capture inter-individual variability in S2 functional anatomy, further limiting sensitivity.

Taken together with the key result here (non-significant differences), the lack of a remapping effect in S2 should be interpreted with caution, which is why we chose to include this analysis in the Supplementary Materials rather than the main text.

If we were to speculate about the meaning of these findings, we would argue that the very dramatic remapping we see in S1 might not have functional relevance, after all. This

interpretation will also be consistent with the (relatively) far less impressive relationship between remapping and behaviour we report.

We have clarified this rationale in the revised Discussion and elaborated on these considerations in the revised supplementary section as follows

‘The impressive compensatory behavioural abilities seen in these clinical cases, such as echolocation-based navigation or Braille reading, may be attributable to learning-related recruitment of higher-level cortical computations that can be functionally re-focused on residual primary area representations. Our current study confirms this perspective and adds to it by showing a lack of clear downstream readout of the S1 remapping effects by S2 (reported in Supplementary Materials and Figure S14). Together, our findings demonstrate that deprivation effects can “passively” contribute to extensive topographic remapping, above and beyond previous accounts.’

‘This pattern of results complements, yet clearly diverges from, the robust deprivation-related remapping observed in S1. Across all groups, pair-wise representational dissimilarity values in S2 correlated significantly with those in the deprived-hand area of S1, confirming that broad somatotopic motifs are captured in our S2 ROI. While we found some evidence to support that S2 continues to refine across development (see Nevalainen et al., 2014 for related results), neither activity levels nor representational distances in S2 were modulated by limb difference. This dissociation is not unexpected given the different functional roles of the two areas. Compared with S1, S2 representations are less spatially segregated (Ruben et al., 2001) and are more susceptible to task demands, including object-centred features (Reed et al., 2004) and multisensory integration (Gentile et al., 2013). Our abstract, body-part-specific stimulation paradigm may therefore have lacked the ethological richness required to expose functional plasticity in S2. Together with the inconclusive Bayes Factor values, which do not provide sufficient evidence to confirm the null hypothesis, the absence of a clear remapping signature in S2 should be interpreted cautiously.’

- 5) **Role of Upper Face in CLD Re-Mapping: More discussion could be included regarding the weak trend of a main effect of limb difference observed in the upper face in S2. The authors could discuss whether this represents a non-specific spread of activation or a small degree of functional reorganization.**

We thank the reviewer for raising this point. We believe there may have been a misreading of the results, as we did not observe any trending effects for the upper face in S2. Instead, the relevant trend occurred in S1, where we observed a weak main effect of limb difference for the upper face (forehead), a body part not typically used for compensatory behaviour in congenital limb difference.

Although this effect did not reach conventional significance ($F_{1,317.97} = 2.99$, $p = 0.084$; $BF_{10} = 0.52$), we interpret it as tentatively consistent with **homeostatic plasticity**, in contrast to the stronger and functionally specific effects observed for compensatory effectors

(e.g., lips, tongue, feet), which likely reflect **Hebbian mechanisms**. The presence of a weak effect for a non-compensatory effector supports the notion that deprived cortex may also become sensitive to irrelevant or neighbouring inputs in an effort to maintain representational balance, as we elaborate in the Results revised Discussion.

‘Interestingly, when adding the upper face in the model, which is the lateral neighbour of the deprived hand and is not widely used for compensatory behaviour, we found a weak trend towards a main effect of limb difference ($F_{1, 317.97} = 2.99$, $p = 0.084$), which remained ambiguous following the Bayesian analysis ($BF_{10}=0.52$). While trend results should be interpreted with caution, this might indicate that the shifts observed in children might not be exclusively shaped by behaviour in earlier stage of development.’

‘As further support to this view, we find that some of the limb differences across S1 may be shared to a small extent (as expressed in trends towards significance) with the upper face, which was specifically included in our experimental paradigm as a control body part that is not relevant for compensatory behaviour. The early establishment of these dramatic remapping suggests a significant role for homeostatic plasticity processes: the deprived cortex may increase its responsiveness to surrounding or residual inputs irrespective of behavioural relevance.’

Reviewer #2 (Remarks to the Author):

This manuscript investigates the representation and plasticity of sensorimotor functions in individuals with congenital limb absence, focusing on the organization of sensorimotor cortical areas, motor intention, and effector-independent action representations. The authors combine functional MRI (fMRI), behavioral tasks, and a cohort of congenital one-handers to show that cortical hand areas are repurposed for multi-effector functions. Notably, they show that sensorimotor hand territories can support actions typically attributed to other body parts (e.g., foot or residual arm), indicating effector-independent functional representation. It's a significance research and the findings are novel and important, with significant implications for our understanding of cortical plasticity, motor intention, and the functional architecture of the sensorimotor system.

Overall, the experimental design is sound, and the data support the main conclusions. The use of multiple cohorts (including those with congenital limb absence and controls), combined with behavioral measures and neuroimaging, strengthens the validity of the claims. The manuscript is well written and generally clear. The introduction contextualizes the work within current debates on plasticity and sensorimotor organization. The discussion appropriately addresses alternative explanations.

We thank the reviewer for their positive feedback

However, several aspects require further clarification or expansion:

Clarifications/expansions

- 6) **The definition and operationalization of "effector-independent" representations could be more clearly explained and distinguished from compensatory plasticity.**

We thank the reviewer for highlighting this important conceptual distinction. We would like to clarify that we intentionally refrained from using the term “effector-independent representations” in our manuscript, precisely to avoid implying a mechanistic or functional claim that the deprived hand cortex encodes actions or goals in an abstract, effector-invariant manner. Instead, our interpretation is rooted in a plasticity-based framework, distinguishing between deprivation-driven (homeostatic) and use-dependent (Hebbian) mechanisms.

Our findings of broad remapping—including shifts of body parts that are not immediate cortical neighbours of the hand area—may indeed *appear* consistent with effector-independent coding. However, we do not claim that the hand cortex encodes functions regardless of which body part is used. Rather, we interpret these findings as evidence of remapping driven by loss of typical input (as demonstrated by our computational model), and further shaped by compensatory behaviours, particularly during early development.

- 7) **The statistical approach in some imaging analyses (e.g., ROI selection, correction for multiple comparisons) needs more detailed description, particularly in the supplementary information.**

We are happy to provide further detail on our statistical approach. We have revised the Methods section and Supplementary Materials to provide a more comprehensive

description of the imaging analyses. Specifically, we have expanded the explanation of: Our ROI selection procedures (as detailed in response to another comment); The specific group-level contrasts performed, including the rationale for each; The method used for correction for multiple comparisons, including details of the cluster-forming threshold and the cluster-level significance threshold. These clarifications pertain to the maps presented in Figure 2C, Supplementary Figures S4, and S5. We have also addressed other methodological concerns raised by the reviewers to improve the overall clarity and rigor of our reporting.

We have revised the manuscript to clarify the group-level analysis in the Methods section:

Before conducting the group-level analysis, we flipped these second-level images along the x-axis for CLD participants missing the right hand, and for control participants who were left-hand dominant. This standardised the left hemisphere as the 'deprived/non-dominant' hand hemisphere across all participants. This step was necessary, as the tactile stimulators were aligned with the missing/non-dominant side. We then conducted two types of group analysis using these second-level contrast images. First, we estimated group-level activation to identify regions engaged during stimulation of each body part for visualisation purposes (Figure 2C, S4). Second, we contrasted body part maps between the CLD and control groups to determine differential activity in response to stimulation (Figure S5). Both analyses employed a random-effect model in FSL (FLAME 1) and a cluster correction to control for multiple comparisons with the following parameters: Z threshold=2.58 (corresponds to a two-tailed p-value of approximately 0.01) and $cluster\ p\ threshold = 0.05$. The resulting thresholded maps were then projected on a standardised surface for visualisation purposes (Figure 2C, S4 and S5-Arm). Where statistical significance was not reached, uncorrected maps ($p < 0.01$) are presented (Supplementary Figure S6, except for Arm).

The behavioral and neuroimaging methodologies are appropriate and robust. The authors use a well-justified set of tasks to isolate motor intent from execution, and they include relevant control conditions. However:

- 8) **Ethical Approval and Data Management: Lines 740-752]:** The manuscript states that recruitment was carried out in accordance with the University College London and Cambridge University research ethics committees (17205/001). Could the authors please clarify whether this reference number corresponds to a single ethics application or to multiple approvals for a multi-site study? Given that data collection took place across multiple locations (BUCNI in London and CBU in Cambridge), it would be helpful to provide additional information on how ethical approval was managed across sites. Specifically:
- 9) Was this study approved under a central ethics committee as a multi-center protocol, or were separate local approvals obtained at each institution?
- 10) Were formal data sharing agreements or protocols established between the sites, particularly given the sensitive nature of data collected from children and clinical populations?

We thank the reviewer for these important clarifications and apologise for the lack of detail in the original submission. The ethics reference number (17205/001) refers to a single ethics application approved by the University College London Research Ethics Committee. This approval explicitly covered data collection at both the London (BUCNI) and Cambridge (CBU) sites, including the recruitment of paediatric participants, and permitted data storage and processing at the Cambridge site.

All data were stored and analysed exclusively on a secure, designated Cambridge University research computing cluster, in full compliance with institutional data protection protocols. These procedures adhered to the ethical guidelines and General Data Protection Regulation (GDPR) standards in place at both institutions, which maintain closely aligned governance frameworks.

As the data were never transferred externally or beyond the core research group covered by the ethics application, there was no requirement for additional data sharing agreements. The study remained fully within the scope of the approved protocol at all times.

We have revised the manuscript to clarify this point accordingly:

‘Ethics approval for this study was granted by the University College of London research ethics committees (17205/001).’

11) More transparency is needed regarding the behavioral task performance of participants with limb , particularly in terms of accuracy and timing metrics

Thank you for highlighting this important detail. In terms of timing, for each task we video coded each frame (each 40 msec) of effector use to obtain a series of effector use bouts. Several body parts were often used simultaneously, resulting in a binary ‘y/n’ coding for whether each body part was used at each frame. To calculate e.g. percentage torso use (Fig 1, Fig S1), we summed the duration of all bouts of torso use and divided that by the total duration of task time across all 15 subtasks. This is now explained in more detail in the *Behaviour indices* section:

‘to calculate the percentage of use metric, in each subtask we summed the durations of all bouts using a particular body part and expressed that as a percentage of the duration of subtask time. These percentages were then averaged across subtasks.’

Accuracy is not directly provided here, as it is not a straightforward metric, considering the task. Participants were encouraged to keep trying until they had completed a task. If a participant was becoming discouraged the experimenter gave a small, standardised prompt. If the participant could not complete the task or asked for help, the experimenter assisted them manually and encouraged them to try again in the remaining tasks. In this context, coding accuracy would not be straightforward. Further, the question of interest was how the goal was ultimately achieved using various body parts, as discussed above.

12) Lines 795-797: The authors mention that four raters evaluated the videos. Were these raters co-authors? If so, please specify their initials for transparency. Additionally, did the raters have prior experience in similar tasks? It would be helpful to provide

more detail on the rating protocol — for example, the instructions they received, who served as the primary rater, and how disagreements were resolved.

This important information is now provided in the section *Materials & Methods/Compensatory behaviour analysis/ video coding*. The primary rater, Laura Bird (L.B.), was also an author. She has extensive ~10-years of experience of assessing typical and atypical children’s performance across a range of standardised and non-standardised tasks. Dorothy Cowie (D.C.) used her own experience of video coding to supervise L.B.: we also took advice from other colleagues on best practice. L.B., supervised by D.C., developed a very full written protocol for video coding. This specified how to deal with common issues identified by L.B.’s initial viewing of the videos: for example, what to do if an object is dropped, or when to cease coding in the instance of inattention. All other raters, named in the acknowledgements, were trained on this using three sample videos for checking and discussion. Raters demonstrated almost perfect agreement on which body part was used (Cohen's kappa = 0.972) and excellent reliability for body part usage duration (ICC = 0.989). As agreement rates were almost indistinguishable, coding was left as per the secondary raters’ rating. Any small disagreements concerned sections of on average <500ms within a 5-minute video.

We have added the information about the raters’ agreement in the Methods section of the manuscript:

Raters demonstrated almost perfect agreement on body part used (Cohen's kappa = 0.972) and excellent reliability for body part usage duration (ICC = 0.989). As agreement rates were almost indistinguishable, coding was left as per the secondary raters’ rating.

13) Lines 808-810: The computation of component C2 is not clearly described. It is also unclear what is meant by “actually engage.” Please clarify how this component was derived and what behavioral or analytical criteria were used

We apologise for not making this point clear. We have made the necessary adjustments to the Methods section and added a new section in the Supplementary Materials to report the equations used to calculate the two components and a related description.

‘The co-use measure used for our behavioural analysis (Figure 1A) is an estimated index that reflects both the number of body parts used and how consistently they were used, with higher values indicating more usage or more consistent usage. The function is the sum of two components: C1 + C2. Component C1 (equation 1) counts the number of body parts that were used during the observation period (an integer value). It is calculated by counting any body part with a usage proportion greater than zero.

$$C1 = \sum_{i=1}^n U_i \quad (1)$$

U_i is a binary indicator variable for the i^{th} body part which is equal to one if the i^{th} body part was used ($p_i > 0$) or zero if it was not used ($p_i = 0$), and n is the total number

of body parts considered. Let's indicate with p_i the proportion of use of the i^{th} body part. Component C2 (equation 2) adjusts the index based on the consistency of usage across the used body parts. It is computed as the average of the difference between each used body part and its proportion.

$$C2 = \frac{\sum_{i=1}^n (U_i - p_i)}{\sum_{i=1}^n U_i} \quad (2)$$

The value of C2 is either **zero** or **negative**. It's zero if all body parts were used for the entire task duration. However, it becomes negative if at least one body part was used for less than the total task time. Essentially, C2 accounts for differences in how long each body part was engaged.

A smaller penalty from C2 (meaning a value closer to zero) results in a higher overall co-use index. This happens when the body parts used have similar or high proportions of engagement time. Conversely, if body part usage is highly uneven—for example, one body part is used extensively while others are barely involved—C2 will significantly decrease the co-use index. This reflects a less cooperative or balanced usage of body parts.

The final equation to estimate the co-use index is indicated in equation 3:

$$\text{co-use} = C1 + C2 \quad (3)$$

- 14) The use of an individualized or group ROI for fMRI analyses should be justified more clearly.

We are happy to provide further details regarding our ROI selection strategy and apologies if this wasn't clear. Our analysis investigated both global and local effects of age and deprivation.

For global effects, we employed a group ROI, defined by Glasser atlas areas 1 and 2 (Glasser et al., 2016) in MNI space. Since individual functional data were also in MNI space, this approach ensured comprehensive coverage of the post-central gyrus across all participants, as we verified visually. This was further validated by our thumb peak analysis (Figure 2D, 2E and S5), which confirmed that all individual peaks were included within the group mask, suggesting that we were using data from the primary somatosensory region. This method provided an unbiased and efficient way of selecting data, whereas individual anatomical ROIs would have been more labor-intensive and susceptible to subjective variability.

For local effects on the hand area, we used individual ROIs. This was motivated by the known anatomical variability of the hand area around the pre-central gyrus knob (Yousry et al., 1997). By leveraging our thumb data, we were able to define the hand area in the deprived/non-dominant hemisphere with greater precision and objectivity, based on the thumb peak in the intact/dominant hemisphere.

Therefore, we believe this combination of group ROIs for global analyses and individual ROIs for local analyses represents the optimal strategy for achieving unbiased and precise ROI selection.

To make our choices clearer to the readers, we added these sentences in the “ROIs definition” section:

‘As we were interested in studying both global (i.e., along S1) and local (i.e., hand territory) changes due to age and/or deprivation, we defined two different types of ROIs within S1. Specifically, we defined a group ROI for the global analysis and individual ROIs for the local analysis. The group ROI, derived from Glasser atlas (Glasser et al., 2016) areas 1 and 2 in MNI space, provided an unbiased and efficient means of capturing global changes across the somatosensory cortex. In contrast, the individual ROIs, defined based on thumb peak locations, allowed for more accurate and sensitive detection of local changes within the hand territory, given its known anatomical variability (Yousry et al., 1997).’

- 15) Information on fMRI preprocessing, movement exclusion thresholds, and participant level exclusion criteria should be expanded.

We have carefully considered the reviewer's comment regarding our fMRI preprocessing and have taken proactive steps to enhance the level of detail. Specifically, we have incorporated an explanation of the fieldmap data preparation and its use in correcting inhomogeneity distortions:

‘Signal distortions caused by B0 field inhomogeneities were corrected using B0 fieldmaps. The fieldmap magnitude and phase images were processed with FSL's *fsL_prepare_fieldmap*. No slice-timing correction was applied because the relatively short TR (1450 ms).’

provided a clearer description of our volume outlier detection:

The 0.9 mm threshold was selected based on previous methodological work by Siegel et al. (2014). Across three runs for children (totalling 99 runs for 33 children), the vast majority of the data (85% of all runs) contained 10 or fewer identified volume outliers which is 4.63% or less of the total volumes with stimulation (around 215 volumes out of 277 total volumes per run) were flagged as outliers. This low proportion of outlier data suggests that children generally remained still during the stimulation.

and updated the description of data exclusion criteria for the peak analysis:

Note that for two participants, a clear peak activity (z-statistic exceeding zero, indicating a statistically significant positive activation) was not identifiable for the torso, and therefore these two contrasts (out of 462 total contrasts, i.e., 7 x 66 participants) were excluded from further analysis. Additionally, double-peaked activity patterns were observed in a subset of participants (n=20) and contrasts.

- 16) **Lines 975–978: The authors mention using a framewise displacement threshold of 0.9 mm to flag volumes for exclusion. It would be helpful to clarify the rationale for selecting this specific threshold, as it can significantly affect data retention. A reference would support this choice, or alternatively, the authors could justify its appropriateness for this particular dataset. Additionally, in line 978, the text states “10” based on the context, this seems likely to be a typo. Please verify and correct as needed.**

The 0.9mm is a default threshold for motion scrubbing, and is based on previous work by Siegel et al. (2014), which systematically investigated optimal thresholding strategies for task-based fMRI data in an age cohort comparable to our study. We routinely use this threshold in our lab in fMRI publications.

‘The 0.9 mm threshold was selected based on previous methodological work by Siegel et al. (2014).’

The “10” in line 978 indicates that 85% of all runs across the 33 children (99 runs in totals) contained 10 or fewer outliers. The value 10 represents the 4.63% of the total stimulated volumes (around 215 volumes out of 277 total volumes per run), as reported in the manuscript. We have made this point clearer in the new version of the manuscript:

‘Across three runs for children (totalling 99 runs for 33 children), the vast majority of the data (85% of all runs) contained 10 or fewer identified volume outliers which is 4.63% or less of the total volumes with stimulation (around 215 volumes out of 277 total volumes per run) were flagged as outliers. This low proportion of outlier data suggests that children generally remained still during the stimulation.’

Minor:

- 17) **Line 115: Supplementary Table S1 no Supplementary Table 1**

We thank the reviewer for pointing out this error. We have made the necessary correction in the manuscript.

- 18) **Figure S1/S2: is quite complex and not easy to interpret, particularly due to the arrangement of different body parts across the subplots. To improve readability, I would suggest adding subtle vertical divider lines between the major body regions (e.g., feet, legs, torso, etc.), which could help the reader more easily distinguish the corresponding sections across subplots.**

We thank the reviewer for the suggestion, we agree the readability could be improved. Following your recommendation, we have added subtle vertical dividers between the body regions in both figures. We hope this modification enhances clarity.

- 19) **Why did the authors choose to report the standard error of the mean in the figures instead of the standard deviation?**

We chose to present the standard error of the mean (SEM) in our figures to indicate the precision of the sample mean in estimating the population mean. This has been our choice in all lab-published manuscripts, and for consistency we prefer to keep this practice. Additionally, we have included individual data points in the figures to directly represent the variability within the sample.

- 20) **Figure 1B: I appreciate the illustrative representation showing the co-use of different body parts (below images). However, I found the image selected for the Arm co-use condition a bit confusing. To me, it looks more like a depiction of Hand co-use, as the participant seems to be using the other hand to perform the task. Could you please clarify this?**

Our intention in Figure 1B was to illustrate the **functional contribution** of different body parts to everyday behaviours. All our tasks are typically performed bimanually, and adults with two hands in our study consistently used both upper limbs (as indicated by our co-use index). In contrast, individuals with CLD naturally rely on their intact hand for most tasks—what differs is **how other body parts contribute to supporting or enabling the action**.

In the Arm example, the participant is using their residual arm to carefully balance and support the object, while the intact hand performs fine-grained manipulation. That supportive role is central to our behavioural framing.

To avoid misinterpretation, we have revised the figure legend to clarify this intention:

‘Images are provided to illustrate functional contributions by various body parts to aid the intact hand during task completion. Please note that multiple body-parts might be simultaneously used, e.g. torso, residual arm and hand.’

- 21) **Line 141 ..Supplementary Tables S2 and specify the specific tables**

We thank the reviewer for pointing out this error. We have made the necessary correction in the manuscript.

- 22) **Figure 2B: To support the text describing the stimulation locations for the different groups (lines 867–878), I would suggest expanding Figure 2B and adding schematics for the CLD and control participants. Include the placement locations for both groups, label each location (e.g., “1”, “2”, “3”, etc.) clearly within the figure, and indicate “non-dominant” for control participants directly in the figure. These additions would make the figure more informative and help readers easily identify the stimulation sites across groups directly from the figure.**

We thank the reviewer for their useful suggestions. We carefully considered how to improve Figure 2B, including adding schematics for control participants. However, this made the figure overly complex, so we retained the original configuration. We believe showing schematics only for the CLD group is sufficiently clear, as the locations are identical in the control group. To enhance clarity, as suggested, we added a numerical label to each location, corresponding to the enumeration in the Methods section and now referenced in the caption. Note that the enumeration of the body parts was updated to be consistent with the order used throughout the manuscript. We also made aesthetic adjustments to improve the figure's balance and clarity.

We report here the updated figure:

REDACTED

- 23) **Line 260:** The text refers to Figure 3A as showing a functional ROI from an exemplar participant. However, the figure legend describes group-averaged Z values and does not clearly indicate whether the ROI brain image represents a single participant or a group-level or template example. For clarity, I suggest the authors explicitly state whether the brain plot is from an individual participant or a standard reference brain in the legend

We thank the reviewer for raising this point and apologies for not making the caption clearer. The Figure (Panel A, B, and C) generally describes group analyses, while the brain plot shows a participant's ROI resampled onto a standard template surface for visualisation purposes. We have made the necessary changes in the revised manuscript:

‘The top-right panel displays a participant's ROI after it has been resampled onto a standard template brain for visualisation purposes.’

- 24) **Line 548: Discussion :** It might be more reader-friendly to present the main results or main findings in the first paragraph, so that readers have a clear sense of the overall direction of the discussion

We have removed the introductory paragraph from our original discussion, to streamline the presentation of our key results.

- 25) **Line 548: Discussion:** When citing the results that support each discussion point, including figure or table references would improve readability

We appreciate the reviewer's suggestion to include figure or table references within the Discussion to enhance the overlap with the Results section. However, we have opted to minimise figure citations in this section in order to maintain the interpretative flow and clarity of the broader narrative, in line with standard conventions for the structure of scientific discussions.

Throughout the manuscript, we have provided detailed citations to figures and tables in the Results section, where the empirical findings are systematically presented and contextualised. We believe that duplicating these references in the Discussion risks overburdening the text and potentially detracting from the broader conceptual interpretation.

That said, we have carefully reviewed the section and ensured that any less obvious references to key results are now clearly linked back to the appropriate earlier descriptions. We hope this strikes a balance between clarity and readability.

- 26) Lines 718-719: Could the authors clarify whether a power analysis was conducted to determine the sample size? If not, please explain how the number of participants (16 CLD children and 16 CLD adults) was established and whether it was based on prior studies, feasibility, or other considerations**

We apologise for not reporting this information. We conducted a power calculation while applying for funding to conduct the study, and our target sample size was informed by prior research. Specifically, we conducted a power analysis based on Hahamy & Makin (2019), which examined similar univariate effects of interest in adults with congenital hand loss compared to controls across three independent datasets. This analysis indicated that a sample size of approximately 9 participants per group was sufficient to detect the effects observed in that adult population. Given the potential for higher noise levels in fMRI data from children and that we used passive stimulation, our initial aim was to recruit a larger sample size of 15-20 participants per group (children and adults). Ultimately, we were able to recruit 16 participants per group (16 CLD children and 16 CLD adults). To achieve this sample, we had to maintain recruitment and data collection over the course of 3.5 years. This recruitment number also reflects the significant challenges associated with studying children with congenital limb differences in a specific age bracket. This is a relatively rare population, and our recruitment efforts were inherently limited by the number of eligible individuals within the desired age range.

- 27) Line 724: was the CLD participant excluded a child or adult participant?**

The excluded CLD participant was a child. We have made the necessary change in the manuscript: "CLD participant" was changed to "CLD child".

- 28) Line 865: The reference to Figure 1B appears to be incorrect; it should be Figure 2B.**

We thank the reviewer for pointing out this error. We have made the necessary correction in the manuscript: "Figure 1B" was changed to "Figure 2B".

- 29) Line 992: While MNI space is commonly understood within the neuroimaging community, I would recommend spelling out the acronym at its first mention, to ensure clarity for readers who may not be experts in the field. Same for FEAT.**

We agree with the reviewer that these acronyms (MNI and FEAT) should be spelt out to ensure clarity. We have done that at the first occurrence of these two terms.

30) Line 1002: Figure 2B instead of general Figure 2

We made the necessary changes in the manuscript.

31) Line 1026 and 1043: please specify which panel of Figure 2 you are referring

We made the necessary changes in the manuscript.

32) Line 1095 you repeated two times lower face

“lower face” was changed to “upper face”.

33) Lines 1102–1106: the authors report summary values such as “(2/469; 7 × all participants)” and later “(20/467; 7 × all participants minus 2)” in reference to cases where peak activity was not identifiable and instances of double-peaked activation patterns, While the general meaning can be followed, I recommend clarifying these expressions more explicitly to help readers understand how these numbers were derived.

In particular, if I understand correctly, the number of participants included in the fMRI analyses was 15 CLD children, 16 CLD adults, 20 control children, and 15 control adults (as stated in lines 737–740), which would yield a total of 66 participants. Therefore, 7 × all participants should result in 462, not 469. Could the authors please confirm whether this discrepancy is due to a typo, inclusion of additional data, or another factor?

The reviewer is correct, it was a typo. Thanks for noticing it. We realised this section was a bit confusing as it was written, so we changed it to make it clearer.

‘Note that for two participants, a clear peak activity (z-statistic exceeding zero, indicating a statistically significant positive activation) was not identifiable for the torso, and therefore these two contrasts (out of 462 total contrasts, i.e., 7 x 66 participants) were excluded from further analysis. Additionally, double-peaked activity patterns were observed in a subset of participants (n=20) and contrasts. In these cases, the peak closer to the group average for that specific contrast, which typically aligned with the expected location based on the classical homunculus model, was chosen manually.’

Reviewer #3 (Remarks to the Author):

Summary: In this article, the authors introduce several key aspects that fill significant gaps in our current understanding of the differences between control subjects and CLD subjects' somatotopic maps. The authors specifically address differences between children and adults with CLD. Interestingly, they find similarities between the children and the adult CLD subjects that indicate somatotopic map changes occur before age 5 and that these changes are maintained into adulthood. These results are not particularly surprising when viewed in the context of animal research on changes in somatotopy. For these changes in the homunculus, they formulated a biologically inspired model that is very simple in construction. Even with its simplicity, they show that it can reproduce many of the differences seen between the CLD subjects and the controls simply with the inclusion of homeostatic plasticity and several justifiable assumptions, such as differences in the distribution of sensory input to the different body regions of CLD vs. CTR subjects. The use of more ecologically valid Semi-Naturalistic movements is a nice addition. However, some of these movements, or sub-tasks, are likely new to the younger subjects compared to the adults, for whom unscrewing something is much more familiar. Overall, the paper is significant and insightful, bringing new insights forward in a clear, easy-to-understand article. Some issues are raised below; most are meant to increase readability.

We thank the reviewer for their support.

Experimental Section:

- 34) **1. The co-use index used has some assumptions built in. One is that the children's apparent variability of body part use is due to some general higher variability in motor planning or a higher expletory rate. Is it possible that the children get fatigued or feel pain when making some of these new movements while interacting with the objects? In essence, how would your measure differentiate subjects with two solutions for completing a sub-task that they switch between rather than something more akin to a higher exploration rate in the motor output space?**

We thank the reviewer for raising these nuanced considerations. While we agree that movement variability can stem from multiple sources, including planning, strategy switching, or exploration, we would like to clarify that the tasks assessed in this study were all familiar daily-life behaviours (e.g., buttons, zippers), rather than novel or physically demanding ones. As such, fatigue or pain are unlikely to be meaningful contributors, especially within the short, semi-ecological testing context we used. Most tasks were completed within 90 seconds; L.B. & M.S., who administered the tasks, report no instances of a participant reporting any pain. Qualitatively, children who switched strategies tended to do so based on their current strategy being ineffectual.

Our primary aim was not to interpret co-use variability in terms of action planning mechanisms per se, but rather to assess the range of body-part involvement — a pragmatic proxy for behavioural flexibility and support strategies in action control.

Regarding the distinction between motor exploration and the use of multiple stable strategies: we acknowledge that this remains a conceptual challenge in naturalistic tasks. This has been addressed in our prior work using a virtual paradigm (Allen et al., 2024 Psychonomic Bulletin & Review), where we showed that children exhibit more versatile

exploration of physical solutions, whereas congenital limb difference — whether in childhood or adulthood — influences their cognitive style for solution exploration.

Here, we note that such switching — whether between fixed strategies or due to exploration — still reflects meaningful variation in support behaviour, which is precisely what our co-use index aims to quantify. Our design does not assume that variability must reflect randomness or inefficiency, and we agree that a much larger sample size of children with an upper limb difference would be needed to tease apart these nuances across naturalistic task demands.

We have added a brief clarification to the Results to reflect this interpretation:

‘While our analysis doesn’t allow us to ascertain whether this also reflects greater exploration or task switching, our finding indicates that adaptive behaviour at the tested age range (5-7 years) still has not matured.’

35) 2. In Figure 1, the picture for Torso seems to be of arm use. The picture for the arm shows the use of the elbow joint to hold an item. Are these terms used regularly in this research area for these body parts?

We apologise for the confusion. We acknowledge that the images selected for the torso and arm in Figure 1B could be interpreted as overlapping in terms of visible body part involvement. Our intention was not to indicate exclusive use of a single body part, but rather to highlight the primary supportive role played by a given body region within each task context.

Specifically, in the "Torso" example, although the arm is present in the image, the torso is actively used to stabilise the object against the body, which is a common compensatory strategy. Similarly, the "Arm" example focuses on the functional contribution of the residual arm to support the object, while the intact hand carries out the manipulation. This is consistent with how such body parts are typically used for compensatory behaviour in populations with limb differences.

We have also revised the figure legend to clarify that these images are intended to illustrate functional contribution rather than anatomical isolation:

‘Images are provided to illustrate functional contributions by various body parts to aid the intact hand during task completion. Please note that multiple body-part might be simultaneously used, e.g. torso, residual arm and hand.’

We also further clarify this in the revised Results section:

‘For example, while CTR participants tend to use both hands to open a tin box, CLD individuals might stabilise the tin with their legs, torso, residual arm (or any combination of these body parts), while lifting the lid with their intact hand.’

Finally, we have also updated the representative images for “Arm” and “Torso” in Figure 1B with examples that hopefully highlight better the supportive roles of these two body parts.

36) 3. Figure 1, Please use the term co-use for subplot A's y-axis, or change the name of the measure you use to co-use complexity.

We agree with the reviewer, the plot axis should be named “co-use”. We have made this change in the figure.

37) 4. It might be helpful to use different colors for the children vs. the adult CLDs, but it could just be this review's eyes that had difficulties seeing the differences in the line plots.

We apologise for this, our colour choice was designed to highlight the focus on the CLDs groups, relative to controls, and to provide an intuitive distinction between older participants (darker colour) and younger participants. Given we are using an array of colours in Figures 4 and 6 to indicate the different body parts, our colour options were relatively restricted.

38) 5. Figure 2.c, It might be helpful to put a simple landmark in the three sub-plots on the right-hand side to allow for easy comparisons between Foot, Leg, and Torso, as these are all within the one box on the main associated image.

We thank the reviewer for this comment and to facilitate comparison, we introduced a common landmark (an arrow) and also made the leg contours visible in the three sub-plots.

REDACTED

39) 6. Figure 5, please indicate what the x-axis is for the reader.

We apologise if the description of the x-axis was unclear. The x-axis values represent medial-to-lateral coordinates, a convention applied to all similar figures and previously stated in the caption of Figure 2. To enhance clarity and ensure consistency, we have now added the label "Medial to Lateral" directly to Figure 5 and included the sentence following in the captions of both Figures 4 and 5:

‘The x-axis values indicate MNI coordinates (medial to lateral).’

40) 7. As divisive normalization is a canonical cortical computation known to occur in sensory areas, it would be nice if the authors had any insights on how such normalization mechanisms might show up in the context of CLD and how such neuro-dynamic mechanisms might muddy the waters of the clean model and interpretation the authors would like to champion.

We thank the reviewer for this comment, as it prompted us to investigate the potentially overlooked connection between divisive normalisation and our synaptic scaling model. In short, we see divisive normalization as a potential network-level mechanism that contributes to the homeostatic synaptic scaling property (Mayzel & Schneidman, 2024).

Divisive normalisation is a well-established computational model describing how a neuron's response to a stimulus is modulated, or normalised, by the combined activity of a population of other neurons (Aqil et al., 2021; Carandini & Heeger, 2012; Heeger, 1992).

In essence, a neuron's activity is scaled down by the summed activity of a related neuronal population (Carandini & Heeger, 2012). This operation is proposed as a fundamental mechanism in various sensory regions, including visual (Burg et al., 2021; Carandini & Heeger, 2012) and somatosensory (Arbuckle et al., 2022; Brouwer et al., 2015) areas. In other words, a neuron's response is not absolute but is relative to the activity of its interconnected neighbours.

Synaptic scaling is a form of homeostatic plasticity that ensures stable activity levels within neural circuits, even when faced with disruptions such as the sensory input loss experienced by individuals with congenital limb differences. To counteract this reduction in neuronal firing rates, synaptic weights are uniformly increased by a multiplicative factor (Keck et al., 2017).

The concept of divisive normalization, initially suggested for neuronal responses (Heeger, 1992), could in principle be extended to synaptic scaling to implement a form of divisive normalization applied to synaptic weights within a specific neural circuit (Mayzel & Schneidman, 2024). Therefore, in scenarios where a neuron might become underactive, such as due to a loss of its original input, homeostatic synaptic normalization (Mayzel & Schneidman, 2024) can contribute to restoring or maintaining neuronal activity by helping to regulate the neuron's firing rate through the adjustment of overall incoming synaptic strengths. However, this interpretation should be cautioned, since our model operates on a different temporal scale than DN. It's quite common for models focusing on slower, long-term processes to not explicitly include faster, transient dynamics. For example, when modelling climate change over decades, we don't necessarily model individual weather events like daily temperature fluctuations — even though those short-term events contribute to the overall climate. Similarly, it's possible that DN contributes to synaptic scaling, alongside other mechanisms operating on faster timescales.

In the revised manuscript, we highlighted the potential role of divisive normalisation in our Discussion:

‘The primary function of homeostatic plasticity is to provide stabilising mechanisms that regulate changes in the system, preventing activity levels from becoming excessively high or low (Keck et al., 2017). This can be achieved by network-level computational mechanisms such as divisive normalization (Mayzel & Schneidman, 2024), where the neuron’s response is modulated by the combined activity of a relevant population (Aqil et al., 2021; Carandini & Heeger, 2012).

Electrophysiological research suggests that local remapping is primarily driven by the unmasking of silent inputs through disinhibition of homeostatic plasticity mechanisms (Gainey & Feldman, 2017). Through our computational model, in the current study we were able to provide a proof of concept that a simple plasticity rule predicts a much more global remapping pattern, causing topographic shifts across the entire homunculus.’

- 41) **8. Would it be helpful for the reader if you drop the term Hand loss, as it implies there was a hand and that it was lost, which would be a rather different situation than the hand never having been formed or not formed completely, as the authors are fully aware.**

We thank the reviewer for this sensitive and important observation. We fully agree that the term “hand loss” can be misleading in the context of congenital limb difference, where a hand was not lost but rather not formed during development.

We have now revised the main text to remove this phrasing and replaced it with: ‘Congenital malformation of the upper-limb’ [Abstract] and ‘born with a missing hand due to an upper limb developmental malformation’ [Discussion].

Modeling:

- 42) **1. The proposed model is elegant in its simplicity. This model could be formulated as a modified Radial Basis Network, which has previously been used in sensorimotor studies. One would simply need to use as input the sparse set of body positions rather than a continuous input space, include a post-activation gain, which is known to exist in biological systems, which would act as the synaptic scaling mechanism, and add the offset included in your work. This may or may not help make your model clearer for comparison with other RBF network publications on sensorimotor map changes. In addition, the simple gain formulation would allow relation to gain-field work in the sensorimotor literature. At least the authors could give the parameters of the function used to determine the initial weight distributions in the model they propose. This would allow others to replicate the results.**

We thank the reviewer for pointing this out. We agree that framing our model as an extended Radial Basis Function (RBF) may help in connecting our work more directly to the broader literature on sensorimotor map changes and gain-field mechanisms. However, due to space constraints and the focus of the current manuscript, we prefer to keep only our current, and arguably simplest, model of homeostatic plasticity. Nevertheless, we find this idea inspiring and will consider exploring it in future work.

As suggested, we have added a new subsection in the Supplementary Materials (“Setting of Initial Connection Weights”) where we clearly describe all parameters used to determine the initial weight distributions. We hope this will facilitate replication and further development of our approach.

We included a reference to the new subsection in the Methods section of the main manuscript.

‘The initial weights are represented by bell-shaped curves, with their peak positions uniformly distributed across the S1 homunculus strip and in line with the typical topography of this region (please see the supplementary section “Setting of Initial Connection Weights” for a detailed description of how these weights are calculated).’

- 43) **2. Many studies have examined different types of neural plasticity and changes in the somatosensory cortex due to stroke, amputation, nerve block, etc. It is unclear from the literature that has been cited if the authors are fully aware of this literature. Perhaps adding some of this background and results from rodents and primates would help round out the story being presented. That is, make clear the models that are available to ask the types of questions addressed in this work that would include**

more than just homeostatic mechanisms and are more biologically realistic.
(optional)

We thank the reviewer for suggesting the inclusion of additional literature on plasticity from stroke, amputation, nerve block, and animal models. We fully agree that this is a rich and important body of work, and we were keen to integrate it more fully into our revised manuscript. However, given the breadth of this literature and the word count limitations, we sought clarification via the editor to ensure we could appropriately prioritise any references (or specific models) the reviewer had in mind. As no specific works were suggested within the revision window, we have opted not to modify this section at this stage. We hope the current framing, based on recent reviews (including our own), and the additional references concerning plasticity mechanisms, adequately captures the broader context. We would be happy to revisit this in future iterations if further input becomes available.

- 44) **3. It seems that the results obtained would be expected given the initial weight distributions, touch probabilities, and coupling of the arm/palm/fingers. It is great to show that this sensible pattern of inputs and a simple Homeostatic Plasticity rule (normalization) give results like those seen in the subjects, but it does seem obvious this model abstraction would give these results.**

While we agree that the model is intuitive, it is challenging to immediately predict the full scope of local deprivation, e.g. the centre of gravity shifts and increased activity for body parts well beyond the hand area. Moreover, the full pattern of our results across the entire S1 have not been predicted by previous research. As such, we find the model instructive, especially for readers that do not have an intimate understanding of such models.

- 45) **4. The model doesn't include Hebbian plasticity, which leaves the modeling short of describing or addressing the individual differences in body part use compared to the mean for the given groups and how this is correlated with differences in somatotopic responses shown in the human subjects' data.**

We thank the reviewer for this important observation. We fully agree that the absence of Hebbian plasticity in our model is a limitation, and as such, the model cannot account for all aspects of the individual variability observed in the empirical data. For example, behavioural differences within the empirically observed range do not generate the critical activity differences between groups; rather, they tend to shrink them. Only very large departures from baseline probabilities disrupt the model's ability to reproduce the data. This is now acknowledged in the Discussion as a limitation.

Nevertheless, we believe that our model as is provides a significant contribution toward understanding the fundamental mechanisms underlying the plasticity following sensory deprivation, and we see it as a foundation for future extensions that could incorporate other plasticity mechanisms such as Hebbian learning.

'Our rudimentary computational model did not include a Hebbian plasticity mechanism, and as such it was not equipped to capture the likely more nuanced relationship between use- and deprivation-related plasticity. Together, our study highlights the crucial role of homeostatic plasticity in shaping the developing homunculus, through both deprivation and compensatory behaviours.'

46) **5. As the model seems to be a large part of the paper, it might make sense to put more of the model results in the main section?**

We agree that the modelling component plays a central role in the paper, and we appreciate the suggestion to foreground it more prominently. In the revised manuscript, we have provided extensive additional details of the model results in the Supplementary Materials, including a description of the initial setting of the model's connection weights (as suggested by the reviewer in comment 42), a parameter sensitivity analysis, the potential role of behaviour, how robust the model is against variations in parameter choices.

However, due to the word limit for the main text, we are unable to include more extensive model results without displacing other key empirical findings or theoretical framing. We have therefore aimed to strike a balance by highlighting the primary results of the model in the main section—specifically those most critical to our theoretical claims—while guiding readers to the Supplementary Materials for a complete account.

If the editor feels that including more model results in the main text would improve the clarity or impact of the paper, we would of course be happy to explore how best to accommodate this within the format constraints.

47) **6. It seems you are using weight matrix indexing that is the opposite of the norm where it is usually $W_{i,j}$ where i is the neuron index that the input j is going to that is $W_{to,from}$, where you are using $W_{from,to}$. This may confuse some readers.**

We thank the reviewer for pointing out this potential source of confusion. We have corrected the indexing of the weight matrices to follow the standard convention, as suggested.

48) **7. In the methods section it may be helpful to make the indexing more explicit as well, such as the summation from $j = 1:10$ as there are 10 body parts.**

We thank the reviewer for this helpful suggestion. We have updated the Materials and Methods section to make the indexing more explicit, including specifying the summation limits.

49) **8. Line 1205, where did the 15% come from and the % for the other body regions?**

Apologies if this was not clearly stated. We have added a new subsection to clarify this point in detail. As there is no available empirical data to estimate stimulation frequencies for individual body parts across daily activities, we adopted a simplified modelling approach to ensure transparency and interpretability of the model.

In this approach, all body parts were assigned the same baseline stimulation probability, except for the hand, which was given a higher probability based on the assumption that hands are more frequently engaged in daily tasks.

The value of 15% for the hand (compared to 5% for other body parts) approximates the empirical data well. However, we tested a range of hand stimulation probabilities (0.05, 0.15, and 0.25), and observed that in this range, the model's behaviour is highly robust with respect to the key Observations.

This is now described in the newly added subsection *Justification of Stimulation Probability Settings* in Supplementary Materials, with corresponding visualizations in Figure S15.

‘Here, we systematically analyse how varying the relative stimulation frequency of the hand (i.e., palm and fingers or the corresponding nerves) influences the model’s ability to satisfy key Observations.

Since no comprehensive empirical data exist on estimated stimulation frequencies of individual body parts across all daily activities, we adopted a parsimonious approach: all body parts share the same probability of stimulation, except for the hand, which was assigned a higher probability, due to its role in tool and object manipulation.

When the ratio of hand to other body parts’ stimulation probabilities is fixed, the specific probability values themselves are less critical. This is because the multiplication of probabilities merely scales the activations after homeostasis; therefore, it does not influence how the model satisfies the key Observations. Such scaling can be counterbalanced by adjusting the homeostatic Mean Neural Activity. For example, doubling all stimulation probabilities yields equivalent activations if the homeostatic Mean Neural Activity of all neurons is also doubled.

To explore the effect of hand stimulation probability, we performed an analysis where the hand was stimulated 1, 3, and 5 times more frequently than other body parts. Specifically, the tested hand probabilities were 0.05 (equal to other body parts, Figure S15A), 0.15 (three times higher, used in the main model, Figure S15B), and 0.25 (five times higher, Figure S15C). The results indicate that lower hand stimulation probabilities lead to an increase in the arm activation peak, whereas higher hand stimulation probabilities result in a decrease of the arm activation peak. For a hand probability of 0.05, only Observation 2 was violated for the foot, while all other Observations remained preserved.

Overall, these findings indicate that the model is highly robust to reasonable changes in hand-stimulation probability. A value of 0.15 was selected for the main model because, based on visual comparison, the relative size of the CLD Arm peak (compared to other peaks) most closely resembled the corresponding relative peak sizes observed in empirical data.’

50) **9. Figure 6, it would be nice to see direct comparisons between the model and the mean of the subjects’ data. Along these lines have you tried optimizing the standard deviation of your gaussian weights? The real data seems to show more localized receptive fields as compared to the model, which would seem to be easily rectified?**

Thank you for this valuable suggestion. We now explored the effect of varying the standard deviation (σ) of the Gaussian weights to produce more localized receptive fields. The figure below illustrates this comparison across different σ values.

In our main model, we initially used $\sigma = 5$ (see panel A). We then tested a narrower configuration with $\sigma = 4$ (panel B), applied only to Gaussians corresponding to body parts excluding the hand and fingers (or their associated nerves). This resulted in more localized receptive fields, as expected from the narrower basis functions. Finally, in panel C, we further reduced the width for those same body parts to $\sigma = 3$.

Even with $\sigma = 3$, the core qualitative properties (Observations 1–4) remained intact. However, we observed a shift in the peak positions for the foot and lower face toward the centre of the strip, as well as a reduced distance between the CLD and CTR peak centres for the arm (compare subfigures A and C). These changes appeared to reduce the visual similarity to the empirical data in these specific aspects.

We therefore newly selected $\sigma = 4$ as a compromise. This setting produced more localized receptive fields while limiting the more pronounced deviations observed with $\sigma = 3$, which were less consistent with the empirical data in the specific aspects mentioned above. The following results were regenerated: Figure 6D, Supplementary Materials: Role of behaviour for homunculus adaptation, Figure S9, S10, S11, S12, S13, Table S4, Table S5. Please note that this modification resulted in minor changes to the article: In the section *Role of behaviour for homunculus adaptation*, we adjusted the leg stimulation increase to elicit comparable behaviour as with the previous setting of sigma. Moreover, we updated the no longer valid sentence “Additionally, for CLD participants, there were slight shifts in the activation peaks for torso and upper face toward the centre of the homunculus strip, consistent with Observation 2” to “Additionally, for CLD participants, there were slight shifts in the activation peaks for arm (lateral direction) and lower face (medial direction), consistent with Observations 1 and 2, respectively.” in the same subsection.

We chose not to implement full numerical optimization of the parameters or extend the model, as our goal was to preserve its conceptual clarity and simplicity. Achieving a more accurate quantitative fit would require modifications—such as supporting negative activity values (present in the empirical data) and likely adopting a more complex configuration of Gaussians—which would introduce additional assumptions and deviate from our guiding principle of parsimony. The model is thus intended primarily for qualitative comparison with empirical data, rather than precise data fitting.

- 51) **10. Figure S9, please indicate the parameters of the assumed gaussian used for the normal receptive fields and the modified ones for the nerves.**

We thank the reviewer for this helpful comment. In response, we have added a new subsection ("Setting of Initial Connection Weights" in the Supplementary Materials) where we clearly describe all parameters used to determine the initial weight distributions, including those for the initial weights shown in Figure S9.

- 52) **11. Figure S12, same as above for S9.**

We kindly refer the reviewer to our response to the previous point.

- 53) **The code: "Activations should be replaced by weight in the figure." perhaps take out such notes.**

The code has now been modified and extra care has been taken to avoid unnecessary notes, we are grateful to the reviewer for pointing this out.

- 54) **There are grammatical issues in the text that you might want to fix = run spell check etc.**

We deeply apologise for this. We have conducted a thorough review using the spell check tool in Microsoft Word, as suggested, and have corrected the identified errors. We have also carefully proofread the text to address any remaining grammatical inconsistencies.

Reviewer #3 (Remarks on code availability):

- 55) **The code does not state who the author of it is. There are plenty of grammatical errors etc. The code works as it should and seems to produce results as it should.**

We thank the reviewer for this observation. We have carefully gone through the code, corrected grammatical and stylistic issues, and added author information as appropriate.

References

- Allen, K. R., Smith, K. A., Bird, L. A., Tenenbaum, J. B., Makin, T. R., & Cowie, D. (2024). Lifelong learning of cognitive styles for physical problem-solving: The effect of embodied experience. *Psychonomic Bulletin and Review*, *31*(3), 1364–1375. <https://doi.org/10.3758/s13423-023-02400-4>
- Aqil, M., Knapen, T., & Dumoulin, S. O. (2021). Divisive normalization unifies disparate response signatures throughout the human visual hierarchy. *Proceedings of the National Academy of Sciences of the United States of America*, *118*(46). <https://doi.org/10.1073/pnas.2108713118>
- Arbuckle, S. A., Pruszynski, J. A., & Diedrichsen, J. (2022). Mapping the Integration of Sensory Information across Fingers in Human Sensorimotor Cortex. *The Journal of Neuroscience*, *42*(26), 5173–5185. <https://doi.org/10.1523/JNEUROSCI.2152-21.2022>
- Ariani, G., Pruszynski, J. A., & Diedrichsen, J. (2022). Motor planning brings human primary somatosensory cortex into action-specific preparatory states. *eLife*, *11*, 1–20. <https://doi.org/10.7554/eLife.69517>
- Brouwer, G. J., Arnedo, V., Offen, S., Heeger, D. J., & Grant, A. C. (2015). Normalization in human somatosensory cortex. *Journal of Neurophysiology*, *114*(5), 2588–2599. <https://doi.org/10.1152/jn.00939.2014>
- Burg, M. F., Cadena, S. A., Denfield, G. H., Walker, E. Y., Tolia, A. S., Bethge, M., & Ecker, A. S. (2021). Learning divisive normalization in primary visual cortex. In *PLoS Computational Biology* (Vol. 17, Issue 6). <https://doi.org/10.1371/journal.pcbi.1009028>
- Carandini, M., & Heeger, D. J. (2012). Normalization as a canonical neural computation. *Nature Reviews Neuroscience*, *13*(1), 51–62. <https://doi.org/10.1038/nrn3136>
- Chen, J. L., Margolis, D. J., Stankov, A., Sumanovski, L. T., Schneider, B. L., & Helmchen, F. (2015). Pathway-specific reorganization of projection neurons in somatosensory cortex during learning. *Nature Neuroscience*, *18*(8), 1101–1108. <https://doi.org/10.1038/nn.4046>
- Dempsey-Jones, H., Themistocleous, A. C., Carone, D., Ng, T. W. C., Harrar, V., & Makin, T. R. (2019). Blocking tactile input to one finger using anaesthetic enhances touch perception and learning in other fingers. *Journal of Experimental Psychology: General*, *148*(4), 713–727. <https://doi.org/10.1037/xge0000514>
- Downey, J. E., Schone, H. R., Foldes, S. T., Greenspon, C., Liu, F., Verbaarschot, C., Biro, D., Satzer, D., Moon, C. H., Coffman, B. A., Yousofzadeh, V., Fields, D., Hobbs, T. G., Okorokova, E., Tyler-Kabara, E. C., Warnke, P. C., Gonzalez-Martinez, J., Hatsopoulos, N. G., Bensmaia, S. J., ... Collinger, J. L. (2024). A Roadmap for Implanting Electrode Arrays to Evoke Tactile Sensations Through Intracortical Stimulation. *Human Brain Mapping*, *45*(18). <https://doi.org/10.1002/hbm.70118>
- Ebrahimi, S., & Ostry, D. J. (2024). The human somatosensory cortex contributes to the encoding of newly learned movements. *Proceedings of the National Academy of Sciences*, *121*(6), 1–9. <https://doi.org/10.1073/pnas.2316294121>
- Ejaz, N., Hamada, M., & Diedrichsen, J. (2015). Hand use predicts the structure of representations in sensorimotor cortex. *Nature Neuroscience*, *18*(7), 1034–1040. <https://doi.org/10.1038/nn.4038>

- Feldman, D. E. (2009). Synaptic mechanisms for plasticity in neocortex. *Annual Review of Neuroscience*, 32, 33–55. <https://doi.org/10.1146/annurev.neuro.051508.135516>
- Gainey, M. A., & Feldman, D. E. (2017). Multiple shared mechanisms for homeostatic plasticity in rodent somatosensory and visual cortex. *Philosophical Transactions of the Royal Society B: Biological Sciences*, 372(1715), 20160157. <https://doi.org/10.1098/rstb.2016.0157>
- Gentile, G., Guterstam, A., Brozzoli, C., & Ehrsson, H. H. (2013). Disintegration of Multisensory Signals from the Real Hand Reduces Default Limb Self-Attribution: An fMRI Study. *Journal of Neuroscience*, 33(33), 13350–13366. <https://doi.org/10.1523/JNEUROSCI.1363-13.2013>
- Glasser, M. F., Coalson, T. S., Robinson, E. C., Hacker, C. D., Harwell, J., Yacoub, E., Ugurbil, K., Andersson, J., Beckmann, C. F., Jenkinson, M., Smith, S. M., & Van Essen, D. C. (2016). A multi-modal parcellation of human cerebral cortex. *Nature*, 536(7615), 171–178. <https://doi.org/10.1038/nature18933>
- Hahamy, A., Macdonald, S. N., van den Heiligenberg, F., Kieliba, P., Emir, U., Malach, R., Johansen-Berg, H., Brugger, P., Culham, J. C., & Makin, T. R. (2017). Representation of Multiple Body Parts in the Missing-Hand Territory of Congenital One-Handers. *Current Biology*, 27(9), 1350–1355. <https://doi.org/10.1016/j.cub.2017.03.053>
- Hahamy, A., & Makin, T. R. (2019). Remapping in Cerebral and Cerebellar Cortices Is Not Restricted by Somatotopy. *The Journal of Neuroscience*, 39(47), 9328–9342. <https://doi.org/10.1523/JNEUROSCI.2599-18.2019>
- Heeger, D. J. (1992). Normalization of cell responses in cat striate cortex. *Visual Neuroscience*, 9(2), 181–197. <https://doi.org/10.1017/S0952523800009640>
- Keck, T., Toyozumi, T., Chen, L., Doiron, B., Feldman, D. E., Fox, K., Gerstner, W., Haydon, P. G., Hübener, M., Lee, H.-K., Lisman, J. E., Rose, T., Sengpiel, F., Stellwagen, D., Stryker, M. P., Turrigiano, G. G., & van Rossum, M. C. (2017). Integrating Hebbian and homeostatic plasticity: the current state of the field and future research directions. *Philosophical Transactions of the Royal Society B: Biological Sciences*, 372(1715), 20160158. <https://doi.org/10.1098/rstb.2016.0158>
- Lee, H. K., & Kirkwood, A. (2019). Mechanisms of Homeostatic Synaptic Plasticity in vivo. *Frontiers in Cellular Neuroscience*, 13(December), 1–7. <https://doi.org/10.3389/fncel.2019.00520>
- LeMessurier, A. M., & Feldman, D. E. (2018). Plasticity of population coding in primary sensory cortex. *Current Opinion in Neurobiology*, 53, 50–56. <https://doi.org/10.1016/j.conb.2018.04.029>
- Mayzel, J., & Schneidman, E. (2024). Homeostatic synaptic normalization optimizes learning in network models of neural population codes. *ELife*, 13, 2023.03.05.530392. <https://doi.org/10.7554/eLife.96566>
- Nevalainen, P., Lauronen, L., & Pihko, E. (2014). Development of human somatosensory cortical functions - What have we learned from magnetoencephalography: A review. *Frontiers in Human Neuroscience*, 8(MAR), 1–15. <https://doi.org/10.3389/fnhum.2014.00158>
- Petersen, C. C. H. (2019). Sensorimotor processing in the rodent barrel cortex. *Nature Reviews Neuroscience*, 20(9), 533–546. <https://doi.org/10.1038/s41583-019-0200-y>

- Rao, R. P. N. (2024). A sensory–motor theory of the neocortex. *Nature Neuroscience*, 27(7), 1221–1235. <https://doi.org/10.1038/s41593-024-01673-9>
- Reed, C. L., Shoham, S., & Halgren, E. (2004). Neural Substrates of Tactile Object Recognition: An fMRI Study. *Human Brain Mapping*, 21(4), 236–246. <https://doi.org/10.1002/hbm.10162>
- Rioutl-Pedotti, M. S., Friedman, D., & Donoghue, J. P. (2000). Learning-induced LTP in neocortex. *Science*, 290(5491), 533–536. <https://doi.org/10.1126/science.290.5491.533>
- Root, V., Muret, D., Arribas, M., Amoruso, E., Thornton, J., Tarall-Jozwiak, A., Tracey, I., & Makin, T. R. (2022). Complex pattern of facial remapping in somatosensory cortex following congenital but not acquired hand loss. *ELife*, 11, 1–31. <https://doi.org/10.7554/ELIFE.76158>
- Ruben, J., Schwiemann, J., Deuchert, M., Meyer, R., Krause, T., Curio, G., Villringer, K., Kurth, R., & Villringer, A. (2001). Somatotopic organization of human secondary somatosensory cortex. *Cerebral Cortex (New York, N.Y. : 1991)*, 11(5), 463–473. <https://doi.org/10.1093/cercor/11.5.463>
- Sanders, Z., Dempsey-Jones, H., Wesselink, D. B., Edmondson, L. R., Puckett, A. M., Saal, H. P., & Makin, T. R. (2023). Similar somatotopy for active and passive digit representation in primary somatosensory cortex. *Human Brain Mapping*, 44(9), 3568–3585. <https://doi.org/10.1002/hbm.26298>
- Schone, H. R., Maimon Mor, R. O., Kollamkulam, M., Szymanska, M. A., Gerrand, C., Woollard, A., Kang, N. V., Baker, C. I., & Makin, T. R. (2025). Stable Cortical Body Maps Before and After Arm Amputation. *BioRxiv : The Preprint Server for Biology*, 0–47. <https://doi.org/10.1101/2023.12.13.571314>
- Shelchkova, N. D., Downey, J. E., Greenspon, C. M., Okorokova, E. V., Sobinov, A. R., Verbaarschot, C., He, Q., Sponheim, C., Tortolani, A. F., Moore, D. D., Kaufman, M. T., Lee, R. C., Satzer, D., Gonzalez-Martinez, J., Warnke, P. C., Miller, L. E., Boninger, M. L., Gaunt, R. A., Collinger, J. L., ... Bensmaia, S. J. (2023). Microstimulation of human somatosensory cortex evokes task-dependent, spatially patterned responses in motor cortex. *Nature Communications*, 14(1), 7270. <https://doi.org/10.1038/s41467-023-43140-2>
- Siegel, J. S., Power, J. D., Dubis, J. W., Vogel, A. C., Church, J. A., Schlaggar, B. L., & Petersen, S. E. (2014). Statistical improvements in functional magnetic resonance imaging analyses produced by censoring high-motion data points. *Human Brain Mapping*, 35(5), 1981–1996. <https://doi.org/10.1002/hbm.22307>
- Striem-Amit, E., Vannuscorps, G., & Caramazza, A. (2018). Plasticity based on compensatory effector use in the association but not primary sensorimotor cortex of people born without hands. *Proceedings of the National Academy of Sciences*, 115(30), 7801–7806. <https://doi.org/10.1073/pnas.1803926115>
- Tucciarelli, R., Ejaz, N., Wesselink, D. B., Kolli, V., Hodgetts, C. J., Diedrichsen, J., & Makin, T. R. (2024). Does Ipsilateral Remapping Following Hand Loss Impact Motor Control of the Intact Hand? *The Journal of Neuroscience*, 44(4), e0948232023. <https://doi.org/10.1523/JNEUROSCI.0948-23.2023>
- Turrigiano, G. G., & Nelson, S. B. (2004). Homeostatic plasticity in the developing nervous system. *Nature Reviews Neuroscience*, 5(2), 97–107. <https://doi.org/10.1038/nrn1327>
- Wesselink, D. B., van den Heiligenberg, F. M., Ejaz, N., Dempsey-Jones, H., Cardinali, L., Tarall-Jozwiak, A., Diedrichsen, J., & Makin, T. R. (2019). Obtaining and maintaining cortical hand

representation as evidenced from acquired and congenital handlessness. *ELife*, 8, 1–19. <https://doi.org/10.7554/eLife.37227>

Xu, T., Yu, X., Perlik, A. J., Tobin, W. F., Zweig, J. A., Tennant, K., Jones, T., & Zuo, Y. (2009). Rapid formation and selective stabilization of synapses for enduring motor memories. *Nature*, 462(7275), 915–919. <https://doi.org/10.1038/nature08389>

Yousry, T. A., Schmid, U. D., Alkadhi, H., Schmidt, D., Peraud, A., Buettner, A., & Winkler, P. (1997). Localization of the motor hand area to a knob on the precentral gyrus. A new landmark. *Brain*, 120(1), 141–157. <https://doi.org/10.1093/brain/120.1.141>